# Nonparametric Learning of Two-Layer ReLU Residual Units

**Zhunxuan Wang**                                                                                     *zhunxuan.wang@gmail.com*
*Amazon**
*London EC2A 2FA, United Kingdom*

**Linyun He**                                                                                                 *lhe85@gatech.edu*
*Georgia Institute of Technology*
*Atlanta, GA 30332, United States*

**Chunchuan Lyu**                                                                                  *chunchuan.lv@gmail.com*
*Instituto de Telecomunicacões Torre Norte*
*1049-001 Lisbon, Portugal*

**Shay B. Cohen**                                                                                    *scohen@inf.ed.ac.uk*
*University of Edinburgh*
*Edinburgh EH8 9AB, United Kingdom*

**Reviewed on OpenReview:** *https://openreview.net/forum?id=YiOIOvqJOn*

## Abstract

We describe an algorithm that learns two-layer residual units using rectified linear unit (ReLU) activation: suppose the input $\mathbf{x}$ is from a distribution with support space $\mathbb{R}^d$ and the ground-truth generative model is a residual unit of this type, given by $\mathbf{y} = \boldsymbol{B}^* \left[ (\boldsymbol{A}^*\mathbf{x})^+ + \mathbf{x} \right]$, where ground-truth network parameters $\boldsymbol{A}^* \in \mathbb{R}^{d \times d}$ represent a full-rank matrix with nonnegative entries and $\boldsymbol{B}^* \in \mathbb{R}^{m \times d}$ is full-rank with $m \geq d$ and for $\boldsymbol{c} \in \mathbb{R}^d$, $[\boldsymbol{c}^+]_i = \max\{0, c_i\}$. We design layer-wise objectives as functionals whose analytic minimizers express the exact ground-truth network in terms of its parameters and nonlinearities. Following this objective landscape, learning residual units from finite samples can be formulated using convex optimization of a nonparametric function: for each layer, we first formulate the corresponding empirical risk minimization (ERM) as a positive semi-definite quadratic program (QP), then we show the solution space of the QP can be equivalently determined by a set of linear inequalities, which can then be efficiently solved by linear programming (LP). We further prove the strong statistical consistency of our algorithm, and demonstrate its robustness and sample efficiency through experimental results on synthetic data and a set of benchmark regression datasets.[1]

## 1 Introduction

Neural networks have achieved remarkable success in various fields such as computer vision (LeCun et al., 1998; Krizhevsky et al., 2012; He et al., 2016a) and natural language processing (Kim, 2014; Sutskever et al., 2014). This success is largely due to the strong expressive power of neural networks (Bengio & Delalleau, 2011), where nonlinear activation units, such as rectified linear units (ReLU; Nair & Hinton 2010) and hyperbolic tangents (tanh) play a vital role to ensure the large learning capacity of the networks (Maas et al., 2013). However, the nonlinearity of neural networks makes them significantly more difficult to train than linear models (Livni et al., 2014). Therefore, with the development of neural network applications, finding efficient algorithms with provable properties to train such nontrivial neural networks has become an important and a relatively new goal.

---

[*]Work done mostly at University of Edinburgh.
[1]Our code is available at `https://github.com/uuzeeex/relu-resunit-learning`.

Residual networks, or ResNets (He et al., 2016a), are a class of deep neural networks that adopt skip connections to feed values between nonadjacent layers, where skipped layers may contain nonlinearities in between. Without loss of expressivity, ResNets avoid the vanishing gradient problem by directly passing gradient information from previous layers to current layers where otherwise gradients might vanish without skipping. In practice, ResNets have shown strong learning efficiency in several tasks, e.g. achieving at least 93% test accuracy on CIFAR-10 classification, lowering single-crop error to 20.1% on the 1000-class ImageNet dataset (Russakovsky et al., 2015; He et al., 2016b; Allen-Zhu & Li, 2019).

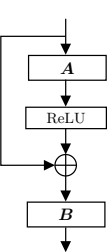

Common ResNets are often aggregated by many repeated shallow networks, where each network acts as a minimal unit with this kind of skip propagation, named a residual unit (He et al., 2016b). Given the flexibility and simplicity of residual units, much theoretical work has been devoted to studying them and developing training algorithms for them in a way that sidesteps from the standard backpropagation regime and provides guarantees on the quality of estimation (see Section 1.1). In this paper, we propose algorithms that learn a general class of single-skip two-layer residual units with ReLU activation as shown on the right by the equation: $\mathbf{y} = \boldsymbol{B}\left[(\boldsymbol{A}\mathbf{x})^+ + \mathbf{x}\right]$, where for a scalar $c$, $c^+ = \max\{0, c\}$ (for a vector, this maximization is applied coordinate-wise), $\mathbf{x}$ is a random vector as the network input with support space $\mathbb{R}^d$, and $\boldsymbol{A} \in \mathbb{R}^{d \times d}$ with nonnegative entries and $\boldsymbol{B} \in \mathbb{R}^{m \times d}$ are weight matrices of *layer 1* and *layer 2*, respectively.

Compared to previous work (Ge et al., 2019; 2018; Zhang et al., 2019; Wu et al., 2019; Tian, 2017; Du et al., 2018; Brutzkus & Globerson, 2017; Soltanolkotabi, 2017; Li & Yuan, 2017; Zhong et al., 2017), the introduction of residual connections simplifies the recovery of the network parameters by removing the permutation and scaling invariance. However, the naive mean square error minimization used in estimating the parameters remains nonconvex. Unlike most previous work, we do not assume a specific input distribution nor that the distribution is symmetric (Ge et al., 2019; Du & Goel, 2018).

We show that under regularity conditions on the weights of the residual unit, the problem of learning the ReLU unit can be formulated through quadratic programming (QP). We use nonparametric estimation (Guntuboyina & Sen, 2018) to estimate the ReLU function values in the networks. We further rewrite our constructed quadratic programs to linear programs (LPs). The LP formulation is simpler to optimize and has the same solution space as the QP for the network parameters.

## 1.1 Related Work

The study of two-layer ReLU networks (and two-layer networks in general, with one hidden layer) has received much attention in recent years because of the balance they present between their practical and theoretical properties. On one hand, two-layer networks represent non-convex learning problems and embody many of the same difficulties that we have with several layers. On the other hand, they are simple enough to study from a theoretical perspective. Arora et al. (2014) recover a multi-layer generative network with sparse connections and Livni et al. (2014) study the learning of multi-layer neural networks with polynomial activation. Goel et al. (2018) learns a one-layer convolution network with a perceptron-like rule. They prove the correctness of an iterative algorithm for exact recovery of the target network. Rabusseau et al. (2019) describe a spectral algorithm for two-layer linear networks. Recent work has connected optimization and two-layer network learning (Ergen & Pilanci, 2021a; Sahiner et al., 2021; Ergen & Pilanci, 2021b; Pilanci & Ergen, 2020; Mishkin et al., 2022) and has shown how to optimize networks layer by layer (Belilovsky et al., 2019) or their relationship to linear classification (Yang et al., 2021). Arjevani & Field (2021) study the symmetries that two-layer ReLU networks present.

For learning a one-layer ReLU network, Wu et al. (2019) optimize the norm and direction of neural network weight vectors separately, and Zhang et al. (2019) use gradient descent with a specific initialization. For learning a two-layer ReLU network, Ge et al. (2018) redesign the optimization landscape such that it is more amenable to theoretical analysis and Ge et al. (2019) use a moment-based method for estimating a neural network (Janzamin et al., 2015). Many others have studied ReLU networks in various settings (Tian, 2017; Du et al., 2018; Brutzkus & Globerson, 2017; Soltanolkotabi, 2017; Li & Yuan, 2017; Zhong et al., 2017; Goel & Klivans, 2019).

The study of ReLU networks with two hidden layers has also been gaining attention (Goel & Klivans, 2019; Allen-Zhu et al., 2019), with a focus on Probably Approximately Correct (PAC) learning (Valiant, 1984). In relation to our work, Allen-Zhu & Li (2019) examined the PAC-learnable function of a specific three-layer neural network with residual connections. Their work differs from ours in two aspects. First, their learnable functions include a smaller (in comparison to the student network) three-layer residual network. Second, the assumptions they make on their three-layer model are rather different than ours.

In relation to nonparametric estimation, Guntuboyina & Sen (2018) treat the final output of a shape-restricted regressor as a parameter, placing some restrictions on the type of function that can be estimated (such as convexity). They provide solutions for estimation with isotonic regression (Brunk, 1955; Ayer et al., 1955; van Eeden, 1956), convex regression (Seijo & Sen, 2011), shape-restricted additive models (Meyer, 2013; Chen & Samworth, 2016) and shape-restricted single index models (Kakade et al., 2011; Kuchibhotla et al., 2021).

### 1.2 Main Results

We design quadratic objective functionals with a linear bounded feasible domain that take network parameters and activation functions as variables to estimate the ground-truth network parameters and nonlinearities. The values of the objectives are moments over the input distribution.

In Theorems 4.1 and 5.1, we show that if a ground-truth residual unit has nondegenerate weights in both layers and nonnegative weights in layer 1, then there are quadratic functionals (see Eqs. (2) and (6)) with a domain defined by linear constraints whose minimizers *a)* are unique, and are the exact ground-truth, or *b)* are not unique, but can be adjusted to the exact ground-truth by a process of rescaling (see Theorem 5.2).

The minimizers above use moments of the input distribution. In practice, exact moments from the underlying distribution are not available. To address that, we use empirical risk minimization (ERM) of the moment-valued objectives by generated samples. With functions as variables in moment-valued objectives being optimized nonparametrically, the empirical objectives become quadratic functions with linear constraints (quadratic program; QP).

We further show the convexity of our QP which guarantees its solution in polynomial time w.r.t. sample size and dimension. With the available QP solution, our learning algorithm is strongly consistent (Theorem 6.3).[2] We show that if samples are generated by a ground-truth residual unit that has nondegenerate weights in both layers and nonnegative weights in layer 1, then our algorithm learns a network closer to the exact ground-truth (in Frobenius norm) as sample size grows.

**Roadmap:** Assume $\boldsymbol{A}$ and $\boldsymbol{B}$ are student network weights of layer 1 and 2. We first give a warm-up vanilla linear regression approach only knowing $\boldsymbol{A}$ is in $\mathbb{R}^{d \times d}$ and $\boldsymbol{B}$ is in $\mathbb{R}^{m \times d}$ (Section 3). We then move to the details of our nonparametric learning for layer 2 (Section 4), and similarly, layer 1 (Section 5). We formalize the quadratic functional minimization result above, showing how nonparametric learning allows us to select the values of $\boldsymbol{A}$ and $\boldsymbol{B}$ from reduced spaces that are seeded by $\boldsymbol{A}^*$ and $\boldsymbol{B}^*$, under which we can use linear regression (LR) effectively compared to vanilla LR. In Section 6, we describe the strong consistency of our methods for respective layers, formalizing the consistency result using the continuous mapping theorem (Mann & Wald, 1943). In Appendix I, we describe a sample complexity analysis of our algorithm. We further discuss the multi-layer case in Appendix C.

## 2 Preliminaries

We describe the notations used in this paper, introduce the model and its underlying assumptions, and state conditions which simplify the problem but which can be removed without loss of learnability.

**Notation** The ReLU residual units we use take a vector $\boldsymbol{x} \in \mathbb{R}^d$ as input and return a vector $\boldsymbol{y} \in \mathbb{R}^m$. We use $\boldsymbol{A}^* \in \mathbb{R}^{d \times d}$ and $\boldsymbol{B}^* \in \mathbb{R}^{m \times d}$ to denote the ground-truth network parameters for layer 1 and 2, respectively. We use circumflex to denote predicted terms (e.g. an empirical objective function $\hat{f}$, estimated layer 1 weights

---

[2]By *strong* consistency, we are referring to almost sure convergence of the estimator to the true parameters, rather than convergence in probability, which is a weaker type of probabilistic convergence.

$\hat{\boldsymbol{A}}$). We use $n$ to denote the number of independently and identically distributed (i.i.d.) samples available for the training algorithm, $\{\boldsymbol{x}^{(i)}, \boldsymbol{y}^{(i)}\}$ to denote the $i$-th sample drawn. For an integer $k$, We define $[k]$ to be $\{1, 2, \ldots, k\}$, and $\boldsymbol{e}^{(j)}$ as the standard basis vector with a 1 at position $j$. All scalar-based operators are element wise in the case of vectors or matrices unless specified otherwise. For a matrix $\boldsymbol{G}$ and an index $j$, $\boldsymbol{G}_{:,j}$ denotes its $j$th column and $\boldsymbol{G}_{j,:}$ denotes its $j$th row. We use $\mathbf{x}$ and $\mathbf{y}$ to refer to the input and output vectors, respectively, as random variables.

**Linear Regression:** Linear regression, or LR, in this paper refers to the problem of estimating $\boldsymbol{L}$ for unbiased model $\boldsymbol{y} = \boldsymbol{L}\boldsymbol{x}$. In this case, estimation is done by minimizing the empirical risk $\hat{R}(\boldsymbol{L}) = \frac{1}{2n} \sum_{i \in [n]} \left\| \boldsymbol{L}\boldsymbol{x}^{(i)} - \boldsymbol{y}^{(i)} \right\|^2$ – using linear least squares (LLS). Its solution has a closed form which we use as given. We refer the reader to Hamilton (2020) for more information.

**Models, Assumptions and Conditions** Following previous work (Livni et al., 2014), we assume a given neural network structure specifies a hypothesis class that contains all the networks conforming to this structure. Learning such a class means using training samples to find a set of weights such that the neural networks predictions generalize well to unseen samples, where both the training and the unseen samples are drawn from an unobserved ground-truth distribution. In this paper, the hypothesis class is given by ReLU residual units and we assume it has sufficient expressive power to fit the ground-truth model. More specifically, we discuss the *realizable* case of learning, in which the ground-truth model is set to be a residual unit taken from the hypothesis class with the form:

$$\mathbf{y} = \boldsymbol{B}^* \left[ \left( \boldsymbol{A}^* \mathbf{x} \right)^+ + \mathbf{x} \right], \tag{1}$$

and is used to draw samples for learning. Unlike other multi-layer ReLU-affine models which do not apply skip connections, we cannot permute the weight matrices of the residual unit and retain the same function because of skip-adding $\mathbf{x}$ (it breaks symmetry). This helps us circumvent issues of identifiability,[3] and allows us to precisely estimate the ground-truth weight matrices $\boldsymbol{A}^*$ and $\boldsymbol{B}^*$.

Our general approach for residual unit layer 2 learns a scaled ground-truth weight matrix that also minimizes the layer 2 objective. The existence of such scaled equivalence of our layer 2 approach comes from what is defined below.

**Definition 2.1** (component-wise scale transformation). A matrix $\boldsymbol{A} \in \mathbb{R}^{d \times d}$ is said to be a scale transformation w.r.t. the $j$-th component if $\left( \boldsymbol{A}_{j,:} \right)^\top = A_{j,j} \cdot \boldsymbol{e}^{(j)}$.

Additionally, estimation is more complex when the layer 2 weight matrix $\boldsymbol{B}^*$ is nonsquare. For simplicity of our algorithm presentation for layer 2, we use Condition 2.1 in the following sections.[4]

**Condition 2.1** (layer 2 objective minimizer unique). $\boldsymbol{A}^*$ is not a scale transformation w.r.t. any components and $\boldsymbol{B}^*$ is a square matrix, i.e. $m = d$.

## 3 Warm-Up: Vanilla Linear Regression

Consider a ground-truth two-layer residual unit. If we assume that the inputs only contain vectors with negative entries, i.e. $\mathbf{x} < 0$, the effect of the ReLU function in the residual unit then disappears because of the nonnegativity of $\boldsymbol{A}^*$. The residual unit turns into linear model $\mathbf{y} = \boldsymbol{B}^* \mathbf{x}$. Thus, direct LR on samples with negative inputs can learn the exact ground-truth layer 2 parameter $\boldsymbol{B}^*$ with at least $d$ samples, when formulating the LR as a solvable full-rank linear equation system.

On the contrary, if the inputs only contain vectors with positive entries, i.e. $\mathbf{x} > 0$, all the neurons in the hidden layer are then activated by the ReLU and the nonlinearity is eliminated. The residual unit in this case turns into $\mathbf{y} = \boldsymbol{B}^* \left( \boldsymbol{A}^* + \boldsymbol{I}_d \right) \mathbf{x}$. Taking the value that left-multiplies $\mathbf{x}$ as a single weight matrix $\boldsymbol{D}^*$, this is also a linear model. Direct LR on at least $d$ samples with positive inputs by the residual unit can

---

[3]Here, we are referring to the ability to identify the true model parameters using infinite samples.
[4]In Appendix E, we show that estimation of $\boldsymbol{B}^*$ remains possible without satisfying Condition 2.1.

learn the exact $\boldsymbol{D}^*$. Since we have the access to the exact $\boldsymbol{B}^*$, solving for the exact $\boldsymbol{A}^*$ can be accomplished through solving a full-rank linear equation system $\boldsymbol{B}^* \cdot \tilde{\boldsymbol{A}} = \boldsymbol{D}^*$, where the unique solution $\tilde{\boldsymbol{A}} = \boldsymbol{A}^* + \boldsymbol{I}_d$.

While simple, this vanilla LR approach requires a large number of redundant samples, since sampled inputs usually have a small proportion of fully negative/positive vectors. Taking random input vectors i.i.d. with respect to each entry as an example, the probability of sampling a vector with all negative entries is $p_-^d$, where $p_-$ is the probability of sampling a negative vector entry, then the expected number of samples to get one negative vector is $1/p_-^d$. Denoting $p_+$ similarly, $1/p_+^d$ samples are expected for a positive vector. In addition, each LR step in this approach requires $d$ such samples respectively to make the linear equation system full-rank, which implies the expected sample size to be $d$-exponential $d \cdot \left(1/p_-^d + 1/p_+^d\right)$. For other common input distributions such as Gaussian, the proportions of fully negative/positive vectors in sampled inputs are also expected to decrease exponentially as $d$ grows, as high-dimensional Gaussian random vectors are essentially concentrated uniformly in a sphere (Johnstone, 2007). Technical and experimental details about sample size expectations and the vanilla LR algorithm are further discussed in Appendix D.

## 4 Nonparametric Learning: Layer 2

We present how we learn a residual unit layer 2 under Condition 2.1 (estimating $\boldsymbol{B}^*$): We first design an objective functional with the arguments being a matrix and a function. The objective uses expectation of a loss over the true distribution generating the data and is uniquely minimized by $[\boldsymbol{B}^*]^{-1}$ and a rectifier function (ReLU). We then formulate its ERM using nonparametric estimation as a standard convex QP, further simplified as an LP that (in the noiseless case) has the same solution as the QP to learn layer 2.

### 4.1 Objective Design and Landscape

Consider the formulation of a residual unit as in Eq. (1). It is possible to rewrite the model as equation: $\boldsymbol{C}^*\mathbf{y} = (\boldsymbol{A}^*\mathbf{x})^+ + \mathbf{x}$, where the output of the hidden neuron with skip addition is on both sides of the equation, and $\boldsymbol{C}^*\boldsymbol{B}^* = \boldsymbol{I}_d$. We aim to estimate the inverse of $\boldsymbol{B}^*$ by matrix variable $\boldsymbol{C}$ and the nonlinearity $\boldsymbol{x} \mapsto (\boldsymbol{A}^*\boldsymbol{x})^+$ by a function variable $h$. The objective is formulated as risk functional by the $L^2$ error between values respectively computed by $\boldsymbol{C}$ and $h$,

$$G_2\left(\boldsymbol{C}, h\right) = \frac{1}{2}\mathbb{E}_{\mathbf{x}}\left[\|h\left(\mathbf{x}\right) + \mathbf{x} - \boldsymbol{C}\mathbf{y}\|^2\right], \tag{2}$$

where the estimator $\boldsymbol{C} \in \mathbb{R}^{d \times m}$, the domain of $h$ is the nonnegative[5] continuous[6] $\mathbb{R}^d \to \mathbb{R}^d_{\geq 0}$ function space, written as $\mathbb{C}^0_{\geq 0}$ in shorthand. This objective is quadratic because the forward mapping $\boldsymbol{x} \mapsto h(\boldsymbol{x}) + \boldsymbol{x}$ and the backward mapping $\boldsymbol{y} \mapsto \boldsymbol{C}\boldsymbol{y}$ are both linear w.r.t. $\boldsymbol{C}$ and $h$, and the two are linearly combined in an $L^2$ norm. The objective in Eq. (2) is minimized by the ground-truth, i.e. $\boldsymbol{C}^*$ and $\boldsymbol{x} \mapsto (\boldsymbol{A}^*\boldsymbol{x})^+$, is one of its minimizers. However, it is not simple to describe other variable values that minimize the objective if any exist. We give that detail under Condition 2.1, the minimizer of $G_2$ is unique, as the exact ground-truth in the given domain.

**Theorem 4.1** (objective minimizer, layer 2). *Define $G_2(\boldsymbol{C}, h)$ as Eq. (2), where $\boldsymbol{C} \in \mathbb{R}^{d \times m}$, $h \in \mathbb{C}^0_{\geq 0}$. Then under Condition 2.1, $G_2(\boldsymbol{C}, h)$ reaches its zero minimum iff $\boldsymbol{C} = [\boldsymbol{B}^*]^{-1}$ and $h : \boldsymbol{x} \mapsto (\boldsymbol{A}^*\boldsymbol{x})^+$.*

Technical details are in Appendix F, where we use Lemma 4.2 (in Section 4.3) to prove a more general theorem which does not require Condition 2.1 and is sufficient for Theorem 4.1. In the next subsection, we construct the ERM of $G_2$ and present our convex QP formulation.

---

[5]Setting $h$ as nonnegative ensures that *a)* only ReLU nonlinearity minimizes $G_2$ (see Theorem 4.1), and *b)* $h$'s nonparametric estimator is linearly constrained (explained in Section 4.2).

[6]If $h$ is not a continuous function, only a null set of discontinuities is possible to make $G_2$ reach zero as its minimum. Setting $h$ as continuous simplifies our theoretical results that still strictly support empirical discussion.

## 4.2 ERM with Nonparametric Estimation is Convex QP

Consider the second layer objective (Eq. (2)) with nonnegative continuous function space as the domain of $h$. We follow Vapnik (1991) and define its standard empirical risk functional:

$$\hat{G}_2(\boldsymbol{C}, h) = \frac{1}{2n} \sum_{i \in [n]} \left\| h(\boldsymbol{x}^{(i)}) + \boldsymbol{x}^{(i)} - \boldsymbol{C}\boldsymbol{y}^{(i)} \right\|^2. \tag{3}$$

The function variable $h \in \mathbb{C}^0_{\geq 0}$ can be optimized either parametrically or nonparametrically. If we parameterize $h$, such nonlinearity w.r.t. its parameters would make $\hat{G}_2$ lose its quadratic form. Instead, we estimate $h$ nonparametrically: for each sample input $\boldsymbol{x}^{(i)}$, we introduce variables $\boldsymbol{\xi}^{(i)}$ that estimate mapped values by $h$. This avoids introducing nonlinearity to the objective and keeps $\hat{G}_2$ quadratic. On the other hand, the domain of $h$, i.e. nonnegative continuous function space, turns into a set of linear inequalities as constraints when optimizing the learning objective nonparametrically. In this sense, learning the second layer of the residual unit can be formulated as the following QP:

$$\min_{\boldsymbol{C}, \boldsymbol{\Xi}} \hat{G}_2^{\mathrm{NPE}}(\boldsymbol{C}, \boldsymbol{\Xi}) \coloneqq \frac{1}{2n} \sum_{i \in [n]} \left\| \boldsymbol{\xi}^{(i)} + \boldsymbol{x}^{(i)} - \boldsymbol{C}\boldsymbol{y}^{(i)} \right\|^2, \text{ s.t. } \boldsymbol{\xi}^{(i)} \geq \boldsymbol{0}, \forall\, i \in [n], \tag{4}$$

where $\boldsymbol{\Xi} = \{\boldsymbol{\xi}^{(i)}\}_{i=1}^n$ is the nonparametric estimator of $\boldsymbol{x} \mapsto (\boldsymbol{A}^*\boldsymbol{x})^+$.

**Nonparametric Estimation Validation:** A solution to the ERM with nonparametric estimation, i.e. the QP, is guaranteed to be sufficient for minimizing the standard empirical risk functional (Eq. (3)). More specifically, assuming $\boldsymbol{C}$ and $\boldsymbol{\Xi} = \{\boldsymbol{\xi}^{(i)}\}_{i=1}^n$ are a solution to layer 2 QP (Eq. (4)), we see now that $\boldsymbol{C}$ and $h \in \mathbb{C}^0_{\geq 0}$ such that $h(\boldsymbol{x}^{(i)}) = \boldsymbol{\xi}^{(i)}$ minimize the empirical risk functional Eq. (3). Conversely, a minimizer of the standard empirical risk Eq. (3), $\boldsymbol{C}$ and $h$, corresponds to a solution to layer 2 QP as we set $\boldsymbol{\xi}^{(i)} = h(\boldsymbol{x}^{(i)})$. Therefore, minimizing $\hat{G}_2$ and solving layer 2 QP are equivalent problems.

**Convexity:** The convexity of the QP: Eq. (4) is also guaranteed. First, its constraints are linear. Second, for each sample with index $i \in [n]$, the $L^2$ norm wraps linearity w.r.t. $\boldsymbol{C}$ and $\boldsymbol{\xi}^{(i)}$. Such formulation ensures the quadratic coefficient matrix is positive semidefinite. Thus, with the sum of convex functions still being convex, the QP objective (Eq. (4)) becomes convex. Even without the knowledge of how samples are generated, this QP would be a convex program. Strict proofs are in Appendix G.

## 4.3 LP Simplification

Consider single-sample error written as $g_2(\boldsymbol{C}, h; \boldsymbol{x}, \boldsymbol{y}) = \frac{1}{2} \|h(\boldsymbol{x}) + \boldsymbol{x} - \boldsymbol{C}\boldsymbol{y}\|^2$. If there is a feasible $\boldsymbol{C}$ such that $\boldsymbol{C}\boldsymbol{y} - \boldsymbol{x} \geq 0$ holds for all $\boldsymbol{x} \in \mathbb{R}^d$, then $\boldsymbol{C}$ and $h : \boldsymbol{x} \mapsto \boldsymbol{C}\boldsymbol{y} - \boldsymbol{x}$ always minimize $g_2$ as zero, and thereby minimize the layer 2 objective (Eq. (2)). Thus, we obtain a condition that is equivalent to $G_2$ reaching the minimum in Theorem 4.1 and avoids randomness (see Lemma 4.2).

**Lemma 4.2.** $G_2(\boldsymbol{C}, h)$ *reaches its zero minimum iff* $\boldsymbol{C}\boldsymbol{y} - \boldsymbol{x} \geq 0$ *holds for any* $\boldsymbol{x} \in \mathbb{R}^d$ *and its corresponding residual unit output* $\boldsymbol{y}$, *and* $h : \boldsymbol{x} \mapsto \boldsymbol{C}\boldsymbol{y} - \boldsymbol{x}$.

The pointwise (for every $\boldsymbol{x} \in \mathbb{R}^d$) satisfaction of the inequality in Lemma 4.2 describes the solution space for the minimization of $G_2$. The sufficiency of satisfying this inequality to minimize $G_2$ is directly obtained by assigning $h : \boldsymbol{x} \mapsto \boldsymbol{C}\boldsymbol{y} - \boldsymbol{x}$ where $\boldsymbol{C}$ complies with $\boldsymbol{C}\boldsymbol{y} - \boldsymbol{x} \geq 0$ for all $\boldsymbol{x}$ in the support space $\mathbb{R}^d$. The necessity of satisfying this inequality comes from its contraposition: If a $\boldsymbol{C}$ violates the inequality, there must be a non-null set of $\boldsymbol{x}$ that yield the violation due to the continuity of $\boldsymbol{C}\boldsymbol{y} - \boldsymbol{x}$. In this sense, the resulting $G_2$ value becomes nonzero.

Empirically speaking, we cannot solve an inequality that holds w.r.t. to every point in the support space if we only observe finite samples. We can only estimate $\boldsymbol{C}$ by solving the inequality that holds w.r.t. each sample. Following this, we formulate such estimation as to find a feasible point in the space defined by a set of linear inequalities, each of which corresponds to a sample: $\boldsymbol{C}\boldsymbol{y}^{(i)} - \boldsymbol{x}^{(i)} \geq 0$. Each point in the feasibility defined by the inequalities has a one-to-one correspondence in the solution space to layer 2 QP

---

**Algorithm 1** Learn a ReLU residual unit, layer 2.

---

1: **Input:** $\{(\boldsymbol{x}^{(i)}, \boldsymbol{y}^{(i)})\}_{i=1}^{n}$, samples drawn by Eq. (1).
2: **Output:** $\hat{\boldsymbol{B}}$, $\hat{\boldsymbol{\Xi}}$, a layer 2 and $\boldsymbol{x} \mapsto (\boldsymbol{A}^*\boldsymbol{x})^+$ estimate.
3: Go to line 4 if QP, line 5 if LP.
4: Solve QP: Eq. (4) and obtain a $\hat{G}_2^{\text{NPE}}$ minimizer, denoted by $\hat{\boldsymbol{C}}$, $\hat{\boldsymbol{\Xi}}$. Go to line 6.
5: Solve LP: Eq. (5) and obtain a minimizer $\hat{\boldsymbol{C}}$, then assign $\hat{\boldsymbol{\xi}}^{(i)} \leftarrow \hat{\boldsymbol{C}}\boldsymbol{y}^{(i)} - \boldsymbol{x}^{(i)}$.
6: **return** $\hat{\boldsymbol{C}}^{-1}$, $\hat{\boldsymbol{\Xi}}$.

---

(Eq. (4)): $\boldsymbol{C} \leftrightarrow (\boldsymbol{C}, \{\boldsymbol{C}\boldsymbol{y}^{(i)} - \boldsymbol{x}^{(i)}\}_{i \in [n]})$. The set of inequalities can be solved by a standard LP with a constant objective and with constraints that are inequalities, i.e.

$$\min_{\boldsymbol{C}} \text{ const, s.t. } \boldsymbol{C}\boldsymbol{y}^{(i)} - \boldsymbol{x}^{(i)} \geq 0, \, \forall \, i \in [n]. \tag{5}$$

With the one-to-one correspondences, LP: Eq. (5) and QP: Eq. (4) have identical solution spaces (feasible regions) for layer 2. Moreover, LP: Eq. (5) is also a convex program.

Algorithm 1 summarizes the layer 2 estimator: Simply solve the QP/LP[7] and return the inverse of $\hat{\boldsymbol{C}}$ as the estimate of layer 2 weights and $\hat{\boldsymbol{\Xi}}$ as $\boldsymbol{x} \mapsto (\boldsymbol{A}^*\boldsymbol{x})^+$ estimate. Regardless of time complexity, QP and LP in Algorithm 1 work equivalently since their solution spaces are equivalent to each other.

Our nonparametric learning directly finds a unique layer 2 estimate under Condition 2.1. In the general case without Condition 2.1 (discussed in Appendix E), nonparametric learning essentially reduces the possible values of $\boldsymbol{B}$ from $\mathbb{R}^{m \times d}$ to a $\boldsymbol{B}^*$ scale-equivalent matrix space, where LR uses sampled data much more efficiently than vanilla LR on layer 2 (Section 3).

## 5 Nonparametric Learning: Layer 1

With layer 2 learned, the outputs of the hidden neurons become observable. The two-layer problem is thereby reduced to single-layer. Consider a ground-truth single-layer model: $\mathbf{h} = (\boldsymbol{A}^*\mathbf{x})^+$. To construct a learning objective for this model, we rewrite the model as a nonlinearity plus a linear mapping by $\boldsymbol{A}^*$: $\mathbf{h} = (-\boldsymbol{A}^*\mathbf{x})^+ + \boldsymbol{A}^*\mathbf{x}$, where on both sides of the equation is the output of layer 1, and the nonlinearity is $\boldsymbol{x} \mapsto (-\boldsymbol{A}^*\boldsymbol{x})^+$. The objective of layer 1 is formulated as:

$$G_1(\boldsymbol{A}, r) = \frac{1}{2}\mathbb{E}_{\mathbf{x}}\left[\|r(\mathbf{x}) + \boldsymbol{A}\mathbf{x} - \mathbf{h}\|^2\right], \tag{6}$$

where $\boldsymbol{A} \in \mathbb{R}^{d \times d}$ is of the layer 1 weights estimator, the domain of $r$ is also $\mathbb{C}_{\geq 0}^0$. The minimizer $\boldsymbol{A}$ of the risk $G_1$ falls into a matrix space such that for any matrix $\boldsymbol{A}$ in the space, each row of $\boldsymbol{A}$ is a scaled-down version of the same row of $\boldsymbol{A}^*$ without changing the direction (Theorem 5.1).

**Theorem 5.1** (objective minimizer space, layer 1). *Define $G_1(\boldsymbol{A}, r)$ as Eq. (6), where $\boldsymbol{A} \in \mathbb{R}^{d \times d}$, $r \in \mathbb{C}_{\geq 0}^0$. Then $G_1(\boldsymbol{A}, r)$ reaches its zero minimum iff for each $j \in [d]$, $\boldsymbol{A}_{j,:} = k_j\boldsymbol{A}_{j,:}^*$ where $0 \leq k_j \leq 1$, and $r : \boldsymbol{x} \mapsto (\boldsymbol{A}^*\boldsymbol{x})^+ - \text{diag}(\boldsymbol{k}) \cdot \boldsymbol{A}^*\boldsymbol{x}$.*

The scale equivalence in the solution space is derived from a ReLU inequality: $(x)^+ \geq kx$ where $0 \leq k \leq 1$. Let the $j$-th row of $\boldsymbol{A}$ be a scaled-down version of $\boldsymbol{A}^*$, i.e. $\boldsymbol{A}_{j,:} = k_j\boldsymbol{A}_{j,:}^*$ where $0 \leq k_j \leq 1$. According to the inequality, we have $(\boldsymbol{A}_{j,:}^*\boldsymbol{x})^+ \geq k_j\boldsymbol{A}_{j,:}^*\boldsymbol{x}$, which indicates $(\boldsymbol{A}^*\boldsymbol{x})^+ \geq \text{diag}(\boldsymbol{k}) \cdot \boldsymbol{A}^*\boldsymbol{x}$. Thus, $r$ minimizing $G_1$ in Theorem 5.1 lies in feasibility $\mathbb{C}_{\geq 0}^0$ when $\boldsymbol{A} = \text{diag}(\boldsymbol{k}) \cdot \boldsymbol{A}^*$.

Due to the existence of scale equivalence, we must compute the scale factor to obtain the ground-truth weights $\boldsymbol{A}^*$. The scale factor $k_j$ is sufficiently obtainable with $(\boldsymbol{A}_{j,:}\mathbf{x})^+$ and $(\boldsymbol{A}_{j,:}^*\mathbf{x})^+$ observable: Conditioned on nonnegative ReLU input, we have a linear model $\boldsymbol{A}_{j,:}\mathbf{x} = k_j\boldsymbol{A}_{j,:}^*\mathbf{x}$ where $\boldsymbol{A}_{j,:}\mathbf{x}$ and $\boldsymbol{A}_{j,:}^*\mathbf{x}$ are observed.

---

[7]We use `CVX` (Grant & Boyd, 2014; 2008) that calls `SDPT3` (Toh et al., 1999) (a free solver under `GPLv3` license) and solves our convex QP/LP in polynomial time. See Appendix B for technical details.

---

**Algorithm 2** Learn a ReLU residual unit, layer 1.

---

1: **Input:** $\{(\boldsymbol{x}^{(i)}, \boldsymbol{h}^{(i)})\}_{i=1}^n$, layer 1 samples.
2: **Output:** $\hat{\boldsymbol{A}}$, a layer 1 estimate.
3: Solve QP: Eq. (7) or LP: Eq. (8) and obtain a $\hat{G}_1^{\text{NPE}}$ minimizer, denoted by $\hat{\boldsymbol{A}}$, $\hat{\boldsymbol{\Phi}}$. {$\hat{\boldsymbol{\Phi}}$ is no longer needed.}
4: **for all** $j \in [d]$ **do**
5: $\quad \hat{k}_j \leftarrow \text{LR}\{h_j^{(i)}, \hat{\boldsymbol{A}}_{j,:}\boldsymbol{x}^{(i)}\}_{h_j^{(i)}>0}$.
6: $\quad \hat{\boldsymbol{A}}_{j,:} \leftarrow \hat{\boldsymbol{A}}_{j,:}/\hat{k}_j$. {Rescale $\hat{\boldsymbol{A}}$.}
7: **end for**
8: **return** $\hat{\boldsymbol{A}}$.

---

Theorem 5.2 summarizes layer 1 scale factor property, allowing us to correct a minimizer of $G_1$ to the ground-truth weights $\boldsymbol{A}^*$ by computing a scalar for each row.

**Theorem 5.2** (scale factor, layer 1)**.** *Assume $\boldsymbol{A}$ is a minimizer of $G_1$. Then for any $j \in [d]$, $(\boldsymbol{A}_{j,:}\mathbf{x})^+ / (\boldsymbol{A}_{j,:}^*\mathbf{x})^+$ is always equal to the scale factor $k_j$ given that $(\boldsymbol{A}_{j,:}\mathbf{x})^+ > 0$.*

The elimination of randomness for $G_1$ follows the same pattern as layer 2. If there is a feasible $\boldsymbol{A}$ such that $(\boldsymbol{A}^*\boldsymbol{x})^+ - \boldsymbol{A}\boldsymbol{x} \geq 0$ holds for all $\boldsymbol{x} \in \mathbb{R}^d$, such $\boldsymbol{A}$ is a solution to our objective by Eq. (6). We can also have the following proposition that avoids randomness.

**Lemma 5.3.** *$G_1(\boldsymbol{A}, r)$ reaches its zero minimum iff $\boldsymbol{h} - \boldsymbol{A}\boldsymbol{x} \geq 0$ holds for any $\boldsymbol{x} \in \mathbb{R}^d$ and its corresponding hidden output $\boldsymbol{h}$, and $r : \boldsymbol{x} \mapsto \boldsymbol{h} - \boldsymbol{A}\boldsymbol{x}$.*

We now turn into empirical discussion. Similar to layer 2, we formulate layer 1 QP by its empirical objective with nonparametric estimation and linear constraints representing the nonnegativity of $r$:

$$\min_{\boldsymbol{A}, \boldsymbol{\Phi}} \hat{G}_1^{\text{NPE}}(\boldsymbol{A}, \boldsymbol{\Phi}) \coloneqq \frac{1}{2n} \sum_{i \in [n]} \left\| \boldsymbol{\phi}^{(i)} + \boldsymbol{A}\boldsymbol{x}^{(i)} - \boldsymbol{h}^{(i)} \right\|^2, \text{ s.t. } \boldsymbol{\phi}^{(i)} \geq \mathbf{0}, \forall i \in [n]. \tag{7}$$

where $\boldsymbol{\Phi} = \{\boldsymbol{\phi}^{(i)}\}_{i=1}^n$ is the function estimator of $\boldsymbol{x} \mapsto (\boldsymbol{A}^*\boldsymbol{x})^+ - \text{diag}(\boldsymbol{k}) \cdot \boldsymbol{A}^*\boldsymbol{x}$ where $\boldsymbol{k}$ refers to the scaling equivalence. Similarly, the solution space of QP: Eq. (7) can be represented by a set of linear inequalities as constraints of the following efficiently solvable LP

$$\min_{\boldsymbol{A}} \text{ const, s.t. } \boldsymbol{h}^{(i)} - \boldsymbol{A}\boldsymbol{x}^{(i)} \geq \mathbf{0}, \forall i \in [n]. \tag{8}$$

Algorithm 2 describes how a layer 1 is learned: First, a $G_1$ minimizer estimate $\hat{\boldsymbol{A}}$ is obtained by solving the QP/LP. Then for the $j$-th row, the scale factor $k_j$ is estimated by running LR on $h_j^{(i)}$ and $\hat{\boldsymbol{A}}_{j,:}\boldsymbol{x}^{(i)}$ s.t. $h_j^{(i)} > 0$ to correct $\hat{\boldsymbol{A}}$, as $\boldsymbol{A}_{j,:}\mathbf{x} = k_j\mathbf{h}_j$ is an unbiased linear model given that $\mathbf{h}_j > 0$. By nonparametric learning, the value space of $\boldsymbol{A}$ is reduced from $\mathbb{R}^{d \times d}$ to $\{\text{diag}(\boldsymbol{k}) \cdot \boldsymbol{A}^* \mid 0 \leq k_j \leq 1\}$, where LR uses sampled data much more efficiently than vanilla LR layer 1 (Section 3).

## 6 Full Algorithm and Analysis

The full algorithm concatenates Algorithms 1 and 2 in a layerwise fashion, with observations of input/output samples from the ground-truth network and follows these steps: *a*) Estimates layer 2 and nonlinearity: $\boldsymbol{x} \mapsto (\boldsymbol{A}^*\boldsymbol{x})^+$ by Algorithm 1. *b*) Estimates layer 1 by running Algorithm 2 on input samples and the nonlinearity estimate. It has provable guarantees. For empirical analysis, we use $\hat{\mathbf{C}}_n$ to denote the estimation of $\boldsymbol{C}^*$ from $n$ random samples. Similar notation is applied to other estimations. First of all, our methods to solve respective layers, Algorithms 1 and 2, are strongly consistent if any convex QPs/LPs involved can be solved exactly.

**Lemma 6.1** (layer 2 strong consistency)**.** *Under Condition 2.1, $\hat{\mathbf{C}}_n \xrightarrow{a.s.} \boldsymbol{C}^*$ and $\hat{\mathbf{B}}_n \xrightarrow{a.s.} \boldsymbol{B}^*$, $n \to \infty$.*

For $\hat{\mathbf{C}}_n$: In Appendix H, we prove its more general a.s. (almost sure) convergence without satisfying Condition 2.1, where the solution space to layer 2 objective is a non-compact continuous set where all the elements are scaling equivalences. We use Hausdorff distance (Rockafellar & Wets, 2009) as metric and prove that the empirical solution space a.s. converges to the theoretical solution space. Then the more general a.s. convergence holds with the strong consistency of layer 2 scale factor estimator where we use LR to estimate the scale factors.

For $\hat{\mathbf{B}}_n$: Using the continuous mapping theorem (Mann & Wald, 1943), we directly propagate the strong consistency of $\boldsymbol{C}^*$ estimator to its inverse $\boldsymbol{B}^*$'s estimator.

According to full algorithm description, layer 1 estimation uses $\hat{\mathbf{C}}_n \mathbf{y}^{(i)} - \mathbf{x}^{(i)}$ as the outputs where $i \in [n]$. Thus, the strong consistency of the hidden neuron estimator is also guaranteed by the continuous mapping theorem. Following the proof sketch of the $\hat{\mathbf{C}}_n$ a.s. convergence, we obtain the strong consistency of layer 1 estimator (See Lemma 6.2).

**Lemma 6.2** (layer 1 strong consistency). $\hat{\mathbf{A}}_n \xrightarrow{a.s.} \boldsymbol{A}^*$, $n \to \infty$.

By the continuous mapping theorem, the strong consistency of the full algorithm (Theorem 6.3), which is commonly defined by a loss function that is continuous on network weights, is implied by the strong consistency of network weights estimators (Lemmas 6.1 and 6.2). See Appendix H for proofs of strong consistency discussions in this section.

**Theorem 6.3** (strong consistency). *Define $L$ as $L^2$ output loss. $L(\hat{\mathbf{A}}_n, \hat{\mathbf{B}}_n) \xrightarrow{a.s.} 0$, $n \to \infty$.*

## 7 Experiments

In our experiments, we describe a first set of synthetic dataset experiments, that show that our algorithm identifies the true parameters from which the data were generated (Section 7.1), and then a second set of experiments that use our algorithm on standard real-world benchmark datasets (Section 7.2). In the appendices we provide further experiments. For example, in Appendix J, we describe what happens when the conditions required for the correctness of our algorithms are not satisfied.

### 7.1 Synthetic Data Experiments

We provide experimental analysis to demonstrate the effectiveness and robustness of our approach in comparison to stochastic gradient descent (SGD) on $L^2$ output loss: $L(\boldsymbol{A}, \boldsymbol{B}) = \frac{1}{2} \mathbb{E}_{\mathbf{x}} \|\hat{\mathbf{y}} - \mathbf{y}\|^2$, where we parameterize the output prediction by $\hat{\mathbf{y}} = \boldsymbol{B} \left[ (\boldsymbol{A}\mathbf{x})^+ + \mathbf{x} \right]$. Our proposed methods outperform SGD in terms of sample efficiency and robustness to different network weights and noise strengths, which indicates a poor optimization landscape of $L^2$ output loss for ReLU residual units.

**Setup**: The ground-truth weights are generated through i.i.d. folded standard Gaussian[8] and standard Gaussian for layer 1 and 2 respectively, i.e. $\mathbf{A}^* \overset{\text{i.i.d.}}{\sim} |\mathcal{N}|(0,1)$, $\mathbf{B}^* \overset{\text{i.i.d.}}{\sim} \mathcal{N}(0,1)$. The input distribution is set to be an i.i.d. zero mean Gaussian-uniform equal mixture $\mathcal{N}(-0.1, 1) - \mathcal{U}(-0.9, 1.1)$. SGD is conducted on mini-batch empirical losses of $L(\boldsymbol{A}, \boldsymbol{B})$ with batch size 32 for 256 epochs in each learning trial. We apply time-based learning rate decay $\eta = \eta_0 / (1 + \gamma \cdot T)$ with initial rate $\eta_0 = 10^{-3}$ and decay rate $\gamma = 10^{-5}$, where $T$ is the epoch number. The above hyperparameters are tuned to outperform other hyperparameters in learning ReLU residual units in terms of output errors.

**Evaluation**: We use relative errors to measure the accuracy of our vector/matrix estimates: For a network with weights $\boldsymbol{A}$ and $\boldsymbol{B}$ and its teacher network with weights $\boldsymbol{A}^*$ and $\boldsymbol{B}^*$, *a)* layer 1 error refers to $\|\boldsymbol{A} - \boldsymbol{A}^*\| / \|\boldsymbol{A}^*\|$, similar to layer 2. *b)* output error refers to $\hat{\mathbb{E}} \left[ \|\hat{\mathbf{y}} - \mathbf{y}\| / \|\mathbf{y}\| \right]$ by test data. Due to the equivalence between the solution spaces of our QP and LP without label noise, we choose LP in noiseless experiments, referred to as "ours". In addition, to reduce variance, the results of learning the same ground-truths are computed as means across 16 trials.

---

[8]A folded Gaussian is the absolute value of a Gaussian, with p.d.f. $p(|\mathbf{x}|)$ where $\mathbf{x} \sim \mathcal{N}$, denoted as $|\mathcal{N}|$. We use folded Gaussian to ensure layer 1 weights are nonnegative.

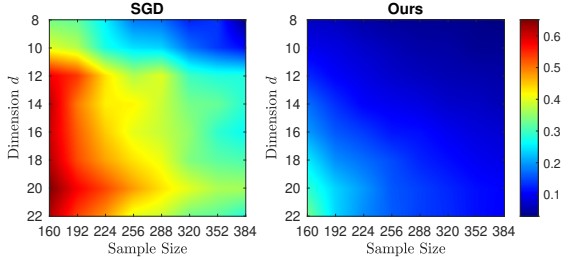

Figure 1: **Output errors by SGD and our method for different dimensions and sample sizes**. For the same $d$, we fix the ground-truth network as sample size grows.

Table 1: **Means / Standard deviations of network estimate errors for different network weights**. Values are computed from the process of learning 128 different ground-truth networks with $d = 16$. 512 training samples are drawn for each learning trial.

|  | Layer 1 | | Layer 2 | | Output | |
| --- | --- | --- | --- | --- | --- | --- |
|  | Mean | Std | Mean | Std | Mean | Std |
| SGD | 0.715 | 0.090 | 1.203 | 0.134 | 0.431 | 0.038 |
| Ours | 0.039 | 0.008 | $\approx 0$ | $\approx 0$ | 0.055 | 0.008 |

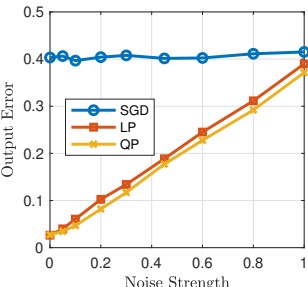 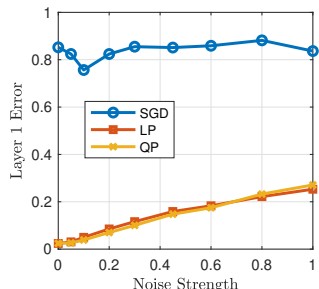 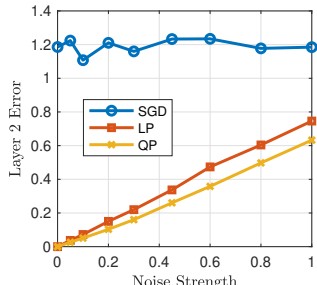

Figure 2: **Respective errors of layers 1 and 2 and outputs of different label noise strengths for SGD, LP and QP**. We fix the ground-truth weights with $d = 10$ and only the noise strength varies. 512 training samples are drawn for each learning trial.

**Sample Efficiency:** Consider Figure 1. It depicts the variation in the prediction errors (warmer color, larger error) as a function of $d$ (input dimension) and the number of samples the learning algorithm is using. We compare SGD against our algorithm. We observe that our approach to the estimating the neural network is more sample efficient. For SGD, the estimation is relatively easy with only up to 10 dimensions. As expected, once the dimension grows, the sample size required for the same level of error as our method is larger. Still, overall, our method is capable of learning robustly with small sample sizes and more efficiently than SGD even for larger sample sizes.

**Network Weight Robustness:** This experiment aims to verify whether our method can learn a broader class of residual units. In Table 1, we see our method shows a light-tailed distribution with nearly-zero means and standard deviations for layers 1 and 2, and output errors across various ground-truth networks, whereas SGD is less robust in the same context. Our method shows strong robustness to network weight changes, indicating its applicability across the whole hypothesis class.

**Noise Robustness:** Figure 2 confirms the robustness of our methods when output noise exists. Samples are generated by a ground-truth residual unit with output noise being i.i.d. zero-mean Gaussian in different strengths (i.e. standard deviations). We try both QP and LP because in noisy setting the two approaches are not equivalent w.r.t. the solution space.[9] First, SGD always gives larger errors than our methods, even though it is hardly affected by tuning the noise strength. For QP/LP, all the errors for layer 1, 2 and output grow almost linearly as noise strength increases, indicating that both QP/LP learn the optima robustly when output noise is present, where QP slightly outperforms LP.

---

[9]See Appendix A for noisy model discussion.

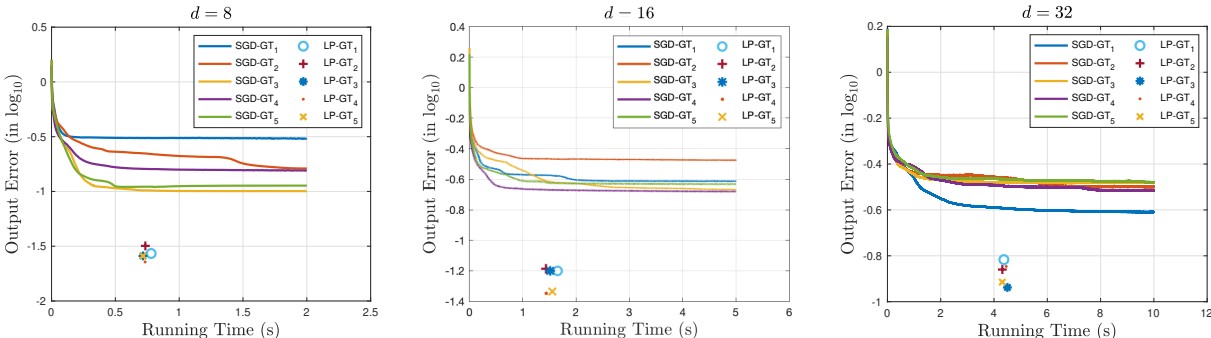

Figure 3: **Output errors against running time by SGD and our LP algorithm for different dimensions of networks**. For each dimension size of $d = 8$, $d = 16$ and $d = 32$, we report the learning curves of SGD and the final output error of our LP algorithm against time used on 5 different ground-truth networks. We used 512 training samples, drawn for learning each ground-truth network. SGD has different epochs, while the LP algorithm runs once. (CPU specification: 2.8 GHz Quad-Core Intel Core i7.)

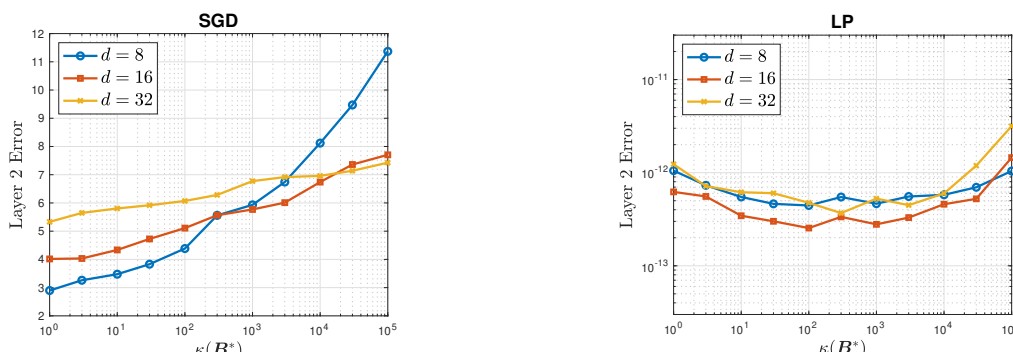

Figure 4: **Layer 2 errors against layer 2 ground-truth weight condition numbers for different dimensions of networks by SGD and our LP algorithm**. Each data point is the mean across learning 32 different ground-truth networks with the same pair of $d$ and a condition number for $\boldsymbol{B}^*$. The condition number is denoted by $\kappa(\boldsymbol{B}^*)$. We used 512 training samples, drawn for learning each ground-truth network.

**Running Time Efficiency versus SGD**   In Figure 3, we compare the running time of our algorithm against the running time of SGD for ground-truth networks sampled at random. The running time of our algorithm is significantly lower, with an error level which is also significantly lower. Furthermore, this is demonstrated across input dimensions, where the difference between the running time of SGD until convergence and our algorithm's running time increases as $d$ increases. In addition, the running time of our algorithm is fixed, and does not depend on the ground-truth network, while the running time of SGD varies based on the network sampled.

**Layer 2 Weights Condition Number Robustness**   Our algorithm depends on a matrix-inversion step to obtain the estimated parameters. This affects the estimation of $\boldsymbol{B}^*$. Due to this, we further investigate the effect of the condition number of $\boldsymbol{B}^*$ in the ground-truth parameters on the accuracy of estimation. For this experiment, to obtain ground-truth network parameters with different condition number, we follow the procedure of Ge et al. (2019), and multiply a diagonal matrix with exponentially-dropping values on the diagonal ($\lambda^{-i}$ for the $i$-th element) by two random orthonormal matrices, $\mathbf{U}$ and $\mathbf{V}^\top$, on the left and on the right respectively: $\mathbf{B} = \mathbf{U} \operatorname{diag}\left(\lambda^{-1}, \ldots, \lambda^{-d}\right) \mathbf{V}^\top$.

In Figure 4, we compare the effect of the condition number of $\boldsymbol{B}^*$ on SGD and on our algorithm. SGD turns out to be quite sensitive to the condition number, especially for lower dimensions. Our algorithm, on the

Table 2: **Results on regression benchmarks.** We give RMSE on several datasets with our algorithm (QP), backpropagation with SGD and backpropagation initialized with our algorithm estimates (SGD+QP). Numbers after slash denote the (average) number of epochs required for backpropagation to converge.

| Dataset | Root Mean Squared Error | | |
| --- | --- | --- | --- |
| | QP | SGD | SGD+QP |
| HOUSING | 19.46 | 18.08 / 459 | 12.43 / 1379 |
| DELTAELEVATORS | 0.00240 | 0.01360 / 8638 | 0.00209 / 843 |
| DELTAAILERONS | 0.00030 | 0.00216 / 16555 | 0.00030 / 3 |
| AILERONS | 0.00070 | 0.00193 / 8039 | 0.00023 / 1047 |
| REDWINE | 2.73 | 2.49 / 5270 | 1.48 / 5610 |
| WHITEWINE | 2.99 | 2.59 / 8200 | 1.61 / 3129 |
| JIGSAW | 1.17 | 2.45 / 6731 | 0.92 / 2413 |

other hand, is almost not affected by the condition number, with a final estimation error consistently close to 0, recovering the matrix $\boldsymbol{B}^*$.

## 7.2 Experiments with Regression Benchmark Datasets

In this set of experiments, we test our algorithm and compare it against stochastic gradient descent for four datasets: HOUSING (506 examples, 13 features), DELTAELEVATORS (9,517 examples, 6 features), DELTAAILERONS (7,129 examples, 5 features), AILERONS (7,154 examples, 41 features), REDWINE (1,599 examples, 11 features), WHITEWINE (4,898 examples, 11 features) and JIGSAW (47,991 examples, 100 features). The first six datasets are standard benchmark datasets taken from Delve[10] and the UCI Machine Learning Repository.[11] The last dataset, JIGSAW, is bigger with its goal to use word FastText[12] embeddings of tweets to predict their level of toxicity. For this set of experiments, with all datasets, we use the MOSEK solver with an academic license (ApS, 2019).

In all of our experiments, we report five-fold cross-validation results, where the first fold is used to tune the hyperparameters, and the last fold is used as both a validation set for early stopping (first half) and to report the results (second half). The models we use for backpropagation and our algorithm are identical, in the form of Eq. (1). We report (the fold-average) Root Mean Square Error (RMSE), defined as the square-root of the average of the squared deviations between the predicted value and the true value. Each fold is run with five different random seeds to initialize the backpropagation algorithm (results are averaged).

To satisfy the constraint on the length of $\mathbf{y}$, we duplicate the regression target multiple times, each time adding Gaussian noise as large as 0.1 of the standard deviation of the regression target. More specifically, let $\boldsymbol{y}^{(i)}$ be the $i$th example in a dataset. Then, we create a new $\boldsymbol{y}^{*,(i)} \in \mathbb{R}^d$ such that $\boldsymbol{y}_j^{*,(i)} = (1 + 0.1\zeta_{i,j})\boldsymbol{y}_{1+(j-1 \mod d)}^{(i)}$ for $j \geq 2$ (for $i = 1$, we add no noise, and just copy $\boldsymbol{y}^{(i)}$), where $\zeta_{i,j}$ are Gaussians with the mean being the standard deviation of $\boldsymbol{y}$ over the dataset.

With backpropagation, we use SGD with a learning rate of 0.000001 and batch size of 500. We run backpropagation until the mean squared error does not change between epochs within a fraction of 1/10000. We experiment with two types of initializations for the backpropagation algorithm: one in which we initialize all the weights randomly with a standard Gaussian distribution, and one in which we initialize it with the result of our quadratic programming algorithm. It has recently been shown that initializing a ResNet with positive values (or even just zero-one values) yields an improvement in estimating the network with backpropagation Zhao et al. (2022), supporting our use of the nonnegativity constraint for the first layer. An additional advantage of such an approach is that it removes the randomness that characterizes neural network learning.

---

[10] https://www.cs.toronto.edu/~delve/data/datasets.html
[11] https://archive.ics.uci.edu/ml/index.php
[12] https://fasttext.cc/

Table 2 gives the RMSE values, comparing our QP algorithm and the backpropagation algorithm. We note that our algorithm is especially potent when used to initialize the backpropagation algorithm: not only then it achieves lower RMSE on the test set, but it also does so in fewer iterations.

## 8 Conclusion

In this paper, we address the problem of learning a general class of two-layer residual units and propose an algorithm based on landscape design and convex optimization: We demonstrate firstly that minimizers of our objective functionals can express the *exact* ground-truth network. Then, we show that the corresponding ERM with nonparametric function estimation can be solved using *convex* QP/LP, which indicates *polynomial-time solvability* w.r.t. sample size and dimension. Moreover, our algorithms that are used to estimate both layers as well as the whole networks are *strongly consistent*, with very weak conditions on input distributions.

**Limitations and Future Work** The main limitation of our work is that the use of $L_2$ loss with the QP objective is not readily adaptable to other loss functions. Our preliminary experiments with classification datasets demonstrate that while our algorithm behaves better than $L_2$ backpropagation on such datasets, using the log-loss is more effective for these datasets. We leave it as future work to generalize our QP to a convex program with an arbitrary loss.

Our algorithm is not yet scalable for large datasets. We believe the learning algorithm could still be added to the ML toolkit for problems at a smaller scale for which we need a predictor, problems where linear regression is a good fit. Finally, we note that while we have nonnegativity constraints on the parameters ($A$), the class of networks we learn is expressive. Previous work that has similar nonnegativity constraints includes binary neural networks (Courbariaux et al., 2015) and others, without a 0/1 constraint (Chorowski & Zurada, 2014; Hosseini-Asl et al., 2015). We leave it for future work to alleviate this constraint. Recently, Zhao et al. (2022) showed that initializing ResNets with 0/1 weights is as effective for training as random initialization.

## Acknowledgments

We thank Yftah Ziser, the anonymous reviewers and the action editor for their useful feedback and comments. Computational resources and software were partially provided by Edinburgh Parallel Computing Centre and through the Amazon Enterprise License for `MATLAB`.

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

## Appendices

We include an outline of the appendices:

- Appendix A: A brief discussion of our learning algorithms in a noisy context, referring to the LP slack variable technique that is used in our experiments with noise in the main paper. Also, this section provides an insight to potential future direction of this work.

- Appendix B: A detailed explanation of the computational complexity and the methods through which the QPs/LPs in this paper are solved.

- Appendix C: A preliminarily proposed general optimization formula for the extension of learning multi-layer ResNets, to show the scalability of our algorithm.

- Appendix D: A formal result and an empirical validation of the example given in the main paper in the context of the vanilla linear regression method (Section 3).

- Appendix E: A generalization of layer 2 learning to the case where Condition 2.1 is not satisfied.

- Appendix F: Proofs regarding the minimizers of the objective functions we use.

- Appendix G: Proofs that justify the convexity of our QPs.

- Appendix H: Proofs that show our estimation algorithm is strongly consistent.

- Appendix I: A sample complexity justification of how our QP/LP approaches improves the exponential convergence rate that the vanilla LR approach (Section 3) has.

- Appendix J: Experiments with negative entries in $\mathbf{A}$ and a demonstration of the practicalities of the nonnegative entries assumption.

## A  Discussion: Noisy Model

Here, we discuss our learning methods in the noisy case. We introduce output noise to our model, namely

$$\mathbf{y} = \boldsymbol{B}^* \left[ (\boldsymbol{A}^*\mathbf{x})^+ + \mathbf{x} \right] + \mathbf{z},$$

where the label noise $\mathbf{z} \in \mathbb{R}^d$ is an i.i.d. random vector with respect to each component, satisfying $\mathbb{E}\left[\mathbf{z}\right] = 0$. In addition $\mathbf{z}$ and $\mathbf{x}$ are statistically independent.

Taking layer 2 as an example, our original objective functional does not reach zero by substituting the ground-truth: $G_2(\boldsymbol{C}^*, \boldsymbol{x} \mapsto (\boldsymbol{A}^*\boldsymbol{x})^+) = \mathbb{E}_{\mathbf{z}}[\|\boldsymbol{C}^*\mathbf{z}\|^2] = \sigma^2 \operatorname{Tr} \boldsymbol{C}^* \boldsymbol{C}^{*\top}$ where $\sigma$ is the noise strength. However, if layer 2 is well conditioned, the ground-truth will still assign a value close to zero to the objective. In this sense, with $G_2$'s continuity, the ground-truth can approximately minimize $G_2$, which validates our QP approach in terms of learning noisy residual units.

Our original LP (Eq. (5)) fails to give feasible solutions due to possible violation of the inequality $\boldsymbol{C}^*\boldsymbol{y} - \boldsymbol{x} \geq 0$, since $\boldsymbol{C}^*\boldsymbol{y} - \boldsymbol{x} = (\boldsymbol{A}^*\boldsymbol{x})^+ + \boldsymbol{C}^*\boldsymbol{z}$ is not necessarily an entrywise nonnegative vector because the term $\boldsymbol{C}^*\boldsymbol{z}$ might have negative entries. So we introduce slack variables $\boldsymbol{\zeta}^{(i)}$ to soften the constraints:

$$\min_{\boldsymbol{C},\boldsymbol{Z}} \; \frac{1}{n} \sum_{i \in [n]} \mathbf{1}^\top \cdot \boldsymbol{\zeta}^{(i)},$$

$$\text{s.t.} \;\; \boldsymbol{C}\boldsymbol{y}^{(i)} - \boldsymbol{x}^{(i)} \geq -\boldsymbol{\zeta}^{(i)}, \, \boldsymbol{\zeta}^{(i)} \geq \mathbf{0}.$$

For a sample $(\boldsymbol{x}^{(i)}, \boldsymbol{y}^{(i)})$ with noise $\boldsymbol{z}^{(i)}$, if $\left(\boldsymbol{A}^*\boldsymbol{x}^{(i)}\right)^+ + \boldsymbol{C}^*\boldsymbol{z}^{(i)} < 0$, then its $L^1$ norm would be added to the objective. This method remedies violations to the inequality $\boldsymbol{C}^*\mathbf{y} - \mathbf{x} \geq 0$. With access to a sufficiently large sample and with the "stability" assumption, our solution $\hat{\boldsymbol{C}}$ would be close to $\boldsymbol{C}^*$ since large deviations seldom rise and their penalties are diluted in the objective.

## B  Discussion: Solving Convex Programs

The theoretical foundation of solving convex QP and LP has been driven to maturity in terms of computational complexity (Kozlov et al., 1980; Murty, 1980) and convergence analysis (Jarre, 1990; Zhang et al., 1992). The time complexity for a convex QP/LP is analyzed in terms of the number of scalar variables $N$ and the number of bits $L$ in the input (Ye & Tse, 1989). For example, under Condition 2.1, $N = d^2 + nd$ for the QP because the matrix variables have $d^2$ scalars and the function estimators have $nd$ scalars. Similarly we have $N = d^2$ for the noiseless LP and $N = d^2 + nd$ for the noisy LP. It is guaranteed that primal-dual interior

---

**Algorithm 3** Learn a ReLU residual unit by LR.

1: **Input:** $\{(\boldsymbol{x}^{(i)}, \boldsymbol{y}^{(i)})\}_{i=1}^n$, $n$ samples drawn by Eq. (1).
2: **Output:** $\hat{\boldsymbol{A}}$, $\hat{\boldsymbol{B}}$, estimated weight matrices.
3: $\hat{\boldsymbol{B}} \leftarrow \mathrm{LR}\{(\boldsymbol{x}, \boldsymbol{y}) \in \{(\boldsymbol{x}^{(i)}, \boldsymbol{y}^{(i)})\}_{i=1}^{\lfloor n/2 \rfloor} \mid \boldsymbol{x} < 0\}$.
4: $\hat{\boldsymbol{D}} \leftarrow \mathrm{LR}\{(\boldsymbol{x}, \boldsymbol{y}) \in \{(\boldsymbol{x}^{(i)}, \boldsymbol{y}^{(i)})\}_{i=\lfloor n/2 \rfloor+1}^n \mid \boldsymbol{x} > 0\}$. {the estimation of $\boldsymbol{B}^* (\boldsymbol{A}^* + \boldsymbol{I}_d)$}
5: Solve full-rank linear equation system $\hat{\boldsymbol{B}} \cdot \tilde{\boldsymbol{A}} = \hat{\boldsymbol{D}}$
6: **return** $\tilde{\boldsymbol{A}} - \boldsymbol{I}_d$, $\hat{\boldsymbol{B}}$.

---

point methods can solve the convex QP/LP in a polynomial number of iterations $\mathcal{O}(\sqrt{N}L)$, where each iteration costs at worst $\mathcal{O}(N^{2.5})$ arithmetic operations from the Cholesky decomposition for the needed matrix inversion (Monteiro & Adler, 1989; Karmarkar, 1984). This indicates that our QPs/LPs are guaranteed to be solvable in $\mathcal{O}(N^3 L)$ arithmetic operations, which is generally an $\mathcal{O}(\mathrm{poly}(n, m, d, L))$ complexity.

To solve convex QPs/LPs in this paper, we mostly use `CVX`, a commonly used package for specifying and solving convex programs (Grant & Boyd, 2014; 2008). For both QPs and LPs, `CVX` calls a solver, `SDPT3` (Toh et al., 1999), which is specified for semidefinite-quadratic-linear programming and applies interior-point methods with the computational complexity mentioned above. Experimentally, `SDPT3` indeed specifies and solves our programs fast and makes our numerical results robust and stable.

## C   Discussion: Extension to Multi-Layer

In this appendix, we propose a generalized optimization formula of nonparametric learning of multi-layer residual networks to show that our algorithm is scalable. Assume we have an $L$-layer network of the form

$$\boldsymbol{r}_0 = \boldsymbol{x}$$
$$\boldsymbol{r}_l = \left(\boldsymbol{W}_{l-1}^* \boldsymbol{r}_{l-1}\right)^+ + \boldsymbol{r}_{l-1}, \, l = 1, \ldots, L-1$$
$$\boldsymbol{y} = \boldsymbol{W}_{L-1}^* \boldsymbol{r}_{L-1}.$$

We can scale the objective to use nonparameter variables $\{\boldsymbol{\xi}_l^{(i)}\}_{l \in [L-1]}$ to estimate every ReLU activation $(\boldsymbol{W}_l^* \boldsymbol{r}_l)^+$ in this network. Generally, for the $l$-th ReLU we can have the outer layer objective

$$\min_{\boldsymbol{\xi}_{[l]}^{(i)}, \boldsymbol{V}_l} \left\| \boldsymbol{\xi}_l^{(i)} + \cdots + \boldsymbol{\xi}_1^{(i)} + \boldsymbol{x}^{(i)} - \boldsymbol{V}_l \hat{\boldsymbol{h}}_l^{(i)} \right\|, \, \text{s.t. } \boldsymbol{\xi}_{[l]}^{(i)} \geq 0$$

where hat denotes estimated values and $\hat{\boldsymbol{h}}_{L-1}^{(i)} = \boldsymbol{y}^{(i)}$, and its corresponding inner layer objective

$$\min_{\boldsymbol{\phi}_l^{(i)}, \boldsymbol{W}_{l-1}, \boldsymbol{\xi}_l^{(i)}} \left\| \boldsymbol{\phi}_l^{(i)} + \boldsymbol{W}_{l-1} \left( \hat{\boldsymbol{\xi}}_{l-1}^{(i)} + \cdots + \hat{\boldsymbol{\xi}}_1^{(i)} + \boldsymbol{x}^{(i)} \right) - \boldsymbol{\xi}_l^{(i)} \right\|, \, \text{s.t. } \boldsymbol{\phi}_l^{(i)} \geq 0, \, \boldsymbol{\xi}_l^{(i)} \geq 0.$$

We can apply techniques, e.g., tune the combinations of those objectives with respect to layers or whether to have a hat or not, or even get them weighted, to keep its convex quadratic form and at the same time improve its optimization landscape. Specifically when $L = 2$, this becomes the two-layer ResNet learning discussed in the main paper.

## D   Vanilla Linear Regression: More Details

Algorithm 3 gives the full list of steps of learning two-layer residual units by LR, where we first split the drawn samples into two halves for the respective two key steps, then for both halves we filter negative and positive vectors, respectively. By running LR on both filtered sets we obtain an estimation of the ground-truth network.

In the following sections, we formalize this intuition and describe the exponential sample complexity of this approach under entry-wise i.i.d. setting as an example of the inefficiency of this method.

**Theorem D.1** (exponential sample complexity, vanilla LR)**.** *Assume the input vectors are i.i.d. with respect to each component. Then Algorithm 3 learns a neural network from a ground-truth residual unit with at least $\mathcal{O}\left(d \cdot 2^{d+1}\right)$ expected number of samples.*

*Proof.* For each $j \in [d]$, let $P_j$ be the marginal probability of $x_j$ being positive, i.e. $P_j = P(x_j > 0) > 0$. Thus, the probability that a sample is positive $P(\mathbf{x} > 0) = \prod_{j \in [d]} P_j$. Then the expected number of sampling trials to obtain $d$ positive samples is $d/\prod_{j \in [d]} P_j$. Similarly, the expected number of sampling trials to obtain $d$ negative samples is $d/\prod_{j \in [d]} [1 - P_j - P(\mathbf{x}_j = 0)]$. Let n be a random natural number s.t. with n samples the network has learned. The expected number of samples that guarantees successful learning is the sum of the two expectations

$$
\begin{aligned}
\mathbb{E}[n] &= d \left\{ \frac{1}{\prod_{j \in [d]} P_j} + \frac{1}{\prod_{j \in [d]} [1 - P_j - P(\mathbf{x}_j = 0)]} \right\} \\
&\geq d \left[ \frac{1}{\prod_{j \in [d]} P_j} + \frac{1}{\prod_{j \in [d]} (1 - P_j)} \right] \\
&\geq d \cdot 2 \prod_{j \in [d]} \frac{1}{\sqrt{P_j (1 - P_j)}} \\
&\geq d \cdot 2 \prod_{j \in [d]} \frac{1}{(P_j + 1 - P_j)/2} \\
&= d \cdot 2^{d+1}.
\end{aligned}
$$

Thus $\mathbb{E}[n] \geq \mathcal{O}(d \cdot 2^{d+1})$. $\square$

With an exponential sample complexity, the time complexity of the LR approach is thereby also exponential because filtering through all the samples costs time with the same complexity as the number of samples, even though the LR itself only costs polynomial time in the number of samples. To summarize, this vanilla LR approach learns exact ground-truth residual units but with exponential complexity in terms of both computational cost and sample size. While this approach to learning a residual unit such as ours is simple and intuitive, we aim to improve on this approach, making full use of all samples available.

**Experiments: Sample Efficiency**   We present the results of the vanilla LR in learning the residual units. As discussed in Section 3, LR requires full-rank linear systems parameterized by samples to learn the exact ground-truth parameters. However, with degenerate linear systems, LR is completely incapable of learning the parameters. Therefore, there exists a hard threshold for the number of samples required by LR that makes the linear equation system full-rank. With such a dichotomous constraint on LR learning, we take the learning success rate among 1000 trials as the metric to evaluate the performance of LR with different sample sizes and number of dimensions.

Table 3 shows the learning success rates with zero-mean Gaussian inputs. For each fixed number of dimensions, it appears there is a hard threshold that switches the learnability of LR. The exponential sample complexity is also reflected there as the number of dimensions grows linearly. Table 4 shows the rates with input mean non-zero, where an overall decline in success rates happens. This observation is explainable because the bottleneck of LR learning is the lower value found between the probability of sampling a positive and a negative vector. A positive mean reduces the probability of the latter, and thereby increases the sample size required.

# E   Generalization of Layer 2 Learning

In this appendix, we discuss how layer 2 is learned without satisfying Condition 2.1. We note that the fact that $\boldsymbol{A}^*$ is a scale transformation w.r.t. some components leads to scaling equivalence to our layer 2 objective functional minimizers just as in layer 1. To be more specific, we give a general version of Theorem 4.1

Table 3: **Learning success rate of vanilla LR on residual units with input x $\overset{\text{i.i.d.}}{\sim} \mathcal{N}(0,1)$**. The success rates shows an exponentially decreasing trend with the same number of samples. The sample sizes to achieve close to the same rate grows exponentially as the number of dimensions grows linearly.

| $d$ \ $n$ | 1e1 | 1e2 | 5e2 | 1e3 | 5e3 | 1e4 | 5e4 | 1e5 |
|---|---|---|---|---|---|---|---|---|
| 4 | 0.002 | 0.137 | 0.999 | 1 | 1 | 1 | 1 | 1 |
| 6 | 0 | 0 | 0.041 | 0.614 | 1 | 1 | 1 | 1 |
| 8 | 0 | 0 | 0 | 0 | 0.579 | 0.998 | 1 | 1 |
| 10 | 0 | 0 | 0 | 0 | 0 | 0.002 | 1 | 1 |
| 12 | 0 | 0 | 0 | 0 | 0 | 0 | 0.001 | 0.322 |

Table 4: **Learning success rate of vanilla LR on residual units with input x $\overset{\text{i.i.d.}}{\sim} \mathcal{N}(0.1,1)$**. With the non-zero Gaussian mean, the success rates show an overall decline compared with the rates shown in Table 3 for zero-mean Gaussian inputs.

| $d$ \ $n$ | 1e1 | 1e2 | 5e2 | 1e3 | 5e3 | 1e4 | 5e4 | 1e5 |
|---|---|---|---|---|---|---|---|---|
| 4 | 0.001 | 0.122 | 0.996 | 1 | 1 | 1 | 1 | 1 |
| 6 | 0 | 0 | 0.030 | 0.346 | 1 | 1 | 1 | 1 |
| 8 | 0 | 0 | 0 | 0 | 0.116 | 0.792 | 1 | 1 |
| 10 | 0 | 0 | 0 | 0 | 0 | 0 | 0.619 | 1 |
| 12 | 0 | 0 | 0 | 0 | 0 | 0 | 0 | 0.001 |

that handles the case without Condition 2.1 and describes the scaling equivalence of $G_2$ minimizers. See Theorem E.1 for a formal description.

**Theorem E.1** (objective minimizer space, layer 2, general). *Let $G_2$ be $\frac{1}{2}\mathbb{E}_{\mathbf{x}}\left[\left\|h\left(\mathbf{x}\right) + \mathbf{x} - \boldsymbol{C}\mathbf{y}\right\|^2\right]$ as a functional, where $\boldsymbol{C} \in \mathbb{R}^{d \times m}$, $h \in \mathbb{C}_{\geq 0}^0$. Then $G_2\left(\boldsymbol{C}, h\right)$ reaches its zero minimum iff $\boldsymbol{C}\boldsymbol{B}^* = \mathrm{diag}(\boldsymbol{k})$ where for each $j \in [d]$,*

*a)* $\frac{1}{1+A_{j,j}^*} \leq k_j \leq 1$, *if $\boldsymbol{A}^*$ is a scale transformation w.r.t. the $j$-th component.*
*b)* $k_j = 1$, *if $\boldsymbol{A}^*$ is not a scale transformation w.r.t. the $j$-th component.*

As in layer 1, the scaling equivalence in layer 2 can also be obtained by assigning $\boldsymbol{A}^*$ as a scale transformation w.r.t. some component and using the properties of the ReLU nonlinearity. See Appendix F.1 for a detailed explanation of Theorem E.1. To obtain the scale factors $\boldsymbol{k}$ and correct a scale-equivalent $G_2$ minimizer $\boldsymbol{C}$ to the ground-truth, we observe linear models parameterized by the scale factors, where for each $j \in [d]$, $[\boldsymbol{C}\mathbf{y}]_j = k_j\mathrm{x}_j$ given that $\mathrm{x}_j < 0$. See Theorem E.2 and its proof for justifications of the linear models that are used to compute $\boldsymbol{k}$.

**Theorem E.2** (scale factor, layer 2, general). *Assume $\boldsymbol{C}$ is a minimizer of $G_1$ in the context of Theorem E.1. Then for any $j \in [d]$, the following three propositions are equivalent:*

*a)* $\boldsymbol{A}^*$ *is a scale transformation w.r.t. the $j$-th component.*
*b)* $[\boldsymbol{C}\mathbf{y}]_j / \mathrm{x}_j$ *is a constant $c_j^{\mathrm{n}}$ given that $\mathrm{x}_j < 0$.*
*c)* $[\boldsymbol{C}\mathbf{y}]_j / \mathrm{x}_j$ *is a constant $c_j^{\mathrm{p}}$ given that $\mathrm{x}_j > 0$.*

*Additionally, if one of the above three propositions is true, then $k_j = c_j^{\mathrm{n}}$.*

---

**Algorithm 4** Rescale a $\hat{G}_2^{\mathrm{NPE}}$ minimizer.

---

1: **Parameters:** $\varepsilon_{\mathrm{tol}} > 0$, LR objective tolerance.
2: **Input:** $\{(\boldsymbol{x}^{(i)}, \boldsymbol{y}^{(i)})\}_{i=1}^n$, samples drawn by Eq. (1); $\hat{\boldsymbol{C}}$, a $\hat{G}_2^{\mathrm{NPE}}$ minimizer.
3: **Output:** $\hat{\boldsymbol{k}}$, a layer 2 scale factor estimate.
4: **for** each $j \in [d]$ **do**
5: $\quad \hat{k}_j \leftarrow \mathrm{LR}\left\{ x_j^{(i)}, \left[\hat{\boldsymbol{C}}\boldsymbol{y}^{(i)}\right]_j \right\}_{x_j^{(i)} < 0}$.
6: $\quad$ **if** the LR objective optimal $\hat{R}_j(\hat{k}_j) > \varepsilon_{\mathrm{tol}}$ **then**
7: $\quad\quad \hat{k}_j \leftarrow 1$.
8: $\quad$ **end if**
9: **end for**
10: **return** $\hat{\boldsymbol{k}}$.

---

*Proof.* For $j \in [d]$, we first prove a) $\implies$ b), c): From $\left(\boldsymbol{A}_{j,:}^*\right)^\top = A_{j,j}^* \cdot \boldsymbol{e}^{(j)}$ we have

$$
\frac{[\boldsymbol{C}\mathbf{y}]_j}{\mathrm{x}_j} = \frac{\left[k_j \boldsymbol{e}^{(j)}\right]^\top \left[(\boldsymbol{A}^*\mathbf{x})^+ + \mathbf{x}\right]}{\mathrm{x}_j}
$$

$$
= \frac{k_j \left[\left(\boldsymbol{A}_{j,:}^*\mathbf{x}\right)^+ + \mathrm{x}_j\right]}{\mathrm{x}_j} = \frac{k_j \left[\left(A_{j,j}^*\mathrm{x}_j\right)^+ + \mathrm{x}_j\right]}{\mathrm{x}_j}
$$

$$
= \begin{cases} k_j, & \mathrm{x}_j < 0 \\ k_j \left(A_{j,j}^* + 1\right), & \mathrm{x}_j > 0 \end{cases}.
$$

Then we prove $\neg$ a) $\implies$ $\neg$ b), $\neg$ c): $\neg$ a) $\implies \exists\, j' \in [d]$ and $j' \neq j$ s.t. $A_{j,j'}^* \neq 0$. Recall

$$
\frac{[\boldsymbol{C}\mathbf{y}]_j}{\mathrm{x}_j} = \frac{k_j \left[\left(\boldsymbol{A}_{j,:}^*\mathbf{x}\right)^+ + \mathrm{x}_j\right]}{\mathrm{x}_j}.
$$

The value of $\mathrm{x}_{j'}$ affects the value of $[\boldsymbol{C}\mathbf{y}]_j / \mathrm{x}_j$ because $A_{j,j'}^* \neq 0$. In fact, from $\mathrm{supp}\, p(\mathbf{x}) = \mathbb{R}^d$ we have $\mathrm{supp}\, p(\mathrm{x}_{j'} \mid \mathrm{x}_j) = \mathbb{R}$, indicating that the value of $[\boldsymbol{C}\mathbf{y}]_j / \mathrm{x}_j$ can never be kept as a constant when given both $\mathrm{x}_j < 0$ and $\mathrm{x}_j > 0$ because $\mathrm{x}_{j'}$ can be any real number. $\qquad\square$

With the exact derivations above, we are able to obtain a left inverse of $\boldsymbol{B}^*$, namely $\boldsymbol{C}$, that satisfies $\boldsymbol{C}\boldsymbol{B}^* = \boldsymbol{I}_d$. Consider Eq. (1) left multiplied by $\boldsymbol{B}^*\boldsymbol{C}$

$$
\boldsymbol{B}^*\boldsymbol{C}\mathbf{y} = \mathbf{y}. \tag{9}
$$

Eq. (9) is also a noiseless unbiased linear model where $\boldsymbol{C}\mathbf{y}$ and $\mathbf{y}$ are observable, and thereby $\boldsymbol{B}^*$ is computable due to the easy solvability of its linearity.

Algorithm 5 describes how a residual unit second layer is learned without satisfying Condition 2.1: We first solve the QP/LP and obtain a scaling equivalence to an estimated left inverse of $\boldsymbol{B}^*$ and the function estimate, namely $\hat{\boldsymbol{C}}$ and $\hat{\boldsymbol{\Xi}}$. Then we compute the scale factor estimate $\hat{\boldsymbol{k}}$ by running Algorithm 4, where for each component index $j \in [d]$, we first use a tolerance parameter as a threshold to determine whether ground-truth layer 1 is a scale transformation, and if so, we run LR to estimate the model $[\boldsymbol{C}\mathbf{y}]_j = k_j \mathrm{x}_j$, otherwise the scale factor is directly assigned by 1. Upon correcting $\hat{\boldsymbol{C}}$ and $\hat{\boldsymbol{\Xi}}$ by $\hat{\boldsymbol{k}}$, we obtain a layer 2 estimate $\hat{\boldsymbol{B}}$ by running LR to estimate the linear model Eq. (9). The strong consistency of our results in this appendix is justified in Appendix H.

## F Exact Derivation of Objective Functional Minimizers

We give the exact derivation of the objective functional minimizers in the next two subsections. We begin with layer 2 and continue to layer 1.

---

**Algorithm 5** Learn a ReLU residual unit layer 2.

---

1: **Input:** $\{(\boldsymbol{x}^{(i)}, \boldsymbol{y}^{(i)})\}_{i=1}^n$, samples drawn by Eq. (1).
2: **Output:** $\hat{\boldsymbol{B}}$, $\hat{\boldsymbol{\Xi}}$, a layer 2 and $\boldsymbol{x} \mapsto (\boldsymbol{A}^*\boldsymbol{x})^+$ estimate.
3: Go to line 4 if QP, line 5 if LP.
4: Solve QP: Eq. (4) and obtain a $\hat{G}_2^{\text{NPE}}$ minimizer, denoted by $\hat{\boldsymbol{C}}$, $\hat{\boldsymbol{\Xi}}$. Go to line 6.
5: Solve LP: Eq. (5) and obtain a minimizer $\hat{\boldsymbol{C}}$, then assign $\hat{\boldsymbol{\xi}}^{(i)} \leftarrow \hat{\boldsymbol{C}}\boldsymbol{y}^{(i)} - \boldsymbol{x}^{(i)}$ for each $i \in [n]$.
6: Run Algorithm 4 on $\{(\boldsymbol{x}^{(i)}, \boldsymbol{y}^{(i)})\}_{i \in [n]}$, $\hat{\boldsymbol{C}}$ and obtain $\hat{\boldsymbol{k}}$.
7: $\hat{\boldsymbol{B}} \leftarrow \text{LR}\left\{\text{diag}^{-1}(\hat{\boldsymbol{k}}) \cdot \hat{\boldsymbol{C}}\boldsymbol{y}^{(i)}, \boldsymbol{y}^{(i)}\right\}_{i \in [n]}$.
8: $\hat{\boldsymbol{\xi}}^{(i)} \leftarrow \text{diag}^{-1}(\boldsymbol{k})\left[\hat{\boldsymbol{\xi}}^{(i)} + \boldsymbol{x}^{(i)}\right] - \boldsymbol{x}^{(i)}$ for each $i \in [n]$. {Correct $\hat{\boldsymbol{\Xi}}$ to the function estimation of $(\boldsymbol{A}^*\boldsymbol{x})^+$.}
9: **return** $\hat{\boldsymbol{B}}$, $\hat{\boldsymbol{\Xi}}$.

---

### F.1 Layer 2

Lemma 4.2 is proved as follows.

*Proof.* " $\Longleftarrow$ ": Since $h(\boldsymbol{x}) = \boldsymbol{C}\boldsymbol{y} - \boldsymbol{x} \geq 0$, random vector $h(\mathbf{x}) + \mathbf{x} - \boldsymbol{C}\mathbf{y}$ is always a zero vector, which implies $G_2(\boldsymbol{C}, h) = 0$. Hence " $\Longleftarrow$ " holds.

" $\Longrightarrow$ ": Since r.v. $\|h(\mathbf{x}) + \mathbf{x} - \boldsymbol{C}\mathbf{y}\|^2 \geq 0$ we have

$$G_2(\boldsymbol{C}, h) = 0 \implies \lambda\left(\left\{\boldsymbol{x} \in \mathbb{R}^d \;\middle|\; \|h(\boldsymbol{x}) + \boldsymbol{x} - \boldsymbol{C}\boldsymbol{y}\|^2 > 0\right\}\right) = 0 \tag{10}$$

where $\lambda$ is the Lebesgue measure on $\mathbb{R}^d$.

Proof by contradiction: Assume that $\exists \boldsymbol{x}' \in \mathbb{R}^d$, $\exists i \in [d]$, s.t. $(\boldsymbol{C}\boldsymbol{y}' - \boldsymbol{x}')_i < 0$, where $\boldsymbol{y}'$ is the corresponding network output. Let $f(\boldsymbol{x}) = (\boldsymbol{C}\boldsymbol{y} - \boldsymbol{x})_i$ which is continuous on $\mathbb{R}^d$. Therefore for $\epsilon = -f(\boldsymbol{x}') > 0$, $\exists \delta > 0$, $\forall \boldsymbol{x} \in B(\boldsymbol{x}; \delta)$ i.e. $\|\boldsymbol{x} - \boldsymbol{x}'\| < \delta$,

$$|f(\boldsymbol{x}) - f(\boldsymbol{x}')| < \epsilon \implies 2f(\boldsymbol{x}') < f(\boldsymbol{x}) < 0 \implies [\boldsymbol{x} - \boldsymbol{C}\boldsymbol{y}]_i > 0$$
$$\implies [h(\boldsymbol{x}) + \boldsymbol{x} - \boldsymbol{C}\boldsymbol{y}]_i > 0 \implies \|h(\boldsymbol{x}) + \boldsymbol{x} - \boldsymbol{C}\boldsymbol{y}\|^2 > 0.$$

Since $\lambda(B(\boldsymbol{x}; \delta)) = \dfrac{\pi^{d/2}}{\Gamma(d/2 + 1)} \delta^d > 0$ and $B(\boldsymbol{x}; \delta)$ is a subset of the measured set in Eq. (10), we have a contradiction. Thus $\boldsymbol{C}\boldsymbol{y} - \boldsymbol{x} \geq 0$ holds in a pointwise manner in $\mathbb{R}^d$, indicating $\{\boldsymbol{x} \in \mathbb{R}^d \mid h(\boldsymbol{x}) \neq \boldsymbol{C}\boldsymbol{y} - \boldsymbol{x}\}$ must be a null set. With $h$'s continuity, $h$ must be $\boldsymbol{x} \mapsto \boldsymbol{C}\boldsymbol{y} - \boldsymbol{x}$. Therefore " $\Longrightarrow$ " holds. $\qquad\square$

**Lemma F.1.** $\boldsymbol{C}\boldsymbol{y} - \boldsymbol{x} \geq 0$ *holds for any* $\boldsymbol{x} \in \mathbb{R}^d$ *and its corresponding output* $\boldsymbol{y}$ *only if* $\boldsymbol{C}\boldsymbol{B}^*$ *is a diagonal matrix.*

*Proof.* Let $\boldsymbol{D} = \boldsymbol{C}\boldsymbol{B}^*$. We rewrite $\boldsymbol{C}\boldsymbol{y} - \boldsymbol{x} \geq 0$ as

$$\boldsymbol{D}\left[(\boldsymbol{A}^*\boldsymbol{x})^+ + \boldsymbol{x}\right] - \boldsymbol{x} \geq 0. \tag{11}$$

For further use, we substitute $\boldsymbol{x}$ by $-\boldsymbol{x}$ and the resulting inequality $\boldsymbol{D}\left[(-\boldsymbol{A}^*\boldsymbol{x})^+ - \boldsymbol{x}\right] + \boldsymbol{x} \geq 0$ still holds. Added by Eq. (11) we have

$$\boldsymbol{D}\left(\left|\boldsymbol{A}_{k,:}^* \cdot \boldsymbol{x}\right|\right)_{d \times 1} \geq 0. \tag{12}$$

Proof by contradiction: Assume $\boldsymbol{D}$ is not diagonal, then $\exists i \neq j$, such that $D_{i,j} \neq 0$. Consider the following two cases:

a) $D_{i,j} > 0$. We take the $i$-th row of Eq. (11) as follows

$$\sum_{k=1}^{d} D_{i,k} \left[ \left( \boldsymbol{A}_{k,:}^* \cdot \boldsymbol{x} \right)^+ + x_k \right] \geq x_i.$$

Let $x_{-j} = 0$ and $x_j < 0$. Then $\sum_{k=1}^{d} D_{i,k} \left[ \left( \boldsymbol{A}_{k,:}^* \cdot \boldsymbol{x} \right)^+ + x_k \right] = D_{i,j} \cdot x_j < 0 \implies \perp$.

b) $D_{i,j} < 0$. We take the $i$-th row of Eq. (12) as follows

$$\sum_{k=1}^{d} D_{i,k} \left| \boldsymbol{A}_{k,:}^* \cdot \boldsymbol{x} \right| \geq 0.$$

Let $\boldsymbol{x} = (\boldsymbol{A}^*)^{-1} \boldsymbol{v}$, where $\boldsymbol{v} = \boldsymbol{e}^{(j)}$. Then $\sum_{k=1}^{d} D_{i,k} \left| \boldsymbol{A}_{k,:}^* \cdot \boldsymbol{x} \right| = D_{i,j} < 0 \implies \perp$.

$\square$

With Lemma 4.2 and Lemma F.1, we prove Theorem E.1 as follows.

*Proof.* With Lemma 4.2, we only need to prove $\forall \boldsymbol{x} \in \mathbb{R}^d$, $\boldsymbol{C}\boldsymbol{y} - \boldsymbol{x} \geq 0 \iff \boldsymbol{C}\boldsymbol{B}^* = \mathrm{diag}(\boldsymbol{k})$.

" $\impliedby$ ": We have

$$\boldsymbol{C}\boldsymbol{y} - \boldsymbol{x} = \boldsymbol{C}\boldsymbol{B}^* \left[ (\boldsymbol{A}^*\boldsymbol{x})^+ + \boldsymbol{x} \right] - \boldsymbol{x} = \mathrm{diag}(\boldsymbol{k}) \left[ (\boldsymbol{A}^*\boldsymbol{x})^+ + \boldsymbol{x} \right] - \boldsymbol{x}. \tag{13}$$

For the $i$-th row of Eq. (13), consider the following two cases:

a) $\boldsymbol{A}_{i,:}^* \cdot \boldsymbol{x} = A_{i,i}x_i$, i.e. $\boldsymbol{A}^*$ is a scale transformation w.r.t. the $i$-th row. With $\frac{1}{1+A_{i,i}^*} \leq k_i \leq 1$ we have

$$[\boldsymbol{C}\boldsymbol{y} - \boldsymbol{x}]_i = k_i \left[ \left( A_{i,i}^* x_i \right)^+ + x_i \right] - x_i = k_i \left( A_{i,i}^* x_i \right)^+ + (k_i - 1) x_i.$$

For $x_i \geq 0$, $[\boldsymbol{C}\boldsymbol{y} - \boldsymbol{x}]_i = \left( k_i A_{i,i}^* + k_i - 1 \right) x_i \geq 0$.
For $x_i < 0$, $[\boldsymbol{C}\boldsymbol{y} - \boldsymbol{x}]_i = (k_i - 1) x_i \geq 0$.

b) $\boldsymbol{A}_{i,:}^* \cdot \boldsymbol{x} \neq A_{i,i}x_i$. With $k_i = 1$ we have

$$[\boldsymbol{C}\boldsymbol{y} - \boldsymbol{x}]_i = k_i \left[ \left( \boldsymbol{A}_{i,:}^*\boldsymbol{x} \right)^+ + x_i \right] - x_i = \left( \boldsymbol{A}_{i,:}^*\boldsymbol{x} \right)^+ \geq 0.$$

Hence " $\impliedby$ " holds.

" $\implies$ ": Let $\boldsymbol{D} = \boldsymbol{C}\boldsymbol{B}^*$. With Lemma F.1, $\boldsymbol{D}$ is diagonal. Consider the following two cases:

a) $\boldsymbol{A}_{i,:}^* \cdot \boldsymbol{x} = A_{i,i}x_i$. The $i$-th inequality can be written as

$$\begin{aligned}
[\boldsymbol{C}\boldsymbol{y} - \boldsymbol{x}]_i &= D_{i,i} \left[ \left( A_{i,i}^* x_i \right)^+ + x_i \right] - x_i \\
&= D_{i,i} \left( A_{i,i}^* x_i \right)^+ + (D_{i,i} - 1) x_i \geq 0.
\end{aligned} \tag{14}$$

Proof by contradiction: we need to find $\boldsymbol{x} \in \mathbb{R}^d$ which contradicts with Eq. (14) in the following three cases:

a) If $D_{i,i} \leq 0$, then, $x_i > 0 \implies D_{i,i} \left( A_{i,i}^* x_i \right)^+ + (D_{i,i} - 1) x_i < 0 \implies \perp$.

b) If $D_{i,i} > 1$, then $x_i < 0 \implies \perp$.

c) If $0 < D_{i,i} < \frac{1}{1+A_{i,i}^*}$, then $\exists a > 0$, s.t. $D_{i,i} = \frac{1}{1+A_{i,i}^*+a}$. Letting $x_i > 0$, we have

$$D_{i,i} \left( A_{i,i}^* x_i \right)^+ + (D_{i,i} - 1) x_i = \frac{\left( A_{i,i}^* x_i \right)^+}{1 + A_{i,i}^* + a} - \frac{\left( A_{i,i}^* + a \right) x_i}{1 + A_{i,i}^* + a} < 0 \implies \perp.$$

Hence $\frac{1}{1+A_{i,i}^*} \le D_{i,i} \le 1$.

b) $\boldsymbol{A}_{i,:}^* \cdot \boldsymbol{x} \ne A_{i,i}x_i$, i.e. $\exists\, j \ne i$, s.t. $\boldsymbol{A}_{i,j}^* > 0$. The $i$-th inequality can be written as

$$
\begin{aligned}
[\boldsymbol{Cy} - \boldsymbol{x}]_i &= D_{i,i}\left[\left(\boldsymbol{A}_{i,:}^*\boldsymbol{x}\right)^+ + x_i\right] - x_i \\
&= D_{i,i}\left(\boldsymbol{A}_{i,:}^*\boldsymbol{x}\right)^+ + (D_{i,i} - 1)\,x_i \ge 0.
\end{aligned}
\tag{15}
$$

Proof by contradiction: we need to find $\boldsymbol{x} \in \mathbb{R}^d$ which contradicts with Eq. (15) in the following three cases:

a) If $D_{i,i} \le 0$, then, $x_i > 0 \implies D_{i,i}\left(\boldsymbol{A}_{i,:}^*\boldsymbol{x}\right)^+ + (D_{i,i} - 1)\,x_i < 0 \implies \perp$.

b) If $D_{i,i} > 1$, then, $x_i < 0 \wedge x_{-i} \le 0 \implies \perp$.

c) If $0 < D_{i,i} < 1$, then, $x_i > 0 \wedge x_j \le -\frac{A_{i,i}^*}{A_{i,j}^*}x_i \wedge x_k \le 0 \implies \perp$, where $k \ne i, j$.

Hence $D_{i,i} = 1$.

Hence $\boldsymbol{D} = \boldsymbol{CB}^* = \mathrm{diag}(\boldsymbol{k})$, and thereby " $\implies$ " holds. □

## F.2 Layer 1

The proof of Lemma 5.3 is similar to that of Lemma 4.2 because the two lemmas follow the same idea, which is to link objective functional minimization with always-hold inequalities. With Lemma 5.3, we prove Theorem 5.1 as follows.

*Proof.* With Lemma 5.3, we only need to prove $\forall\, \boldsymbol{x} \in \mathbb{R}^d$, $(\boldsymbol{A}^*\boldsymbol{x})^+ - \boldsymbol{Ax} \ge 0 \iff \forall\, i \in [d], \boldsymbol{A}_{i,:} = k_i\boldsymbol{A}_{i,:}^*$, where $0 \le k_i \le 1$. This is equivalent to what it is for a single row, i.e. $\forall\, \boldsymbol{x} \in \mathbb{R}^d$, $\left(\boldsymbol{a}^{*\top}\boldsymbol{x}\right)^+ - \boldsymbol{a}^\top\boldsymbol{x} \ge 0 \iff \boldsymbol{a} = k\boldsymbol{a}^*$, where $0 \le k \le 1$.

" $\impliedby$ ": The case where $\boldsymbol{a}^* = 0$ is clear. If $\boldsymbol{a}^*$ is not a zero vector and $\boldsymbol{a} = k\boldsymbol{a}^*$, we have

a) If $\boldsymbol{a}^{*\top}\boldsymbol{x} \ge 0$, $\left(\boldsymbol{a}^{*\top}\boldsymbol{x}\right)^+ - \boldsymbol{a}^\top\boldsymbol{x} = \boldsymbol{a}^{*\top}\boldsymbol{x} - k\boldsymbol{a}^{*\top}\boldsymbol{x} = (1-k)\,\boldsymbol{a}^{*\top}\boldsymbol{x} \ge 0$.

b) If $\boldsymbol{a}^{*\top}\boldsymbol{x} < 0$, $\left(\boldsymbol{a}^{*\top}\boldsymbol{x}\right)^+ - \boldsymbol{a}^\top\boldsymbol{x} = -k\boldsymbol{a}^{*\top}\boldsymbol{x} \ge 0$.

Hence " $\impliedby$ " holds.

" $\implies$ ": If $\boldsymbol{a}^* = 0$, $\boldsymbol{a}$ must be a zero vector, otherwise let $\boldsymbol{x} = \boldsymbol{a}$, then $-\boldsymbol{a}^\top\boldsymbol{x} < 0 \implies \perp$. If $\boldsymbol{a}^*$ is not a zero vector, consider two cases below:

a) If $\boldsymbol{a} \ne k\boldsymbol{a}^*$ where $k \ge 0$. Let $\boldsymbol{x} = \frac{\|\boldsymbol{a}^*\|}{\|\boldsymbol{a}\|} \cdot \boldsymbol{a} - \boldsymbol{a}^*$, then $\boldsymbol{x}$ is not a zero vector, and

$$
\left.\begin{aligned}
\boldsymbol{a}^{*\top}\boldsymbol{x} &= \|\boldsymbol{a}^*\|^2 (\cos\theta - 1) < 0\,, \\
\boldsymbol{a}^\top\boldsymbol{x} &= \|\boldsymbol{a}\|\,\|\boldsymbol{a}^*\| (1 - \cos\theta) > 0
\end{aligned}\right\} \implies \left(\boldsymbol{a}^{*\top}\boldsymbol{x}\right)^+ - \boldsymbol{a}^\top\boldsymbol{x} < 0 \implies \perp
$$

where $\theta$ denotes the angle between $\boldsymbol{a}$ and $\boldsymbol{a}^*$.

b) If $\boldsymbol{a} = k\boldsymbol{a}^*$ where $k > 1$, then let $\boldsymbol{x} = \boldsymbol{a}^*$ we have

$$
\left(\boldsymbol{a}^{*\top}\boldsymbol{x}\right)^+ - \boldsymbol{a}^\top\boldsymbol{x} = (1 - k)\,\|\boldsymbol{a}^*\|^2 < 0 \implies \perp.
$$

Hence $\boldsymbol{a} = k\boldsymbol{a}^*$ where $0 \le k \le 1$ if $\boldsymbol{a}^*$ is not zero, and thereby " $\implies$ " holds. □

## G QP Convexity

The LPs in the main paper are trivially convex. So in this appendix, we only justify the convexity of our QPs: We first prove the convexity of single-sample objectives, then the convexity of the empirical objectives with nonparametric estimation, i.e. $\hat{G}_1^{\mathrm{NPE}}$ and $\hat{G}_2^{\mathrm{NPE}}$ is obtained by the convexity of convex function summations.

**Lemma G.1.** *Suppose $f(\boldsymbol{u}) = \frac{1}{2}\|\boldsymbol{Tu} - \boldsymbol{b}\|^2$ where $\boldsymbol{u}$ is a real matrix. Then $f$ is convex w.r.t. $\boldsymbol{u}$.*

Lemma G.1 is easily obtained since the Hessian $f''(\boldsymbol{T}) = \boldsymbol{T}^\top \boldsymbol{T}$ is positive semidefinite. In the following, we demonstrate and justify the convexity of the QPs of both layers by rewriting their single-sample objectives into the formulation of $f$ and summing them without loss of convexity.

**Theorem G.2.** *QP: Eq. (4), and QP: Eq. (7) are convex optimization problems.*

*Proof.* First of all, constraints of both QPs are trivially linear and convex. Thus, we only need to justify the convexity of the two empirical objectives, $\hat{G}_1^{\mathrm{NPE}}$ and $\hat{G}_2^{\mathrm{NPE}}$ (see Eqs. (4) and (7)). Consider the single-sample version of $\hat{G}_1^{\mathrm{NPE}}$, namely $g_1^{\mathrm{NPE}}(\boldsymbol{A}, \boldsymbol{\phi}; \boldsymbol{x}, \boldsymbol{h}) = \frac{1}{2} \|\boldsymbol{\phi} + \boldsymbol{A}\boldsymbol{x} - \boldsymbol{h}\|^2$, which, in the formulation of $f$, can be rewritten with

$$
\boldsymbol{T} = \begin{bmatrix} \boldsymbol{x}^\top & & & & 1 & & & \\ & \boldsymbol{x}^\top & & & & 1 & & \\ & & \ddots & & & & \ddots & \\ & & & \boldsymbol{x}^\top & & & & 1 \end{bmatrix}, \ \boldsymbol{u} = \begin{bmatrix} \boldsymbol{A}_{1,:}^\top \\ \boldsymbol{A}_{2,:}^\top \\ \vdots \\ \boldsymbol{A}_{d,:}^\top \\ \boldsymbol{\phi} \end{bmatrix}, \ \boldsymbol{b} = \boldsymbol{h}
$$

which guarantees the convexity of $g_1^{\mathrm{NPE}}$ w.r.t. $\boldsymbol{A}$ and $\boldsymbol{\phi}$ by Lemma G.1. For $g_2^{\mathrm{NPE}}(\boldsymbol{C}, \boldsymbol{\xi}; \boldsymbol{x}, \boldsymbol{y}) = \frac{1}{2} \|\boldsymbol{\xi} + \boldsymbol{x} - \boldsymbol{C}\boldsymbol{y}\|^2$, we have

$$
\boldsymbol{T} = \begin{bmatrix} -\boldsymbol{y}^\top & & & & 1 & & & \\ & -\boldsymbol{y}^\top & & & & 1 & & \\ & & \ddots & & & & \ddots & \\ & & & -\boldsymbol{y}^\top & & & & 1 \end{bmatrix}, \ \boldsymbol{u} = \begin{bmatrix} \boldsymbol{C}_{1,:}^\top \\ \boldsymbol{C}_{2,:}^\top \\ \vdots \\ \boldsymbol{C}_{m,:}^\top \\ \boldsymbol{\xi} \end{bmatrix}, \ \boldsymbol{b} = -\boldsymbol{x}
$$

which guarantees the convexity of $g_2^{\mathrm{NPE}}$ w.r.t. $\boldsymbol{C}$ and $\boldsymbol{\xi}$ by Lemma G.1. Now we consider the summation. Taking layer 1 as an example, by definition we have

$$
\hat{G}_1^{\mathrm{NPE}}(\boldsymbol{A}, \boldsymbol{\Phi}) = \sum_{i \in [n]} g_1^{\mathrm{NPE}}(\boldsymbol{A}, \boldsymbol{\phi}^{(i)}; \boldsymbol{x}^{(i)}, \boldsymbol{h}^{(i)}).
$$

For each $i \in [n]$, equivalently, we take $\boldsymbol{\Phi}$ as a variable instead of $\boldsymbol{\phi}^{(i)}$ in $g_1^{\mathrm{NPE}}$, but with only $\boldsymbol{\phi}^{(i)} \in \boldsymbol{\Phi}$ determining the value of $g_1^{\mathrm{NPE}}$. In this sense, $g_1^{\mathrm{NPE}}$ is convex w.r.t. $\boldsymbol{A}$ and $\boldsymbol{\Phi}$ for each $i \in [n]$. Thus, the sum $\hat{G}_1^{\mathrm{NPE}}$ is convex w.r.t. $\boldsymbol{A}$ and $\boldsymbol{\Phi}$. Similarly, $\hat{G}_2^{\mathrm{NPE}}$ is convex w.r.t. $\boldsymbol{C}$ and $\boldsymbol{\Xi}$. $\qquad\square$

# H  Strong Consistency

In this appendix, we justify the strong consistency of our estimators for the residual unit layer 1/2 learning and the whole network.

According to Theorem E.1 and Theorem 5.1, the solutions to our objective functionals are continuous sets. In addition, we also present intermediate results in terms of continuous sets, e.g. possible left-inverse matrices for $\boldsymbol{B}^*$ in the results for layer 2. Thus, to analyze the consistency of our learning algorithm, we define distances between sets, so that the convergence of sets can be well defined. Further point convergence results, i.e. layer 1/2 estimator strong consistency, are based on the set convergence we define.

**Definition H.1** (deviation). Let $\mathbb{A} \subseteq \mathbb{M}$ and $\mathbb{B} \subseteq \mathbb{M}$ be two non-empty sets from a metric space $(\mathbb{M}, d)$. The deviation of set $\mathbb{A}$ from the set $\mathbb{B}$, denoted by $D(\mathbb{A}, \mathbb{B})$, is

$$
D(\mathbb{A}, \mathbb{B}) = \sup_{\boldsymbol{a} \in \mathbb{A}} d(\boldsymbol{a}, \mathbb{B}) = \sup_{\boldsymbol{a} \in \mathbb{A}} \inf_{\boldsymbol{b} \in \mathbb{B}} d(\boldsymbol{a}, \boldsymbol{b}),
$$

where sup and inf represent supremum and infimum, respectively.

**Definition H.2** (Hausdorff distance). Let $\mathbb{A} \subseteq \mathbb{M}$ and $\mathbb{B} \subseteq \mathbb{M}$ be two non-empty sets from a metric space $(\mathbb{M}, d)$. The Hausdorff distance between $\mathbb{A}$ and $\mathbb{B}$, denoted by $D_{\mathrm{H}}(\mathbb{A}, \mathbb{B})$, is

$$
D_{\mathrm{H}}(\mathbb{A}, \mathbb{B}) = \max\{D(\mathbb{A}, \mathbb{B}), D(\mathbb{B}, \mathbb{A})\}.
$$

*Remark.* In the following, we use the Frobenius norm to define the distance between matrices, i.e. $d(\boldsymbol{X}, \boldsymbol{Y}) = \|\boldsymbol{X} - \boldsymbol{Y}\|_{\mathrm{F}}$.

### H.1 Layer 2

In this subsection, we prove the strong consistency of the layer 2 estimator.

#### H.1.1 Objective Minimizer Space Estimator

For simplicity of notation, we use $\hat{\mathbb{S}}_n$ to denote our layer 2 QP/LP solution space by $n$ random samples[13] as a random set

$$\hat{\mathbb{S}}_n := \{\boldsymbol{C} \in \mathbb{R}^{d \times m} : \boldsymbol{C}\mathbf{y}^{(i)} - \mathbf{x}^{(i)} \geq 0, \, \forall\, i \in [n]\} \tag{16}$$

and $\mathbb{S}^*$ to denote the value space of $\boldsymbol{C}$ in Lemma 4.2 which minimizes layer 2 objective functional

$$\mathbb{S}^* := \{\boldsymbol{C} \in \mathbb{R}^{d \times m} : \boldsymbol{C}\boldsymbol{y} - \boldsymbol{x} \geq 0, \, \forall\, \boldsymbol{x} \in \mathbb{R}^d \text{ and its corresponding output } \boldsymbol{y}\}.$$

For further use, we name $\hat{\mathbb{P}}_n$ as the set of $n$ sampled inputs which define $\hat{\mathbb{S}}_n$, i.e. $\hat{\mathbb{P}}_n := \{\mathbf{x}^{(i)}\}_{i \in [n]}$ where each $\mathbf{x}^{(i)}$ is the same random variable in Eq. (16). By the definitions above, we describe the strong consistency of our QP/LP as Lemma H.1.

**Lemma H.1** (QP/LP strong consistency, layer 2). $D_{\mathrm{H}}(\hat{\mathbb{S}}_n, \mathbb{S}^*) \xrightarrow{a.s.} 0 \text{ as } n \to \infty.$

*Proof.* First we prove that $D_{\mathrm{H}}(\hat{\mathbb{S}}_n, \mathbb{S}^*) \xrightarrow{\mathrm{P}} 0$ as $n \to \infty$. Recall Theorem E.1, $\boldsymbol{C}\mathbf{y} - \mathbf{x} \geq 0$ only if $\boldsymbol{C}\boldsymbol{B}^* = \mathrm{diag}(\boldsymbol{k})$. We inherit the notation as defining $\boldsymbol{D} = \boldsymbol{C}\boldsymbol{B}^*$. Theorem E.1 is based on Lemma F.1, and we prove them by raising points that show contradiction, i.e. violate the inequality that holds in a pointwise fashion:

a) In the proof of Lemma F.1, we use $d$ points: $-\boldsymbol{e}^{(i)}$, for $i \in [d]$, to make $\sum_{k=1}^{d} D_{i,k}\left[\left(\boldsymbol{A}_{k,:}^* \cdot \boldsymbol{x}\right)^+ + x_k\right] < x_i$, so that $D_{i,j}$ $(i \neq j)$ cannot be positive; and another $d$ points: $(\boldsymbol{A}^*)^{-1}\boldsymbol{e}^{(i)}$, to show $\sum_{k=1}^{d} D_{i,k}\left|\boldsymbol{A}_{k,:}^* \cdot \boldsymbol{x}\right| < 0$, so that $D_{i,j}$ $(i \neq j)$ cannot be negative. For each $-\boldsymbol{e}^{(i)}$ we use here, since the violations follow strict inequalities, we know there exists a neighborhood of $-\boldsymbol{e}^{(i)}$, $\mathbb{N}_i = \mathbb{N}(-\boldsymbol{e}^{(i)})$, such that $\forall \boldsymbol{z} \in \mathbb{N}_i$, $\sum_{k=1}^{d} D_{i,k}\left[\left(\boldsymbol{A}_{k,:}^* \cdot \boldsymbol{z}\right)^+ + z_k\right] < z_i$. We can similarly find such neighborhood of each $(\boldsymbol{A}^*)^{-1}\boldsymbol{e}^{(i)}$ that the strict inequality holds within the neighborhood respectively. We index them as $\mathbb{N}_{d+1}$ to $\mathbb{N}_{2d}$.

b) In the proof of Theorem E.1, we further construct $d$ points: for each $i \in [d]$, we take a point $x$ such that $x_i > 0 \wedge x_j \leq -\frac{A_{i,i}^*}{A_{i,j}^*}x_i \wedge x_k \leq 0$, where $k \neq i, j$. This counterexample shows $[\boldsymbol{C}\boldsymbol{y} - \boldsymbol{x}]_i < 0$, and eliminates the possibility of $0 < D_{i,i} < 1$ when $\boldsymbol{A}_{i,:}^*$ is not a scale transformation. We can similarly find neighborhood of each point and index them as $\mathbb{N}_{2d+1}$ to $\mathbb{N}_{3d}$. Note that we omit some cases in the proof of Theorem E.1, because the first $2d$ points are sufficient to use in those cases to show contradiction.

In the sampling procedure, if we sample at least one point in each neighborhood $\mathbb{N}_i$, Theorem E.1 assures the solution we get $\hat{\mathbf{C}}_n$ would lie in the true optimal set $\mathbb{S}^*$. The probability that the sampling procedure "omits" any of the neighborhoods is

$$P\left(\hat{\mathbb{P}}_n \bigcap \mathbb{N}_1 = \emptyset \text{ or } \hat{\mathbb{P}}_n \bigcap \mathbb{N}_2 = \emptyset \text{ or } \ldots \text{ or } \hat{\mathbb{P}}_n \bigcap \mathbb{N}_{3d} = \emptyset\right) \leq \sum_{i=1}^{3d} P\left(\hat{\mathbb{P}}_n \bigcap \mathbb{N}_i = \emptyset\right)$$
$$\leq 3d[1 - \min_{i \in [3d]} P(\mathbb{N}_i)]^n \tag{17}$$

Since the measure on each neighborhood $P(\mathbb{N}_i) = \int_{\boldsymbol{x} \in \mathbb{N}_i} p(\boldsymbol{x}) > 0$,

$$P\left(\hat{\mathbb{P}}_n \bigcap \mathbb{N}_1 = \emptyset \text{ or } \hat{\mathbb{P}}_n \bigcap \mathbb{N}_2 = \emptyset \text{ or } \ldots \text{ or } \hat{\mathbb{P}}_n \bigcap \mathbb{N}_{3d} = \emptyset\right) \to 0, \text{ as } n \to \infty.$$

---

[13]Here, we take samples as random variables for empirical analysis.

Here we obtain $D_{\mathrm{H}}(\hat{\mathbb{S}}_n, \mathbb{S}^*) \xrightarrow{\mathrm{P}} 0$ as $n \to \infty$. Now we take the infinite sum over the both sides of Eq. (17)

$$\sum_{n \in [\infty]} P\left(\hat{\mathbb{P}}_n \bigcap \mathbb{N}_1 = \emptyset \text{ or } \hat{\mathbb{P}}_n \bigcap \mathbb{N}_2 = \emptyset \text{ or } \ldots \text{ or } \hat{\mathbb{P}}_n \bigcap \mathbb{N}_{3d} = \emptyset\right)$$

$$\leq 3d \sum_{n \in [\infty]} \left[1 - \min_{i \in [3d]} P(\mathbb{N}_i)\right]^n = 3d \left[\frac{1}{\min_{i \in [3d]} P(\mathbb{N}_i)} - 1\right] < +\infty.$$

By the Borel-Cantelli lemma (Borel, 1909), $D_{\mathrm{H}}(\hat{\mathbb{S}}_n, \mathbb{S}^*) \xrightarrow{\text{a.s.}} 0$ as $n \to \infty$. $\qquad \square$

### H.1.2   Scale Factor Estimator

To avoid ambiguity, we use $n_{\mathrm{sf}}$ to denote the number of samples used in Algorithm 4. The samples pairs are $\{(\mathbf{x}^{(i)}, \mathbf{y}^{(i)})\}_{i=1}^{n_{\mathrm{sf}}}$. Without loss of generality, the following discussion focuses on some fixed index $j \in [d]$. In Algorithm 4, we plug in our estimator $\hat{\mathbf{C}}_n$ and use LR to estimate $k_j$ given that $\mathrm{x}_j^{(i)} < 0$

$$\mathrm{k}_{n_{\mathrm{sf}}}(\hat{\mathbf{C}}_n) = \arg\min_k \frac{1}{2n_{\mathrm{sf}}} \sum_{i \in [n_{\mathrm{sf}}]} \left\|[\hat{\mathbf{C}}_n \mathbf{y}^{(i)}]_j - k\mathrm{x}_j^{(i)}\right\|^2 = \frac{\sum_{i \in [n_{\mathrm{sf}}]} \mathrm{x}_j^{(i)} [\hat{\mathbf{C}}_n \mathbf{y}^{(i)}]_j}{\sum_{i \in [n_{\mathrm{sf}}]} \left(\mathrm{x}_j^{(i)}\right)^2}. \tag{18}$$

We first give the strong consistency of layer 2 scale factor estimator for as $n_{\mathrm{sf}} \to \infty$, as described in Lemma H.2.

**Lemma H.2** (scale factor estimator strong consistency, layer 2). *Suppose $\mathbf{A}^*$ is a scale transformation w.r.t. the $j$-th component, and $\mathrm{k}_{n_{sf}}(\hat{\mathbf{C}}_n)$ is the $n_{sf}$-sample estimator of $k_j$ via LR: Eq. (18) given that $\mathbf{x}_j^{(i)} < 0$. Define sets*

$$\mathbb{U}_{n_{sf},n} := \{\mathrm{k}_{n_{sf}}(\mathbf{C}) : \mathbf{C} \in \hat{\mathbb{S}}_n\}, \text{ and } \mathbb{U}^* := \left[\frac{1}{1 + A_{j,j}^*}, 1\right].$$

*Then $\lim_{n_{sf} \to \infty} \lim_{n \to \infty} D_{\mathrm{H}}(\mathbb{U}_{n_{sf},n}, \mathbb{U}^*) \overset{\text{a.s.}}{=} 0$.*

*Proof.* Following our notation, Theorem E.1 and Theorem E.2 ensure that if $\mathbf{A}^*$ is a scale transformation w.r.t. the $j$-th component, for any $\mathbf{C}$ belonging to the true optimal set $\mathbb{S}^*$, $\mathrm{k}_{n_{\mathrm{sf}}}(\mathbf{C}) \in \left[\frac{1}{1+A_{j,j}^*}, 1\right]$. And the "iff" statement strengthens that $\mathbb{U}^* = \{\mathrm{k}_{n_{\mathrm{sf}}}(\mathbf{C}) : \mathbf{C} \in \mathbb{S}^*\}$ for any $n_{\mathrm{sf}} \in \mathbb{Z}^+$. Note that since $\mathbb{S}^* \subset \hat{\mathbb{S}}_n$, we have $\mathbb{U}^* \subset \mathbb{U}_{n_{\mathrm{sf}},n}$. We only need to prove $D(\mathbb{U}_{n_{\mathrm{sf}},n}, \mathbb{U}^*) \xrightarrow{\text{a.s.}} 0$ as $n \to \infty$.

$\forall \hat{\mathbf{C}}_n \in \hat{\mathbb{S}}_n$ and $\forall \mathbf{C} \in \mathbb{S}^*$,

$$\left|\mathrm{k}_{n_{\mathrm{sf}}}(\hat{\mathbf{C}}_n) - \mathrm{k}_{n_{\mathrm{sf}}}(\mathbf{C})\right|$$

$$= \frac{1}{\sum_{i \in [n_{\mathrm{sf}}]} \left(\mathrm{x}_j^{(i)}\right)^2} \left|\sum_{i \in [n_{\mathrm{sf}}]} \mathrm{x}_j^{(i)} \left[(e^{(j)})^\top \left(\hat{\mathbf{C}}_n - \mathbf{C}\right) \mathbf{B}^* \left[\left(\mathbf{A}^* \mathbf{x}^{(i)}\right)^+ + \mathbf{x}^{(i)}\right]\right]\right|$$

$$\leq \frac{1}{\sum_{i \in [n_{\mathrm{sf}}]} \left(\mathrm{x}_j^{(i)}\right)^2} \left[\sum_{i \in [n_{\mathrm{sf}}]} \left|\mathrm{x}_j^{(i)}\right| \left\|(e^{(j)})^\top \left(\hat{\mathbf{C}}_n - \mathbf{C}\right) \mathbf{B}^*\right\|_2 \left(\left\|\mathbf{A}^* \mathbf{x}^{(i)}\right\|_2 + \left\|\mathbf{x}^{(i)}\right\|_2\right)\right]$$

$$\leq \frac{1}{\sum_{i \in [n_{\mathrm{sf}}]} \left(\mathrm{x}_j^{(i)}\right)^2} \left[\sum_{i \in [n_{\mathrm{sf}}]} \left|\mathrm{x}_j^{(i)}\right| \left\|\hat{\mathbf{C}}_n - \mathbf{C}\right\|_{\mathrm{F}} \|\mathbf{B}^*\|_{\mathrm{F}} \left(\|\mathbf{A}^*\|_{\mathrm{F}} + 1\right) \left\|\mathbf{x}^{(i)}\right\|_2\right]$$

$$= \frac{\sum_{i \in [n_{\mathrm{sf}}]} \left[\left|\mathrm{x}_j^{(i)}\right| \|\mathbf{B}^*\|_{\mathrm{F}} \left(\|\mathbf{A}^*\|_{\mathrm{F}} + 1\right) \left\|\mathbf{x}^{(i)}\right\|_2\right]}{\sum_{i \in [n_{\mathrm{sf}}]} \left(\mathrm{x}_j^{(i)}\right)^2} \left\|\hat{\mathbf{C}}_n - \mathbf{C}\right\|_{\mathrm{F}}.$$

Then

$$\sup_{\hat{\mathbf{C}}_n \in \hat{\mathbb{S}}_n} \inf_{\boldsymbol{C} \in \mathbb{S}^*} \left| k_{n_{\mathrm{sf}}}(\hat{\mathbf{C}}_n) - k_{n_{\mathrm{sf}}}(\boldsymbol{C}) \right|$$

$$\leq \frac{\sum_{i \in [n_{\mathrm{sf}}]} \left[ \left| x_j^{(i)} \right| \|\boldsymbol{B}^*\|_{\mathrm{F}} \left( \|\boldsymbol{A}^*\|_{\mathrm{F}} + 1 \right) \|\mathbf{x}^{(i)}\|_2 \right]}{\sum_{i \in [n_{\mathrm{sf}}]} \left( x_j^{(i)} \right)^2} \sup_{\hat{\mathbf{C}}_n \in \hat{\mathbb{S}}_n} \inf_{\boldsymbol{C} \in \mathbb{S}^*} \left\| \hat{\mathbf{C}}_n - \boldsymbol{C} \right\|_{\mathrm{F}}$$

which implies

$$D(\mathbb{U}_{n_{\mathrm{sf}},n}, \mathbb{U}) \leq \frac{\sum_{i \in [n_{\mathrm{sf}}]} \left[ \left| x_j^{(i)} \right| \|\boldsymbol{B}^*\|_{\mathrm{F}} \left( \|\boldsymbol{A}^*\|_{\mathrm{F}} + 1 \right) \|\mathbf{x}^{(i)}\|_2 \right]}{\sum_{i \in [n_{\mathrm{sf}}]} \left( x_j^{(i)} \right)^2} D(\hat{\mathbb{S}}_n, \mathbb{S}^*). \tag{19}$$

Take the $n_{\mathrm{sf}} \to \infty$ limit over both sides of Eq. (19). With the strong law of large numbers[14] we have

$$\lim_{n_{\mathrm{sf}} \to \infty} D(\mathbb{U}_{n_{\mathrm{sf}},n}, \mathbb{U}) \leq \frac{\mathbb{E}\left[ \left| x_j^{(i)} \right| \|\boldsymbol{B}^*\|_{\mathrm{F}} \left( \|\boldsymbol{A}^*\|_{\mathrm{F}} + 1 \right) \|\mathbf{x}^{(i)}\|_2 \right]}{\mathbb{E}\left[ x_j^{(i)} \right]^2} D(\hat{\mathbb{S}}_n, \mathbb{S}^*), \text{ w.p. } 1.$$

Since $D(\hat{\mathbb{S}}_n, \mathbb{S}^*) \xrightarrow{\text{a.s.}} 0$ as $n \to \infty$, we have $D(\mathbb{U}_{n_{\mathrm{sf}},n}, \mathbb{U}^*) \xrightarrow{\text{a.s.}} 0$ as $n \to \infty$ then $n_{\mathrm{sf}} \to \infty$. $\qquad \square$

### H.1.3 Layer 2 Weights Estimator

In Algorithm 5, we solve $\boldsymbol{B}$ via LR. Let $\hat{\mathbf{z}}^{(i)} = \mathrm{diag}^{-1}(\hat{\mathbf{k}}) \cdot \hat{\mathbf{C}}_n \mathbf{y}^{(i)} \in \mathbb{R}^d$, where $\hat{\mathbf{k}}$ is obtained through Algorithm 4 with input $\hat{\mathbf{C}}_n$. Assume we are using sample size of $n_{\mathrm{w}}$ to do the LR. The optimization problem is

$$\min_{\boldsymbol{B}} \sum_{i \in [n_{\mathrm{w}}]} \left\| \mathbf{y}^{(i)} - \boldsymbol{B}\hat{\mathbf{z}}^{(i)} \right\|^2 \tag{20}$$

Now we present the strong consistency of layer 2 estimator, as described in Theorem H.3.

**Theorem H.3** (strong consistency, layer 2). *Suppose $\hat{\mathbf{B}}_{n_{sf}}$ is the solution to Eq.* (20). *Then $\hat{\mathbf{B}}_{n_w} \xrightarrow{a.s.} \boldsymbol{B}^*$ as $n, n_{sf}, n_w \to \infty$.*

*Proof.* Let $\boldsymbol{\beta}$ denote $\mathrm{vec}\,\boldsymbol{B}$ (flattening $\boldsymbol{B}$ into a vector), then $\boldsymbol{B}\hat{\mathbf{z}}^{(i)} = \left(\hat{\mathbf{z}}^{(i)}\right)^{\top} \otimes \boldsymbol{I}_m$. Here the operation $\otimes$ denotes the Kronecker product. Then we can define an equivalent optimization problem

$$\min_{\boldsymbol{\beta}} \frac{1}{2n_{\mathrm{w}}} \sum_{i \in [n_{\mathrm{w}}]} \left\| \mathbf{y}^{(i)} - \left[ \left(\hat{\mathbf{z}}^{(i)}\right)^{\top} \otimes \boldsymbol{I}_m \right] \boldsymbol{\beta} \right\|^2 .$$

Take the derivatives of $\boldsymbol{\beta}$, we obtain

$$-2 \sum_{i \in [n_{\mathrm{w}}]} \left[ \left[ \left(\hat{\mathbf{z}}^{(i)}\right)^{\top} \otimes \boldsymbol{I}_m \right]^{\top} \left( \mathbf{y}^{(i)} - \left[ \left(\hat{\mathbf{z}}^{(i)}\right)^{\top} \otimes \boldsymbol{I}_m \right] \boldsymbol{\beta} \right) \right] = 0$$

---

[14] Here we assume that the Kolmogorov's strong law assumption on moments (Sen & Singer, 1994) is met as is commonly done in similar analysis.

Then the optimal solution $\hat{\boldsymbol{\beta}}_{n_{\mathrm{w}}}$ of this optimization can be written in closed form:

$$
\begin{aligned}
\hat{\boldsymbol{\beta}}_{n_{\mathrm{w}}} &= \left[ \sum_{i\in[n_{\mathrm{w}}]} \left[ \left(\hat{\mathbf{z}}^{(i)}\right)^{\top} \otimes \boldsymbol{I}_m \right]^{\top} \left[ \left(\hat{\mathbf{z}}^{(i)}\right)^{\top} \otimes \boldsymbol{I}_m \right] \right]^{-1} \left( \sum_{i\in[n_{\mathrm{w}}]} \left[ \left(\hat{\mathbf{z}}^{(i)}\right)^{\top} \otimes \boldsymbol{I}_m \right]^{\top} \mathbf{y}^{(i)} \right) \\
&= \left[ \sum_{i\in[n_{\mathrm{w}}]} \left[ \hat{\mathbf{z}}^{(i)} \otimes \boldsymbol{I}_m \right] \left[ \left(\hat{\mathbf{z}}^{(i)}\right)^{\top} \otimes \boldsymbol{I}_m \right] \right]^{-1} \left( \sum_{i\in[n_{\mathrm{w}}]} \left[ \hat{\mathbf{z}}^{(i)} \otimes \boldsymbol{I}_m \right] \mathbf{y}^{(i)} \right) \\
&= \left[ \sum_{i\in[n_{\mathrm{w}}]} \left[ \hat{\mathbf{z}}^{(i)} \left(\hat{\mathbf{z}}^{(i)}\right)^{\top} \right] \otimes \boldsymbol{I}_m \right]^{-1} \left( \left[ \sum_{i\in[n_{\mathrm{w}}]} \hat{\mathbf{z}}^{(i)} \otimes \boldsymbol{I}_m \right] \mathbf{y}^{(i)} \right) \\
&= \left[ \left[ \sum_{i\in[n_{\mathrm{w}}]} \hat{\mathbf{z}}^{(i)} \left(\hat{\mathbf{z}}^{(i)}\right)^{\top} \right] \otimes \boldsymbol{I}_m \right]^{-1} \left( \left[ \sum_{i\in[n_{\mathrm{w}}]} \hat{\mathbf{z}}^{(i)} \otimes \boldsymbol{I}_m \right] \mathbf{y}^{(i)} \right)
\end{aligned}
$$

We inherit the notation from the last two subsections. By Lemma H.1, $d(\hat{\mathbf{C}}_n, \mathbb{S}^*) \xrightarrow{\text{a.s.}} 0$ as $n \to \infty$. Thus $\forall\, \varepsilon > 0$, $\exists\, N$ such that $\forall\, n \geq N$, $d(\hat{\mathbf{C}}_n, \mathbb{S}^*) \leq \varepsilon$ w.p. 1, i.e. $\exists\, \boldsymbol{C}_n \in \mathbb{S}^*$ s.t. $d(\hat{\mathbf{C}}_n, \boldsymbol{C}_n) \leq \varepsilon$ w.p. 1. Then by Lemma H.2, $\exists\, K > 0$, for $\varepsilon$ that is small enough, $\exists\, N_{\mathrm{sf}}$ such that $\forall\, n_{\mathrm{sf}} \geq N_{\mathrm{sf}}$, we have $\left| \mathrm{k}_{n_{\mathrm{sf}}}(\hat{\mathbf{C}}_n) - \mathrm{k}_{n_{\mathrm{sf}}}(\boldsymbol{C}_n) \right| \leq K\varepsilon$ w.p.1. For simplicity, we omit the under-script $n_{\mathrm{sf}}$ of $\mathrm{k}_{n_{\mathrm{sf}}}$ in the following discussion. In fact,

$$
\begin{aligned}
\left| \hat{\mathbf{z}}_j^{(i)} - \mathbf{z}_j^{(i)} \right| &= \left| \frac{1}{\mathrm{k}(\hat{\mathbf{C}}_n)} \left(\boldsymbol{e}^{(i)}\right)^{\top} \hat{\mathbf{C}}_n \mathbf{y}^{(i)} - \frac{1}{\mathrm{k}(\boldsymbol{C}_n)} \left(\boldsymbol{e}^{(i)}\right)^{\top} \boldsymbol{C}_n \mathbf{y}^{(i)} \right| \\
&= \left| \frac{1}{\mathrm{k}(\hat{\mathbf{C}}_n)\mathrm{k}(\boldsymbol{C}_n)} \left[ \mathrm{k}(\boldsymbol{C}_n) \left(\boldsymbol{e}^{(i)}\right)^{\top} \hat{\mathbf{C}}_n - \mathrm{k}(\hat{\mathbf{C}}_n) \left(\boldsymbol{e}^{(i)}\right)^{\top} \boldsymbol{C}_n \right] \mathbf{y}^{(i)} \right| \\
&\leq \left| \frac{1}{\mathrm{k}(\hat{\mathbf{C}}_n)\mathrm{k}(\boldsymbol{C}_n)} \right| \left| \mathrm{k}(\boldsymbol{C}_n) \left(\boldsymbol{e}^{(i)}\right)^{\top} \hat{\mathbf{C}}_n \mathbf{y}^{(i)} - \mathrm{k}(\boldsymbol{C}_n) \left(\boldsymbol{e}^{(i)}\right)^{\top} \boldsymbol{C}_n \mathbf{y}^{(i)} \right| \\
&\quad + \left| \frac{1}{\mathrm{k}(\hat{\mathbf{C}}_n)\mathrm{k}(\boldsymbol{C}_n)} \right| \left| \mathrm{k}(\boldsymbol{C}_n) \left(\boldsymbol{e}^{(i)}\right)^{\top} \boldsymbol{C}_n \mathbf{y}^{(i)} - \mathrm{k}(\hat{\mathbf{C}}_n) \left(\boldsymbol{e}^{(i)}\right)^{\top} \boldsymbol{C}_n \mathbf{y}^{(i)} \right| \\
&\leq \frac{\left(1 + \boldsymbol{A}_{j,j}^*\right)^2}{1 - K\varepsilon\left(1 + \boldsymbol{A}_{j,j}^*\right)} \left( \left\|\mathbf{y}^{(i)}\right\| \left\|\hat{\mathbf{C}}_n - \boldsymbol{C}_n\right\|_{\mathrm{F}} + \left|\mathrm{k}(\hat{\mathbf{C}}_n) - \mathrm{k}(\boldsymbol{C}_n)\right| \left\|\boldsymbol{C}_n \boldsymbol{B}^*\right\|_{\mathrm{F}} \left(\left\|\boldsymbol{A}^*\right\|_{\mathrm{F}} + 1\right) \left\|\mathbf{x}^{(i)}\right\| \right) \\
&\leq \frac{\left(1 + \boldsymbol{A}_{j,j}^*\right)^2}{1 - K\varepsilon\left(1 + \boldsymbol{A}_{j,j}^*\right)} \left( \left\|\mathbf{y}^{(i)}\right\| + dK\left(\left\|\boldsymbol{A}^*\right\|_{\mathrm{F}} + 1\right) \left\|\mathbf{x}^{(i)}\right\| \right) \varepsilon \qquad (21)
\end{aligned}
$$

Then

$$
\begin{aligned}
&\left\| \hat{\mathbf{z}}^{(i)} \left(\hat{\mathbf{z}}^{(i)}\right)^{\top} - \mathbf{z}^{(i)} \left(\mathbf{z}^{(i)}\right)^{\top} \right\|_{\mathrm{F}} \\
&= \left\| \hat{\mathbf{z}}^{(i)} \left(\hat{\mathbf{z}}^{(i)}\right)^{\top} - \mathbf{z}^{(i)} \left(\hat{\mathbf{z}}^{(i)}\right)^{\top} + \mathbf{z}^{(i)} \left(\hat{\mathbf{z}}^{(i)}\right)^{\top} - \mathbf{z}^{(i)} \left(\mathbf{z}^{(i)}\right)^{\top} \right\|_{\mathrm{F}} \\
&\leq \left\| \left[\hat{\mathbf{z}}^{(i)} - \mathbf{z}^{(i)}\right] \left(\hat{\mathbf{z}}^{(i)}\right)^{\top} \right\|_{\mathrm{F}} + \left\| \mathbf{z}^{(i)} \left[ \left(\hat{\mathbf{z}}^{(i)}\right)^{\top} - \left(\mathbf{z}^{(i)}\right)^{\top} \right] \right\|_{\mathrm{F}} \\
&\leq \left\| \left[\hat{\mathbf{z}}^{(i)} - \mathbf{z}^{(i)}\right] \left[\hat{\mathbf{z}}^{(i)} - \mathbf{z}^{(i)}\right]^{\top} \right\|_{\mathrm{F}} + 2 \left\| \mathbf{z}^{(i)} \left[ \left(\hat{\mathbf{z}}^{(i)}\right)^{\top} - \left(\mathbf{z}^{(i)}\right)^{\top} \right] \right\|_{\mathrm{F}} \\
&\leq \sum_{j\in[d]} \left( \hat{\mathbf{z}}_j^{(i)} - \mathbf{z}_j^{(i)} \right)^2 + 2 \left\|\mathbf{z}^{(i)}\right\| \left\|\hat{\mathbf{z}}^{(i)} - \mathbf{z}^{(i)}\right\|
\end{aligned}
$$

It follows that $\left\| \hat{\mathbf{z}}^{(i)} \left( \hat{\mathbf{z}}^{(i)} \right)^{\top} - \mathbf{z}^{(i)} \left( \mathbf{z}^{(i)} \right)^{\top} \right\|_{\mathrm{F}}$ is also bounded by $\mathcal{O}(\varepsilon)$. With similar techniques we can prove

$$\left\| \sum_{i \in [n_{\mathrm{w}}]} \hat{\mathbf{z}}^{(i)} \left( \hat{\mathbf{z}}^{(i)} \right)^{\top} - \sum_{i \in [n_{\mathrm{w}}]} \mathbf{z}^{(i)} \left( \mathbf{z}^{(i)} \right)^{\top} \right\|_{\mathrm{F}} \leq \mathcal{O}(\varepsilon)$$

and

$$\left\| \hat{\mathbf{z}}^{(i)} \otimes \boldsymbol{I}_m - \mathbf{z}^{(i)} \otimes \boldsymbol{I}_m \right\|_{\mathrm{F}} \leq \mathcal{O}(\varepsilon)$$

Denote $\left[ \sum_{i \in [n_{\mathrm{w}}]} \mathbf{z}^{(i)} \left( \mathbf{z}^{(i)} \right)^{\top} \right]$ as $\boldsymbol{P}$ and $\sum_{i \in [n_{\mathrm{w}}]} \mathbf{z}^{(i)}$ as $\boldsymbol{Q}$. Substitute $\mathbf{z}^{(i)}$ with $\hat{\mathbf{z}}^{(i)}$ in the above expression we have $\hat{\boldsymbol{P}}$ and $\hat{\boldsymbol{Q}}$. Hence,

$$
\begin{aligned}
\left\| \hat{\beta}_{n_{\mathrm{w}}} - \beta^* \right\|_{\mathrm{F}} &= \left\| \left[ \hat{\boldsymbol{P}} \otimes \boldsymbol{I}_m \right]^{-1} \left( \left[ \hat{\boldsymbol{Q}} \otimes \boldsymbol{I}_m \right] \mathbf{y}^{(i)} \right) - \left[ \boldsymbol{P} \otimes \boldsymbol{I}_m \right]^{-1} \left( \left[ \boldsymbol{Q} \otimes \boldsymbol{I}_m \right] \mathbf{y}^{(i)} \right) \right\|_{\mathrm{F}} \\
&\leq \left\| \left[ \hat{\boldsymbol{P}} \otimes \boldsymbol{I}_m \right]^{-1} \left( \left[ \hat{\boldsymbol{Q}} \otimes \boldsymbol{I}_m \right] \mathbf{y}^{(i)} \right) - \left[ \hat{\boldsymbol{P}} \otimes \boldsymbol{I}_m \right]^{-1} \left( \left[ \boldsymbol{Q} \otimes \boldsymbol{I}_m \right] \mathbf{y}^{(i)} \right) \right\|_{\mathrm{F}} \\
&\quad + \left\| \left[ \hat{\boldsymbol{P}} \otimes \boldsymbol{I}_m \right]^{-1} \left( \left[ \boldsymbol{Q} \otimes \boldsymbol{I}_m \right] \mathbf{y}^{(i)} \right) - \left[ \boldsymbol{P} \otimes \boldsymbol{I}_m \right]^{-1} \left( \left[ \boldsymbol{Q} \otimes \boldsymbol{I}_m \right] \mathbf{y}^{(i)} \right) \right\|_{\mathrm{F}} \\
&= \left\| \left[ \hat{\boldsymbol{P}} \otimes \boldsymbol{I}_m \right]^{-1} \right\|_{\mathrm{F}} \left\| \left[ \left( \hat{\boldsymbol{Q}} - \boldsymbol{Q} \right) \otimes \boldsymbol{I}_m \right] \mathbf{y}^{(i)} \right\|_{\mathrm{F}} + \left\| \left[ \hat{\boldsymbol{P}} \otimes \boldsymbol{I}_m \right]^{-1} - \left[ \boldsymbol{P} \otimes \boldsymbol{I}_m \right]^{-1} \right\|_{\mathrm{F}} \\
&\quad \left\| \left[ \boldsymbol{Q} \otimes \boldsymbol{I}_m \right] \mathbf{y}^{(i)} \right\|_{\mathrm{F}}
\end{aligned}
$$

In the first part, by triangle inequality,

$$\left\| \left[ \hat{\boldsymbol{P}} \otimes \boldsymbol{I}_m \right]^{-1} \right\|_{\mathrm{F}} \leq \left\| \left[ \hat{\boldsymbol{P}} \otimes \boldsymbol{I}_m \right]^{-1} - \left[ \boldsymbol{P} \otimes \boldsymbol{I}_m \right]^{-1} \right\|_{\mathrm{F}} + \left\| \left[ \boldsymbol{P} \otimes \boldsymbol{I}_m \right]^{-1} \right\|_{\mathrm{F}}$$

So we only need to prove $\left\| \left[ \hat{\boldsymbol{P}} \otimes \boldsymbol{I}_m \right]^{-1} - \left[ \boldsymbol{P} \otimes \boldsymbol{I}_m \right]^{-1} \right\|_{\mathrm{F}} \leq \mathcal{O}(\varepsilon)$ to claim $\left\| \hat{\beta}_{n_{\mathrm{w}}} - \beta^* \right\|_{\mathrm{F}} \leq \mathcal{O}(\varepsilon)$. Denote $\hat{\boldsymbol{P}} - \boldsymbol{P} = \Delta \boldsymbol{P}$. From Eq. (21), we know every entry of $\Delta \boldsymbol{P} \otimes \boldsymbol{I}_m$ can be bounded by $\mathcal{O}(\varepsilon)$. By simple calculation we have

$$(\boldsymbol{P} + \Delta \boldsymbol{P})^{-1} = \boldsymbol{P}^{-1} - \boldsymbol{P}^{-1} \Delta \boldsymbol{P} \boldsymbol{P}^{-1} + \mathcal{O}(\varepsilon^2).$$

Then we have

$$\boldsymbol{P}^{-1} - \hat{\boldsymbol{P}}^{-1} = \boldsymbol{P}^{-1} \Delta \boldsymbol{P} \boldsymbol{P}^{-1} + \mathcal{O}(\varepsilon^2) = \mathcal{O}(\varepsilon).$$

$\square$

## H.2  Layer 1

In this subsection, we justify the strong consistency of layer 1 objective functional minimizer estimator in detail, i.e. the layer 1 QP/LP solution space. We will omit the detailed proof of Algorithm 2 line 4 to 7 strong consistency since it is similar to the proof of Lemma H.2. Additionally, we also omit the strong consistency of the $\boldsymbol{x} \mapsto (\boldsymbol{A}^* \boldsymbol{x})^+$ function estimator because it can be directly obtained by Lemmas H.1 and H.2 and the continuous mapping theorem.

We use a new optimization problem equivalent to the optimization of $G_1$. Before that, we first define the equivalence between two optimization problems as follows.

**Definition H.3.** Let opt1 and opt2 be two optimization problems, with $f_1$, $f_2$ as the respective objective functions. Then opt1 and opt2 are said to be equivalent if given a feasible solution to opt1, namely $\boldsymbol{x}_1$, a feasible solution to opt2 is uniquely corresponded, namely $\boldsymbol{x}_2$, such that $f_1(\boldsymbol{x}_1) = f_2(\boldsymbol{x}_2)$, and vice versa.

The new optimization problem and its equivalence to the optimization of $G_1$ is described in Lemma H.4.

**Lemma H.4.** *The optimization of $G_1$ (Eq. (6)) is equivalent to*

$$\min_{\boldsymbol{A}} f(\boldsymbol{A}) = \frac{1}{2} \mathbb{E}_{\mathbf{x}} \left[ \left\| (\boldsymbol{A}\mathbf{x} - \mathbf{h})^+ \right\|^2 \right]. \tag{22}$$

*Proof.* To see this, suppose $\boldsymbol{A}_1$ is one optimal solution to Eq. (22), then we can construct $r_1(\boldsymbol{x}) = (\boldsymbol{A}_1\boldsymbol{x} - \boldsymbol{h})^+$ so that $G_1(\boldsymbol{A}_1, r_1) = f(\boldsymbol{A}_1)$ and the optimality implies

$$\min_{\boldsymbol{A}, r} G_1(\boldsymbol{A}, r) \le \min_{\boldsymbol{A}} f(\boldsymbol{A}).$$

On the other hand, suppose $(\boldsymbol{A}_2, r_2)$ is an optimum of $G_1$. Let $r_3(\boldsymbol{x}) = (\boldsymbol{h} - \boldsymbol{A}_2\boldsymbol{x})^+$, then $\forall \boldsymbol{x} \in \mathbb{R}^d$ and $\boldsymbol{h}$ be the corresponding hidden output, if $[\boldsymbol{h} - \boldsymbol{A}_2\boldsymbol{x}]_j \ge 0$, then $[r_3(\boldsymbol{x}) + \boldsymbol{A}_2\boldsymbol{x} - \boldsymbol{h}]_j = 0$, otherwise $[r_3(\boldsymbol{x}) + \boldsymbol{A}_2\boldsymbol{x} - \boldsymbol{h}]_j^2 = [\boldsymbol{A}_2\boldsymbol{x} - \boldsymbol{h}]^2 \le [h_2(\mathbf{x}) + \boldsymbol{A}_2\boldsymbol{x} - \boldsymbol{h}]_j^2$ since $h_2$ is nonnegative. So that we know that

$$\min_{\boldsymbol{A}, r} G_1(\boldsymbol{A}, r) = G_1(\boldsymbol{A}_2, r_2) = G_1(\boldsymbol{A}_2, r_3) = f(\boldsymbol{A}_2)$$

From the optimality, we further have

$$\min_{\boldsymbol{A}, r} G_1(\boldsymbol{A}, r) \ge \min_{\boldsymbol{A}} f(\boldsymbol{A})$$

From the simple calculation above, we can see that one optimal solution to Eq. (22) has a one-to-one correspondence to an optimal solution to $G_1$. $\qquad \square$

Similarly, the empirical version of the two problems are equivalent, which indicates their consistency in the empirical estimation being equivalent. In the following, we justify the strong consistency of empirical Eq. (22) instead of $G_1$

$$\min_{\boldsymbol{A}} \hat{f}_n(\boldsymbol{A}) = \frac{1}{2n} \sum_{i \in [n]} \left\| \left( \boldsymbol{A}\boldsymbol{x}^{(i)} - \boldsymbol{h}^{(i)} \right)^+ \right\|^2.$$

Denote $\mathbb{T}^* := \{\mathrm{diag}(\boldsymbol{k}) \cdot \boldsymbol{A}^* \mid 0 \le k_j \le 1, j \in [d]\}$ as the true optimal solution set, and $\hat{\mathbb{T}}_n$ as the optimal solution set corresponding to the $n$-sample problem. In the following, we justify four conditions in a row that hold for $f$ to derive the strong consistency of its optimal solution estimator.

**Lemma H.5.** *Let $\hat{\mathbb{T}}_{n'}$ be the layer 1 QP/LP solution space by $n'$ samples. Then there exists a compact set $\mathbb{C}$ determined by $\boldsymbol{A}^*$, namely $\mathbb{C}(\boldsymbol{A}^*)$, s.t. $\hat{\mathbb{T}}_{n'} \subset \mathbb{C}(\boldsymbol{A}^*)$ w.p. 1 as $n \to \infty$.*

*Proof.* $\forall l \in [d]$, let $\boldsymbol{a}^*$ be the $l$-th row of $\boldsymbol{A}^*$, and $\hat{\boldsymbol{a}}_l$ be the $l$-th row of $\hat{\boldsymbol{A}}_{n'}$. We'd like first to prove that the set

$$\hat{\mathbb{T}}_{n'}^l = \{\boldsymbol{a}_l : \boldsymbol{a}_l \text{ is the } l\text{-th row of } \boldsymbol{A}, \text{ where } \boldsymbol{A} \in \hat{\mathbb{T}}_{n'}\}$$

is compact w.p.1.

Suppose $n' > d$, and among the $n'$ samples, we classify them into two folds. To avoid ambiguity, let $\boldsymbol{u}^{(i)}$ be the points such that $(\boldsymbol{a}^*)^\top \boldsymbol{u}^{(i)} > 0$, $i \in [q]$; and $\boldsymbol{v}^{(j)}$ be the points such that $(\boldsymbol{a}^*)^\top \boldsymbol{v}^{(j)} < 0$, $j \in [n-q]$. From the analysis of Theorem 5.1, we have $(\boldsymbol{a}^*)^\top \boldsymbol{u}^{(i)} \ge \hat{\boldsymbol{a}}^\top \boldsymbol{u}^{(i)}$, $\forall i \in [q]$ and $\hat{\boldsymbol{a}}^\top \boldsymbol{v}^{(j)} \le 0$, $\forall j \in [n-q]$. It follows that we can rewrite $\hat{\mathbb{T}}_{n'}^l$ as a polyhedron

$$\hat{\mathbb{T}}_{n'}^l = \{\boldsymbol{a} \in \mathbb{R}^d : \boldsymbol{a}^\top u^{(i)} \le (\boldsymbol{a}^*)^\top \boldsymbol{u}^{(i)}, \boldsymbol{a}^\top \boldsymbol{v}^{(j)} \le 0\}$$

We are going to show the polyhedron $\hat{\mathbb{T}}_{n'}^l$ is bounded by contradiction. If it is not bounded, then $\exists \boldsymbol{d} \in \mathbb{R}^d$, $\boldsymbol{d} \ne \boldsymbol{0}$ and $\tilde{\boldsymbol{a}} \in \hat{\mathbb{T}}_{n'}^l$, such that $\forall \lambda > 0$, $\tilde{\boldsymbol{a}} + \lambda \boldsymbol{d} \in \hat{\mathbb{T}}_{n'}^l$. Then

$$(\tilde{\boldsymbol{a}} + \lambda \boldsymbol{d})^\top \boldsymbol{u}^{(i)} = \tilde{\boldsymbol{a}}^\top \boldsymbol{u}^{(i)} + \lambda \boldsymbol{d}^\top \boldsymbol{u}^{(i)} \le (\boldsymbol{a}^*)^\top \boldsymbol{u}^{(i)} \iff \lambda \boldsymbol{d}^\top \boldsymbol{u}^{(i)} \le (\boldsymbol{a}^*)^\top \boldsymbol{u}^{(i)} - \tilde{\boldsymbol{a}}^\top \boldsymbol{u}^{(i)}$$

similarly,

$$(\tilde{\boldsymbol{a}} + \lambda \boldsymbol{d})^\top \boldsymbol{v}^{(i)} = \tilde{\boldsymbol{a}}^\top \boldsymbol{v}^{(j)} + \lambda \boldsymbol{d}^\top \boldsymbol{v}^{(j)} \le 0$$

From the definition, $\lambda$ can be arbitrarily big, then $\boldsymbol{d}^\top \boldsymbol{u}^{(i)} \leq 0$, $\forall i \in [q]$, and $\boldsymbol{d}^\top \boldsymbol{v}^{(j)} \leq 0$, $\forall j \in [n-q]$.

Since we know $\mathrm{span}\{\boldsymbol{u}^{(i)}\} = \mathbb{R}^d$ w.p. 1, then there $\exists$ some $i^*$ such that $\boldsymbol{d}^\top \boldsymbol{u}^{(i^*)} < 0$, w.p. 1. (Otherwise if $\boldsymbol{d}^\top \boldsymbol{u}^{(i)} = 0$ for all $i \in [q]$, then either $\mathrm{span}\{\boldsymbol{u}^{(i)}\} \neq \mathbb{R}^d$ or $\boldsymbol{d} = \boldsymbol{0}$.) Under our assumption that, $\hat{\mathbb{T}}_{n'}^l$ is not bounded, we know the following system (w.r.t $\boldsymbol{x}$) has a feasible solution w.p. 1

$$\begin{bmatrix} -\boldsymbol{U} \\ -\boldsymbol{V} \end{bmatrix} \boldsymbol{x} \geq 0, \ \boldsymbol{x}^\top \boldsymbol{u}^{(i^*)} < 0 \tag{I}$$

where every row of $\boldsymbol{U}$ and $\boldsymbol{V}$ is $\left(\boldsymbol{u}^{(i)}\right)^\top$ and $\left(\boldsymbol{v}^{(j)}\right)^\top$ respectively. By Farkas' Lemma (Farkas, 1902), the system

$$[-\boldsymbol{U}^\top, -\boldsymbol{V}^\top] \cdot \boldsymbol{x} = \boldsymbol{u}^{(i^*)}, \ \boldsymbol{x} \geq 0 \tag{II}$$

is not feasible (w.p. 1). We claim that $\boldsymbol{u}^{(i^*)}$ lies in the conic hull of $-\boldsymbol{v}^{(j)}$'s w.p. 1. So that the second system actually has a feasible solution and thus it raises the contradiction.

Denote the conic hull as

$$\mathbb{H} = \left\{ \boldsymbol{t} \in \mathbb{R}^d : \boldsymbol{t} = \sum_{j \in [n-q]} \lambda_j \left(-\boldsymbol{v}^{(j)}\right), \lambda_j \geq 0 \text{ for } \forall j \in [n-q] \right\}$$

Now suppose $\boldsymbol{u}^{(i^*)} \notin \mathbb{H}$, by the supporting hyperplane theorem (Luenberger, 1997), $\exists \boldsymbol{b} \in \mathbb{R}^d$, $\boldsymbol{b} \neq \boldsymbol{0}$, such that $\boldsymbol{b}^\top \boldsymbol{u}^{(i^*)} \leq \boldsymbol{b}^\top \boldsymbol{t}$ for $\forall \boldsymbol{t} \in \mathbb{H}$. Then by definition,

$$-\boldsymbol{v}^{(j)} \in \{\boldsymbol{t} : \boldsymbol{b}^\top \boldsymbol{t} \leq \boldsymbol{b}^\top \boldsymbol{u}^{(i^*)}\}, \text{ for } \forall j \in [n-q]$$

Denote the hyperplane $\mathbb{J} = \{\boldsymbol{t} : \boldsymbol{b}^\top \boldsymbol{t} \leq \boldsymbol{b}^\top \boldsymbol{u}^{(i^*)}\}$, then

$$P\left(\boldsymbol{u}^{(i^*)} \notin \mathbb{H}\right) \leq P\left(-\boldsymbol{v}^{(j)} \in \mathbb{J}, \text{ for } \forall j \in [n-q]\right) = P\left(-\boldsymbol{v}^{(j)} \in \mathbb{J}\right)^{n-q}$$

Since we have in this case a geometric sequence, we know its infinite sum is bounded. By Borel-Cantelli lemma (Borel, 1909), we conclude that $\boldsymbol{u}^{(i^*)} \in \mathbb{H}$ w.p. 1. Then system II is feasible w.p. 1. So that $\hat{\mathbb{T}}_{n'}^l$ is compact w.p. 1.

Now we prove that there exists a compact set $\mathbb{C}(\boldsymbol{A}^*)$, s.t. $\hat{\mathbb{T}}_{n'} \subset \mathbb{C}(\boldsymbol{A}^*)$ w.p. 1 as $n \to \infty$. Similarly, we focus on the analysis of one row. As discussed above, $\hat{\mathbb{T}}_{n'}^l$ is compact w.p. 1. Let $\hat{\mathbb{W}}_{n'}^1$ be the set of all $\boldsymbol{u}^{(i)}$ sampled in estimating $\hat{\mathbb{T}}_{n'}$, and $\hat{\mathbb{W}}_{n'}^2$ be the set of all $\boldsymbol{v}^{(j)}$ sampled. Now for another set $\hat{\mathbb{T}}_{n''}$, similarly define sample point sets $\hat{\mathbb{W}}_{n''}^1$ and $\hat{\mathbb{W}}_{n''}^2$. We claim that $\forall \boldsymbol{u}^{(i)} \in \hat{\mathbb{W}}_{n'}^1$, $\boldsymbol{u}^{(i)}$ lies in the conic hull of $\hat{\mathbb{W}}_{n''}^1$. Actually, this part of the proof is very similar to the way we prove $\boldsymbol{u}^{(i^*)} \in \mathbb{H}$ w.p. 1, so we will omit it here.

$\forall \boldsymbol{a} \in \hat{\mathbb{T}}_{n''}$, let $\boldsymbol{x}^{(1)}$ and $\boldsymbol{x}^{(2)}$ be two different points in $\hat{\mathbb{W}}_{n''}^1$. Then $\boldsymbol{a}^\top (\lambda_1 \boldsymbol{x}^{(1)} + \lambda_2 \boldsymbol{x}^{(2)}) \leq (\boldsymbol{a}^*)^\top (\lambda_1 \boldsymbol{x}^{(1)} + \lambda_2 \boldsymbol{x}^{(2)})$ for $\forall \lambda_1 \geq 0$ and $\lambda_2 \geq 0$. This simple calculation reveals $\boldsymbol{a}^\top \boldsymbol{u}^{(i)} \leq (\boldsymbol{a}^*)^\top \boldsymbol{u}^{(i)}$ for $\forall \boldsymbol{u}^{(i)} \in \hat{\mathbb{W}}_{n'}^1$ (from the claim we made). This implies that $\hat{\mathbb{T}}_{n''} \subset \mathbb{C}(\boldsymbol{A}^*)$ w.p. 1 as $n \to \infty$, too. $\qquad \square$

**Lemma H.6.** *The minimizer space of $f(\boldsymbol{A})$, i.e. $\mathbb{T}^* = \{\mathrm{diag}(\boldsymbol{k}) \cdot \boldsymbol{A}^* \mid 0 \leq k_j \leq 1, j \in [d]\}$, is contained in $\mathbb{C}(\boldsymbol{A}^*)$.*

**Lemma H.7.** *$f(\boldsymbol{A})$ is finite valued and continuous on $\mathbb{C}(\boldsymbol{A}^*)$.*

Lemmas H.6 and H.7 are easily obtained by the formulation of $f$ (see Eq. (22)) and Lemma H.5.

**Lemma H.8** (uniform a.s. convergence). *$\hat{f}_n(\boldsymbol{A}) \xrightarrow{a.s.} f(\boldsymbol{A})$ as $n \to \infty$, uniformly in $\boldsymbol{A} \in \mathbb{C}(\boldsymbol{A}^*)$.*

*Proof.* Name single-sample objective $g(\boldsymbol{x}, \boldsymbol{A}) = \frac{1}{2} \left\| (\boldsymbol{A}\boldsymbol{x} - \boldsymbol{h})^+ \right\|^2$. The uniform a.s. convergence is guaranteed by the uniform law of large numbers (Jennrich, 1969):

    a) By Lemma H.5, $\mathbb{C}(\boldsymbol{A}^*)$ is a compact set.
    b) $g$ is continuous w.r.t. $\boldsymbol{A}$ by its formulation and measurable over $\boldsymbol{x}$ at each $\boldsymbol{A} \in \mathbb{C}(\boldsymbol{A}^*)$.

c) In fact,

$$
\begin{aligned}
g(\boldsymbol{x}, \boldsymbol{A}) &= \frac{1}{2} \left\| (\boldsymbol{A}\boldsymbol{x} - \boldsymbol{h})^+ \right\|^2 \\
&\leq \|\boldsymbol{A}\boldsymbol{x}\|^2 + \|\boldsymbol{A}^*\boldsymbol{x}\|^2 \\
&\leq (\|\boldsymbol{A}\|_{\mathrm{F}} + \|\boldsymbol{A}^*\|_{\mathrm{F}}) \|\boldsymbol{x}\|^2.
\end{aligned}
$$

Since $\boldsymbol{A} \in \mathbb{C}(\boldsymbol{A}^*)$ is in a compact set,

$$
g(\boldsymbol{x}, \boldsymbol{A}) \leq \left[ \sup_{\boldsymbol{A} \in \mathbb{C}(\boldsymbol{A}^*)} \|\boldsymbol{A}\|_{\mathrm{F}} + \|\boldsymbol{A}^*\|_{\mathrm{F}} \right] \|\boldsymbol{x}\|^2.
$$

Thus the dominating function exists.[15]

$\square$

By Lemmas H.5 to H.8, all of the conditions are satisfied in (Shapiro et al., 2014, Thm. 5.3). Thus, we have the strong consistency of layer 1 objective optima estimator as described in Lemma H.9.

**Lemma H.9** (QP/LP strong consistency, layer 1)**.** $D_{\mathrm{H}}(\hat{\mathbb{T}}_{n'}, \mathbb{T}^*) \xrightarrow{a.s.} 0$ *as* $n \to \infty$.

Similar to Lemma H.2, we have the strong consistency of the layer 1 scale factor estimator as described in Lemma H.10.

**Lemma H.10** (scale factor estimator strong consistency, layer 1)**.** *Let* $\mathrm{k}_{n'_{sf}}(\hat{\mathbf{C}}_n)$ *be the* $n'_{sf}$*-sample estimator of* $k_j$ *via LR: Algorithm 2 line 5. given that* $\mathrm{h}_j^{(i)} > 0$. *Define sets*

$$
\mathbb{V}_{n'_{sf}, n} \coloneqq \{ \mathrm{k}_{n'_{sf}}(\boldsymbol{A}) : \boldsymbol{A} \in \hat{\mathbb{T}}_{n'} \}, \ and \ \mathbb{V}^* \coloneqq [0, 1].
$$

*Then* $\lim_{n'_{sf} \to \infty} \lim_{n' \to \infty} D_{\mathrm{H}}(\mathbb{V}_{n'_{sf}, n'}, \mathbb{V}^*) \overset{a.s.}{=\!=} 0$.

*Remark.* In case $k_j = 0$, suppose the algorithm finds a solution over a continuous distribution with $[0, 1]$ as support and the probability that it finds a solution with scale factor 0 is 0.

With Theorem 5.2 and the continuous mapping theorem, the strong consistency of layer 1 estimation is guaranteed.

**Theorem H.11** (strong consistency, layer 1)**.** *Suppose* $\hat{\mathbf{A}}_{n'}$ *is scaled by* $\hat{\mathbf{k}}_{n'_{sf}}$. *Then* $\hat{\mathbf{A}}_{n'} \xrightarrow{a.s.} \boldsymbol{A}^*$ *as* $n', n'_{sf} \to \infty$.

By Theorems H.3 and H.11, Theorem 6.3 is guaranteed by the continuous mapping theorem.

## I   Sample Complexity Results

In this appendix, we begin with an intuition that justifies how our core QP/LP approaches eliminate the exponential sample efficiency compared to the vanilla LR approach. In layer 2 learning, $\boldsymbol{C}$'s optimization is to find a feasible point in the space determined by $n$ inequalities each of which corresponds to a sample. Taking the $j$-th row of the inequality, i.e. $C_{j,:}\boldsymbol{y} \geq x_j$. Every time we get a new sample $\boldsymbol{x}^{(i)}, \boldsymbol{y}^{(i)}$ in,

- The inequality $C_{j,:}\boldsymbol{y}^{(i)} \geq x_j^{(i)}$ eliminates the solution space of $C_{j,:}$ by one of the spaces divided by plane $C_{j,:}\boldsymbol{y}^{(i)} = x_j^{(i)}$. This property guarantees fast convergence speed at early phase since $C_{j,:}$'s feasibility starts from $\mathbb{R}^d$.

---

[15] Here, we assume $\mathbb{E}[\|\mathbf{x}\|^2] < +\infty$ as is commonly done in empirical analysis.

- In fact, when ReLU does not activate $\boldsymbol{A}^*\mathbf{x}^{(i)}$ at the $j$-th row, i.e. $\boldsymbol{A}^*_{j,:}\mathbf{x}^{(i)} \leq 0$, the theoretical solution of $C_{j,:}$ that $C_{j,:}\boldsymbol{B}^* = (0, \ldots, 0, 1, 0, \ldots, 0)$ (a 1 at index $j$) directly lies on the separating plane $C_{j,:}\boldsymbol{y}^{(i)} = x_j^{(i)}$:

$$C_{j,:}\boldsymbol{y}^{(i)} = C_{j,:}\boldsymbol{B}^* \left[ \left( \boldsymbol{A}^*\boldsymbol{x}^{(i)} \right)^+ + \boldsymbol{x}^{(i)} \right] = x_j^{(i)}.$$

  This property remarkably speeds up further constraints on the solution space of $C_{j,:}$ to the correct estimate and is with high probability, since it directly depends on the sign of $\boldsymbol{A}^*\mathbf{x}^{(i)}$.

With the vanilla LR approach mentioned in Section 3 all the dimensions need to have the correct sign at the same time. This is the reason for the exponential complexity of this approach. Our main algorithm only requires one dimension to get the correct sign for a single sample, which avoids the exponential sample size expectation. The convergence speed is determined by the actual probability of each dimension not getting activated by ReLU, and such probability is determined by specified input distribution.

**Roadmap**   Our sample complexity results follow several intermediate results. We first demonstrate that the constraints that define our optimization objective give a bounded polytope (Theorem I.2). This happens with a high probability, given a set of samples. Once we have shown that, we then show that the optimization problem constraints we use define a polytope with a small diameter with a high probability given a sufficient number of samples, where the true network parameters exist. The boundnesses of the constraint polytope is required for this result.

**Details**   We follow the above intuitive explanation with a more rigorous analysis that describes the sample complexity of recovering $C_{j,:}$ for $j \in [d]$.

Let us assume we have $n$ samples of $\boldsymbol{y}^{(i)}$, $i \in [n]$. We denote by $F \in \mathbb{R}^{n \times d}$ such that $F_{ij} = -y_j^{(i)}$. We assume $n > d$. We denote by $L_j = \mathbb{E}[\mathbf{x}_j^2]$.

**Condition I.1** (Haar condition on samples).  $F$ satisfies the Haar condition, i.e. that every submatrix of size $d \times d$ is invertible, almost surely.

We will assume that Condition I.1 holds for the rest of the discussion. For a discussion of the Haar condition, see Schmidt & Mattheiss (1977).

Fix $j \in [d]$. The first step in our process is to show that with high likelihood, the inequalities $C_{j,:}\boldsymbol{y}^{(i)} \geq x_j^{(i)}$ for $i \in [n]$, or equivalently $C_{j,:}(-\boldsymbol{y}^{(i)}) \leq (-x_j^{(i)})$, or equivalently $FC_{j,:} \leq -x_j^{(:)}$ define a bounded polytope with high probability.

Schmidt & Mattheiss (1977) describe a necessary and sufficient condition for this set of inequalities, which describe an intersection of half-spaces, to be bounded. More specifically, under Condition I.1, such an intersection is unbounded if and only if the subspace spanned by the columns of $F$ intersected with the negative quadrant of $\mathbb{R}^n$ is non-empty.

This means that this halfspace intersection is unbounded if and only if there exists $\alpha \in \mathbb{R}^d$, $\alpha \neq 0$, such that $F_{i,:}\alpha \leq 0$ for $i \in [n]$.

For a given $\alpha \in \mathbb{R}^d$, $\alpha \neq 0$, let $h_\alpha(\boldsymbol{y}) = \text{sgn}(\sum_{i=1}^d \alpha_i \boldsymbol{y}_i)$ where $\text{sgn}(z)$ for $z \in \mathbb{R}$ is 0 if $z \leq 0$ and 1 otherwise. Define:

$$P(\alpha) = \mathbb{E}[h_\alpha(\boldsymbol{y})],$$

$$P_n(\alpha) = \frac{1}{n}\sum_{i=1}^n h_\alpha(\boldsymbol{y}^{(i)}).$$

A well-known result of VC-theory (Vapnik, 1999), specialized to linear separators, states that:

**Theorem I.1** (Vapnik 1999). *For any $t > \sqrt{2/n}$, it holds that:*

$$p\left(\sup_{\alpha \in \mathbb{R}^d} |P_n(\alpha) - P(\alpha)| \geq t\right) \leq 4\left(\frac{2en}{d+1}\right)^{(d+1)} \exp(-nt^2/8).$$

We continue with the assuming the following separability condition:

**Condition I.2** (separability of **y**). There exists $a > 0$ such that for any $\alpha \in \mathbb{R}^d$, $\alpha \neq 0$, $P(\alpha) > a$.

**Theorem I.2** (unboundedness of halfspace intersection). *Under Condition I.2 and Condition I.1, the halfspace intersection defined by $F$ for identifying $C_{j,:}$ is bounded with probability*

$$p\left(F \text{ defines unbounded intersection}\right) \leq 4\left(\frac{2en}{d+1}\right)^{(d+1)} \exp(-na^2/8)$$

*if $n > \max\{2/a^2, d\}$.*

*Proof.* For any $\alpha \in \mathbb{R}^d$, $\alpha \neq 0$, it holds that if $P_m(\alpha) = 0$ then for all $i \in [n]$, $\text{sgn}(\sum_{j=1}^d \alpha_j \boldsymbol{y}_j^{(i)}) = 0$. Each such case means that for this specific $\alpha$, we found a span of the columns of $F$ such that its $i$th coordinate is 0, and as such, a case in which $F$ is unbounded.

This means that if for all $\alpha \in \mathbb{R}^d$, it holds that $P_n(\alpha) > 0$, then the $n$ samples define a bounded halfspace intersection.

In addition, if $P_n(\alpha) = 0$, then

$$P_n(\alpha) - P(\alpha) = -P(\alpha), \tag{23}$$

therefore,

$$|P_n(\alpha) - P(\alpha)| = P(\alpha).$$

Since $a < P(\alpha)$, it holds that if Eq. (23) holds then:

$$|P_n(\alpha) - P(\alpha)| > a.$$

Therefore,

$$p\left(F \text{ defines unbounded intersection}\right) \leq p\left(\exists \alpha \neq 0, \ P_n(\alpha) = 0\right)$$
$$\leq p\left(\sup_{\alpha} |P_n(\alpha) - P(\alpha)| > a\right)$$
$$\leq 4\left(\frac{2en}{d+1}\right)^{(d+1)} \exp(-na^2/8)$$

where the last inequality is the result of Theorem I.1.

$\square$

Theorem I.2 gives a well-behaving sample complexity for the $m$ samples to define halfspace intersection that is bounded, when solving for $C_{j,:}$. Note that the theorem does not depend on $j$, because the use of $\mathbf{x}_j$ is not necessary for the boundedness result of Schmidt & Mattheiss (1977).

We now further show that *when* the halfspace intersection is bounded, then the diameter $r$ of the bounded space is smaller (where diameter is defined as the maximal distance between any two points).

The diameter is bounded by the maximal distance between all pairs of vertices of the intersection (the intersection is a bounded polytope). Let $E \subseteq \{v \mid Fv \leq -\mathbf{x}_j^{(:)}\}$ be the set of these vertices. The size $|E|$ is bounded from above by $O(n^{d/2})$ (by McMullen's Upper Bound theorem; Toth et al. 2017).

**Condition I.3** (equality saturation for vertices)**.** For every $v \in E$, there are $d$ equalities that are satisfied, in the form of $F_{I,:}v = -x_j^{(I)}$, such that $I \subset [n]$ and $|I| = d$.

**Condition I.4** (bounded input distribution)**.** For every $j \in [d]$, $|\boldsymbol{x}_j| \leq b$ for some $b > 0$.

**Lemma I.3** (Hoeffding's inequality)**.** *Let $Z_1, \ldots Z_n$ be $n$ random variables such that $Z_i \in [a_i, b_i]$ almost surely. Then, for all $t > 0$:*

$$p\left(\sum_{i=1}^n Z_i - \sum_{i=1}^n \mathbb{E}[Z_i] \geq t\right) \leq \exp\left(-\frac{2t^2}{\sum_{i=1}^n (b_i - a_i)^2}\right).$$

**Lemma I.4.** *Assume Condition I.1, Condition I.3 and Condition I.4 are satisfied.*

*In addition, let $\sigma^* = \mathbb{E}[\sigma]$ where $\sigma$ is the smallest singular value of $F_{I,:}$ for $I = \{1, \ldots, d\}$. Let $\varepsilon > \frac{\sqrt{db^2}}{\sigma^* \nu}$, where $0 \leq \nu = \frac{\log(\delta/4)}{\sigma^*} + 1 \leq 1$ for any $1 > \delta > 0$.*

*Define:*

$$L\left(\varepsilon, d, \sigma^*, b, L_j, \delta\right) = \sqrt[d]{\frac{\delta}{4D^2}} \exp\left(\frac{2((\sigma^*)^2 \nu^2 \varepsilon^2 - dL_j)^2}{d^2 b^2}\right).$$

*For any set of samples $(\mathbf{x}^{(i)}, \mathbf{y}^{(i)})$, $i \in [n]$ such that the halfspace intersection is bounded, it holds that with probability at least $1 - \delta > 0$ the diameter of the halfspace intersection is smaller than $\varepsilon$ if the following inequality is satisfied:*

$$n \geq L(\varepsilon, d, \sigma^*, b, \delta) \geq d.$$

*Proof.* The diameter of the intersection is not larger than the maximal distance between two pairs of vertices $||v - u||$, $u, v \in E$. By Condition I.3, let $I, J$ be the two subsets of integers such that $|I| = |J| = d$ and

$$F_{I,:}u = -\boldsymbol{x}_j^{(I)},$$
$$F_{J,:}v = -\boldsymbol{x}_j^{(J)}.$$

By the Haar assumption on $F$, the inverse of $F$ exists, and it holds that:

$$||u - v|| = ||F_{I,:}^{-1}\boldsymbol{x}_j^{(I)} - F_{J,:}^{-1}\boldsymbol{x}_j^{(J)}|| \leq ||F_{I,:}^{-1}||^* \cdot ||\boldsymbol{x}_j^{(I)}|| + ||F_{J,:}^{-1}||^* \cdot ||\boldsymbol{x}_j^{(J)}||.$$

Consider $||F_{I,:}^{-1}||^*$, the operator norm of $F_{I,:}^{-1}$ which equals $1/\sigma_I$ where $\sigma_I$ is the smallest singular value of $F_{I,:}$. Similarly, we define $\sigma_J$. Therefore:

$$||u - v|| \leq ||\boldsymbol{x}_j^{(I)}||/\sigma_I + ||\boldsymbol{x}_j^{(J)}||/\sigma_J.$$

The event that the diameter $r$ is larger than $\varepsilon$ can be bounded by ($A(I, J)$ is the event $\sigma_I \geq c, \sigma_J \geq c$ for some $c$):

$$p\left(r \geq \varepsilon\right) \leq p\left(r \geq \varepsilon, A(I, J)\right) + p\left(\sigma_I \leq c \wedge \sigma_J \leq c\right).$$

In addition,

$$
\begin{aligned}
p\left(r \geq \varepsilon, A(I, J)\right) &\leq p\left(\max_{u,v \in E} ||u - v|| \geq \varepsilon, A(I, J)\right) \\
&\leq p\left(\max_{u,v} ||\boldsymbol{x}_j^{(I)}||/c + ||\boldsymbol{x}_j^{(J)}||/c \geq \varepsilon\right) \\
&\leq \sum_{u,v \in E} p\left(||\boldsymbol{x}_j^{(I)}||/c + ||\boldsymbol{x}_j^{(J)}||/c \geq \varepsilon\right),
\end{aligned}
$$

We know that $|E| \leq D(n^{d/2})$ for some constant $D$ (Toth et al., 2017), therefore $p\left(r \geq \varepsilon, A(I, J)\right)$ can be further bounded by

$$D^2 n^d \cdot p\left(||\boldsymbol{x}_j^{(I)}||/c + ||\boldsymbol{x}_j^{(J)}||/c \geq \varepsilon\right).$$

Since the distribution of $\boldsymbol{x}_j^{(I)}$ and $\boldsymbol{x}_j^{(J)}$ (and the singular values) is identical, $p\left(r \geq \varepsilon, A(I, J)\right)$ can be further bounded by

$$2D^2 n^d \cdot p\left(||\boldsymbol{x}_j^{(I)}||/c \geq \varepsilon\right), \tag{24}$$

for any $I$, for example $I = \{1, \ldots, d\}$.

We can use any value for $c = \mathbb{E}[\sigma_I] \cdot \nu$ to be smaller than $\sigma_I$, and the inequality still holds. Therefore, Eq. (24) can be further bounded by (assuming $\nu < 1$):

$$2D^2 n^d \cdot \left(p\left(||\boldsymbol{x}_j^{(I)}||/\mathbb{E}[\sigma_I]\nu \geq \varepsilon\right)\right).$$

This can be further bounded by:

$$2D^2 n^d \cdot \left(p\left(||\boldsymbol{x}_j^{(I)}||^2 \geq \mathbb{E}[\sigma_I]^2 \nu^2 \varepsilon^2\right)\right), \tag{25}$$

where we also square the norm of $\boldsymbol{x}_j^{(I)}$ (and its bounding term).

The term $p\left(\sigma_I \leq \mathbb{E}[\sigma_I] \cdot \nu \wedge \sigma_J \leq \mathbb{E}[\sigma_J] \cdot \nu\right)$ for small $\nu$ is going to be close to 0, more specifically, it measures the probability of the complement of the event $A(I, J)$. This probability is at most twice the probability that $\exp(-\sigma_I) \geq \exp(-\nu \cdot \mathbb{E}[\sigma_I])$, which by Markov's inequality is smaller than $\mathbb{E}\left[\exp(-\sigma_I)\right] / \exp(-\nu \cdot \mathbb{E}[\sigma_I]) \leq \exp(\mathbb{E}\left[(\nu - 1) \cdot \sigma_I\right]) = \exp((\nu - 1)\sigma^*)$ (by Jensen's inequality). Therefore, the probability of the complement of $A(I, J)$ is bounded by:

$$p\left(\sigma_I \leq \nu\sigma^* \wedge \sigma_J \leq \sigma^*\right) \leq 2\exp((\nu - 1) \cdot \sigma^*). \tag{26}$$

Consider that $||\boldsymbol{x}_j^{(I)}||^2$ is the sum of $d$ iid samples $\boldsymbol{x}_j^2$. Under Condition I.4, we assume $|\boldsymbol{x}_j^2| \leq b$. Therefore, by Hoeffding's inequality, we can get an upper bound on the probability term in Eq. (25) to hold:

$$p\left(||\boldsymbol{x}_j^{(I)}||^2 - d \cdot \mathbb{E}[\boldsymbol{x}_j^2] \geq (\sigma^*)^2 \nu^2 \varepsilon^2 - d \cdot \mathbb{E}[\boldsymbol{x}_j^2]\right) \leq \exp\left(\frac{-2((\sigma^*)^2 \nu^2 \varepsilon^2 - d \cdot \mathbb{E}[\boldsymbol{x}_j^2])^2}{db^2}\right), \tag{27}$$

remembering it is stated in the theorem that $\sigma^* \nu \varepsilon \geq \sqrt{db^2}$ and the fact that $\sqrt{db^2} \geq \sqrt{d \cdot \mathbb{E}[\boldsymbol{x}_j^2]}$ by Condition I.4. Hence, $\mathbb{E}[\sigma_I]^2 \nu^2 \varepsilon^2 - d \cdot \mathbb{E}[\boldsymbol{x}_j^2] > 0$, and the use of Hoeffding's inequality is allowed.

Let $\delta > 0$, then taking Eq. (27) with the union bound constant from $E$ and setting it to $\delta/2$:

$$2D^2 n^d \exp\left(\frac{-2((\sigma^*)^2 \nu^2 \varepsilon^2 - dL_j)^2}{db^2}\right) \leq \delta/2$$

We can now get an upper bound on $n$:

$$n \leq \sqrt[d]{\frac{\delta}{4D^2}} \exp\left(\frac{2((\sigma^*)^2 \nu^2 \varepsilon^2 - dL_j)^2}{d^2 b^2}\right) = L(\varepsilon, d, \sigma^*, b, L_j, \delta).$$

We want, in addition, the complement of $A(I, J)$ to have probability at most $\delta/2$, so therefore, we choose $\nu = \frac{\log(\delta/4)}{\sigma^*} + 1$ according to Eq. (26).

Noting that the diameter of the sphere gets smaller as $n$ increases (because there are more halfspaces that possibly intersect into a smaller space).

$\square$

Note that Lemma I.4 requires $\delta \geq \exp(-\sigma^*)$ in its statement (this can be inferred from the relationship between $\nu$ and $\delta$). To have an arbitrary confidence $1 - \delta$ in the diameter being smaller than $\epsilon$, we can run the algorithm $k$ times with an $n$ as needed. The probability of all of them having diameter larger than $\varepsilon$ is smaller than $\delta^k$.

Lemma I.4 describes under which condition we get a small error for $C_{j,:}$ if we fix $j$. Using a union bound, we bound the probability that for any $j \in [d]$, the corresponding feasible halfspace intersection space is small. This will require adding a factor of $d$ to $\delta$, and a factor of $d$ to $n$. We overcome this by making sure that the probability for each $j$ having a diameter larger than $\epsilon$ is smaller than $\delta/d$, and we use $dn$ samples for a choice of $n$ satisfied by Lemma I.4. Using a union bound, we can show that the probability that for *any* $j$ the diameter is larger than $\varepsilon$ is smaller than $\delta$.

In addition, note that Lemma I.4 works only when we assume the halfspace intersection is bounded. This is the case where we can use the diameter bound through the set of vertices. To take this into account, we also use Theorem I.2. This leads to the following result regarding the sample complexity of our algorithm.

**Theorem I.5** (sample complexity of learning ReLU two-layered networks (layer 2)). *Assume Condition I.1, Condition I.2, Condition I.3 and Condition I.4 are satisfied.*

*In addition, let $\sigma^* = \mathbb{E}[\sigma]$ where $\sigma$ is the smallest singular value of $F_{I,:}$ for $I = \{1, \ldots, d\}$. Let $\varepsilon > \frac{\sqrt{db^2}}{\sigma^* \nu}$, where $0 \leq \nu = \frac{\log(\delta/8d)}{\sigma^*} + 1 \leq 1$ for any $1 > \frac{\delta}{2d} > 0$.*

*Define:*

$$L^*(\varepsilon, d, \sigma^*, b, L_j, \delta) = \min_j \sqrt[d]{\frac{\delta}{8dD^2}} \exp\left(\frac{2((\sigma^*)^2 \nu^2 \varepsilon^2 - dL_j)^2}{d^2 b^2}\right).$$

*and define:*

$$L_u(a, \delta, d) = \frac{(d+1)\log 6 + \log 4 - \log \dfrac{\delta}{2}}{a^2/8 - 1/e}.$$

*For any set of samples $(\mathbf{x}^{(i)}, \mathbf{y}^{(i)})$, $i \in [n]$, it holds that with probability at least $1 - \delta > 0$ the diameter of the halfspace intersection is smaller than $\varepsilon$ if the following inequalities hold:*

$$n \geq d \cdot L^*(\varepsilon, d, \sigma^*, b, \delta),$$
$$L^*(\varepsilon, d, \sigma^*, b, \delta) \geq \max\left\{d, L_u(a, \delta, d), 2/a^2\right\}.$$

*Proof.* Let $U$ be the event that the diameter is larger than $\varepsilon$ for any $j \in [d]$. Let $V$ be the event that $F$ represents bounded space. The event $U$ is the one we need to show has small probability.

By Theorem I.2, we know that if $n \geq L_u(a, \delta, d)$ then $F$ is unbounded with probability smaller than $\delta/2$. Therefore, $p(V) \geq 1 - \delta/2$.

From Lemma I.4 and the analysis before this theorem statement, we know that for $n$ as required by this theorem statement, any time $F$ represents an bounded half-space intersection, the probability of the diameter this intersection (for any $j$) being larger than $\epsilon$ is smaller than $\delta/2$. Therefore, $p(U \mid V) \leq \delta/2$.

Hence,

$$p(U) = p(U \mid V) p(V) + p(U \mid \neg V)(1 - p(V)) \leq p(U \mid V) + (1 - p(V)) \leq \delta/2 + \delta/2 = \delta.$$

$\square$

Table 5: **Prediction errors (RMSE)** as $p$ (the fraction of nonnegative entries in $\mathbf{A}$) increases ($n = 512$, $d = 16$). Each set of rows describes an experiment with a different number of scale vector rows in $\mathbf{A}$ ($s$).

| $s$ | Alg. | $p =0$ | 0.1 | 0.2 | 0.3 | 0.4 | 0.5 | 0.6 | 0.7 | 0.8 | 0.9 | 1.0 |
|---|---|---|---|---|---|---|---|---|---|---|---|---|
| | SGD | 0.4353 | 0.4135 | 0.3844 | 0.3511 | 0.3364 | 0.3255 | 0.3310 | 0.3480 | 0.3792 | 0.4158 | 0.4417 |
| 0 | LP | 0.0923 | 0.0991 | 5.2210 | 18.3364 | $\infty$ | $\infty$ | $\infty$ | $\infty$ | $\infty$ | $\infty$ | 55.4519 |
| | QP | 0.0492 | 0.0583 | 0.5445 | 10.0328 | $\infty$ | $\infty$ | $\infty$ | $\infty$ | $\infty$ | $\infty$ | 0.4129 |
| 2 | SGD | 0.4252 | 0.4058 | 0.3777 | 0.3510 | 0.3315 | 0.3216 | 0.3293 | 0.3447 | 0.3704 | 0.4060 | 0.4234 |
| | QP | 0.0729 | 0.0819 | 0.5072 | 7.8984 | 9.8862 | $\infty$ | $\infty$ | $\infty$ | $\infty$ | 5.8306 | 0.4297 |
| 4 | SGD | 0.4152 | 0.3962 | 0.3711 | 0.3423 | 0.3286 | 0.3213 | 0.3228 | 0.3382 | 0.3683 | 0.3944 | 0.4118 |
| | QP | 0.0926 | 0.1070 | 0.4480 | 5.6844 | $\infty$ | $\infty$ | $\infty$ | $\infty$ | $\infty$ | 12.0391 | $\infty$ |
| 8 | SGD | 0.3941 | 0.3823 | 0.3637 | 0.3410 | 0.3289 | 0.3248 | 0.3256 | 0.3428 | 0.3578 | 0.3743 | 0.3874 |
| | QP | 0.1221 | 0.1527 | 0.2835 | 1.3792 | 8.4902 | 7.2903 | 7.4253 | $\infty$ | 5.9887 | 8.2727 | 0.5928 |
| 16 | SGD | 0.3820 | 0.3829 | 0.3827 | 0.3812 | 0.3825 | 0.3825 | 0.3827 | 0.3815 | 0.3831 | 0.3818 | 0.3827 |
| | QP | 0.1810 | 0.1841 | 0.1853 | 0.1823 | 0.1833 | 0.1845 | 0.1823 | 0.1823 | 0.1825 | 0.1840 | 0.1803 |

## J   Nonnegativity of $\mathbf{A}$ and the Scale Transformation Condition

Throughout the paper, we assume that the first layer, the matrix $\mathbf{A}$, has nonnegative entries. As we mention before in Section 8 and Section 7.2, such restrictions on neural network weights have successfully been studied before and shown to be effective in practice.

While the theory of our algorithm does require $\mathbf{A}$ to be nonnegative, we use this condition in a specific part of the proof of Theorem E.1 in Appendix F.1. This theorem is only necessary to prove that the objective functional for layer 2 achieves the correct estimate of the network. It is not necessary for proofs for layer 1 or for the proofs of the transition from the layer objectives into the QP or LP formulation. In that sense, we can aim to alleviate this nonnegativity condition quite in a modular way when addressing this issue. For example, we could parametrize the number of negative entries in $\mathbf{A}$ and make more nuanced arguments (albeit more complex) in the proofs in Appendix F.1.

This condition does not prevent us from running the algorithm as-is with negative entries in $\mathbf{A}$. To test our algorithm sensitivity to this assumption, we repeated the experiment in Section 7.1, with $d = 16$ and

$n = 512$, and checked what error the algorithm gives as a function of $p \in \{k/10 \mid k \in \{0, \ldots, 10\}\}$ – a fraction of random entries that are set to be negative in $\mathbf{A}$. To test the effect of Condition 2.1 (which is alleviated in Appendix E), we also vary the number of rows that are a scaling vector (through parameter $s \in \{0, 2, 4, 8, 16\}$, which denotes the number of such rows in $\mathbf{A}$). We repeated each experiment five times with different parameters (and each experiment includes multiple executions with different samples). We experimented both with the LP and the QP objectives for $s = 0$ and the QP objective for $s > 0$, and compared them against the SGD algorithm.

The results are given in Table 5. When the assumption about nonnegative entries holds, both the QP algorithm and the LP algorithm (for $s = 0$) give an error much lower than SGD. This holds even for a small $p > 0$. However, as $p$ increases, the results of LP and QP degrade quickly and then recover (for the QP) when $p = 1.0$. This latter result could be due to the symmetry of our input distribution. we also note that in general, the QP algorithm, as expected, is more robust to $p > 0$ than the LP algorithm. As we increase the number of rows in $\mathbf{A}$ that are scale vectors, we see that our algorithm becomes much more robust to the number of negative entries in the matrix $\mathbf{A}$ (with respect to $\infty$ entries). However, having scale vectors in $\mathbf{A}$ does seem to make the learning problem more difficult for the QP algorithm.

