# OpenReview forum: "Nonparametric Learning of Two-Layer ReLU Residual Units"
_TMLR — Accepted by TMLR_

### Review · Reviewer_adXy · 2022-07-22

**Summary Of Contributions:**

In this paper, the authors present an algorithm (Algorithm 1+2) that learns a two-layer residual unit/network with ReLU activation in a teacher-student setup, under the assumption that the first-layer weights are nonnegative (without specific assumption on the data distribution), and shows that the student network is able to recover, with the proposed layer-by-layer approach, the teacher parameters in the infinite data limit (Theorem 6.3).
Empirical results are provided in Section 7 to further elaborate the proposed method on both synthetic data (in Section 7.1) and real-world data (in Section 7.2).

**Broader Impact Concerns:**

The authors discussed the limitation of this work and their thoughts on future work.
It would be great if the authors could discuss more on how to extend the proposed approach to generic and more involved neural nets with, e.g., not necessarily nonnegative weights, other activation rather than ReLU, more layers, etc.

**Requested Changes:**

* Abstract: should one understand "nonnegative" as nonnegative definite for symmetric matrices or matrices having non-negative entries? This remains unclear, at least at this point of the paper.
* Section 1.2 needs some reworking: I understand, in the case of some ML conferences, the use of informal theorems to allow the readers to have a first grasp of the main results and ideas of the paper. But I am not so sure that this is appropriate in a 12-page long journal paper and, even if this is the case, the current versions of Theorem 1.1 and 1.2 are not easy to understand.
   * Theorem 1.1 needs some more explanations, for example on "quadratic functionals defined in linear-constrained domains" and how they correspond to the (teacher or student) NN model defined above, and on the precise meaning of "can be adjusted to." A few clarifying sentences should be enough. Also, the paragraph after Theorem 1.1 is on the computation of the moments, but the motivation is not clear at all: the only available information at this point is "The values of the objectives are moments over the input distribution," but it remains unclear if the estimation (from the available samples) of these moments are difficult or not.
   * What does "that learns a network $\overset{a.s.}{\to}$ the exact ground-truth" mean? Should we take this as "recover the ground-truth network parameters"? in which metric sense? and the almost sure convergence is with respect to what? the input data distribution I imagine?
   * More generally speaking, the statements in Theorem 1.1 and 1.2 are very unclear and vague. Even worse, it is not easy to find the corresponding "formal" version of them (since they are announced as "informal") in the remainder of the paper. I can understand that for conference papers, this allows readers to have a quick grasp of the main contribution of this work, but I am not sure if this is a good idea for a (more lengthy) journal paper here.
* In the paragraph "Roadmap", please first define and then use LR.
* "further simplified as an LP that has the same capability as the QP to learn layer 2": what does "same capacity" mean here?
* after Theorem 4.1: "to prove a more general theoremwhich"
* Above Section 4.3 "and solving layer 2 QP are empirically equivalent": please clarify the precise meaning of "empirically equivalent."
* I find the discussion in Sec 4.3 on the LP simplification somewhat unclear: what do the authors mean by "LP and QP" have equivalent solution space? Cannot one say anything about the solutions achieved by these two programs? Are they the same, or are they different? Can any formal statement be announced here?
* Could the authors comment on the limitation of this work by focusing on the *non-negativity* of the first layer weight? Is this assumption intrinsic? Can it be relaxed with some additional effort? Or from a different perspective, does this assumption correspond to a practical scenario or application?


**Strengths And Weaknesses:**

The paper focuses on the important problem of learning a single-hidden-layer (thus non-convex) neural network having ReLU activation and a residual unit, by providing an efficient algorithm with theoretical guarantees (in the sense that one is able to recover the ground-truth parameters in a teacher-student setup).
From a technical perspective, this paper contains some interesting ideas that are, I would say, inspired by previous efforts but well adapted to the problem under study.
My major concern is the presentation of this paper: as a theoretical paper, it contains many vague and unclear statements (specifically in Section 1.2) that need to be clarified.
I believe that this paper needs to be carefully revised by adding clarifications (in words and also possibly in figures/tables, etc.) to help the interested readers (myself being one of them) better digest the results.
See my detailed comments below.

---

> ### Author Response · Authors · 2022-08-05
> **response to review**
>
> Thank you for your comments, they are really helpful, and we appreciate the quick review!
>
> We include a response below, quoting parts of your review. We will be happy to follow with further clarifications.
>
> > Abstract: should one understand "nonnegative" as nonnegative definite for symmetric matrices or matrices having non-negative entries? This remains unclear, at least at this point of the paper.
>
> We changed that to refer to "nonnegative entries".
>
> > Section 1.2 needs some reworking: I understand, in the case of some ML conferences, the use of informal theorems to allow the readers to have a first grasp of the main results and ideas of the paper. But I am not so sure that this is appropriate in a 12-page long journal paper and, even if this is the case, the current versions of Theorem 1.1 and 1.2 are not easy to understand.
>
> We have significantly re-organized section 1.2 to follow up on these comments.
>
> > Theorem 1.1 needs some more explanations, for example on "quadratic functionals defined in linear-constrained domains" and how they correspond to the (teacher or student) NN model defined above,
>
> We now refer to Eq. (2) and (6) to explain what we mean by quadratic functions. Basically, these are functionals that accept an activation function, and the relevant layer parameters, and
>
> The mentioned h's domain, nonnegative continuous function space $R \rightarrow R_{\ge 0}$, is linear bounded, e.g. when d = 1, the function space is $R \rightarrow R_{\ge 0}$ which in x-y plane is an area bounded by the x-axis. We corrected the notation to $R_{\ge 0}$ in Section 4.1 line 7. This leads to a corresponding linear inequalities in the empirical version of the objective function.
>
> > and on the precise meaning of "can be adjusted to."
> > A few clarifying sentences should be enough.
>
> We explain what we mean by the adjustment and refer to the proper part in the paper ("... but can be adjusted to the exact ground-truth by a process of rescaling (see Theorem 5.2).")
>
> > Also, the paragraph after Theorem 1.1 is on the computation of the moments, but the motivation is not clear at all:
> > the only available information at this point is "The values of the objectives are moments over the input distribution," but it remains unclear if the estimation
> > (from the available samples) of these moments are difficult or not.
>
> Our algorithm leads to an estimation that is not difficult from the samples. It requires solving two quadratic programs, where the quadratic programs use moments from the samples. We clarify this a bit better now and refer to the relevant places in the paper.
>
> > What does "that learns a network the exact ground-truth" mean? Should we take this as "recover the ground-truth network parameters"? in which metric sense? and the almost sure convergence is with respect to what? the input data distribution I imagine?
>
> The metric we consider here is Frobenius norm of the difference between the true parameters and the learned parameters. THe almost sure convergence is indeed with respect to the input data distribution (over n samples). We clarify this now.
>
> > More generally speaking, the statements in Theorem 1.1 and 1.2 are very unclear and vague. Even worse, it is not easy to find the corresponding "formal" version of them (since they are announced as "informal") in the remainder of the paper.
> > I can understand that for conference papers, this allows readers to have a quick grasp of the main contribution of this work, but I am not sure if this is a good idea for a (more lengthy) journal paper here.
>
> We now refer to the formal theorems in the paper when we give the high-level roadmap.
>
> > In the paragraph "Roadmap", please first define and then use LR.
>
> LR refers to Linear Regression, we now explain that.
>
> > "further simplified as an LP that has the same capability as the QP to learn layer 2": what does "same capacity" mean here?
>
> We rephrased that. The idea is that in the noiseless case, the QP for layer 2 can be further simplified to a linear program that has the same feasible region and leads to the same solution.
>
> The LP can recover the network parameters to the same level of precision as the QP when learning from noiseless examples.
>
>
> > after Theorem 4.1: "to prove a more general theoremwhich"
>
> We fixed that.
>
> > Above Section 4.3 "and solving layer 2 QP are empirically equivalent": please clarify the precise meaning of "empirically equivalent."
>
> We rephrased that, again, we are referring to programs that with the same set of samples, have the same feasible region and solution.
>
> > I find the discussion in Sec 4.3 on the LP simplification somewhat unclear: what do the authors mean by "LP and QP" have equivalent solution space? Cannot one say anything about the solutions achieved by these two programs? Are they the same, or are they different? Can any formal statement be announced here?

---

> > ### Author Response · Authors · 2022-08-05
> > **and in addition**
> >
> > We just mean that the QP can have a reformulation as an LP, and these two have equivalent feasible regions / solutions (in the noiseless case, when the samples are generated from the underlying model for y).
> >
> > >Could the authors comment on the limitation of this work by focusing on the non-negativity of the first layer weight? Is this assumption intrinsic? Can it be relaxed with some additional effort? Or from a different perspective, does this assumption correspond to a practical scenario or application?
> >
> > We now include a short discussion in section 8. We claim that there is quite a bit of work that previously also chose to make the weights nonnegative, perhaps most prominently, the work about Boolean neural networks.
> >
> > > It would be great if the authors could discuss more on how to extend the proposed approach to generic and more involved neural nets with, e.g., not necessarily nonnegative weights, other activation rather than ReLU, more layers, etc.
> >
> > We have already included a discussion of the use of multiple layers in Appendix C in the original submission. In regards to using activations other than ReLU, we can make the functional h which we optimize over be of a different type than the one we currently use that leads to ReLU activation. We will discuss this further within a new page limit.

---

### Review · Reviewer_Rwh4 · 2022-08-08

**Summary Of Contributions:**

The paper proposes an algorithm which learns a 2-layered residual network with ReLU activation. The algorithm works in two steps, first by learning the weights of the second layer and then those of the first layer. The paper proves that this algorithm recovers the ground truth network (the data is assumed to be generated from a 2 layer residual network), if the ground truth network has first layer which has non-negative weights. The algorithm's run time is polynomial, and seems to outperform SGD on synthetic and small real datasets.

**Broader Impact Concerns:**

None.

**Requested Changes:**

Critical:
1. To verify the claim that first layer being all non-negative is indeed not restrictive, please re-run the experiments on synthetic dataset, but where the ground truth comes from a residual network with negative values allowed in first layer (use Gaussian instead of folded Gaussian). Then, please report the comparison of output error for your algorithm vs SGD. Note that even if the proposed algorithm performs much worse than SGD, the comparison should be included, so that the readers can have a better understanding of the algorithm's performance.

Non-critical:

2. In the experiments measuring output error, was SGD constrained to have non-negative first layer?
3. It would be nice to have a discussion in the main paper why the requirement of A* being non-negative is needed, and if the authors have any ideas about how it can be bypassed by future work.
4. Can the idea of learning the last layer and then the ones before it layer-wise be extended to deeper networks?

Minor:
5. This seems to be an incomplete sentence on Page 5:
     "If we parameterize h, and show that the nonlinearity w.r.t. its parameters would make Gˆ2 lose its quadratic form."

**Strengths And Weaknesses:**

Strengths:
1. The proposed algorithm is novel, and the paper proves that it can indeed recover ground truth network under certain assumptions. Further, the algorithm runs in polynomial time.
2. Parts of the proof techniques can be of general interest. In particular, I liked the idea of learning the network layer by layer, by formulating the problem of learning the second layer as minimization problem with constraints, and then reducing the problem to that of learning a single (first) layer.
3. The paper is well written, easy to understand. The concepts and proof ideas are explained really well.

Weaknesses:
1. The major weakness in my opinion, which the paper also acknowledges, is that the weight matrix of the first layer of the ground truth network needs to have all positive entries. The paper says that such networks are still quite expressive. However, I think in high dimensions, such matrices would have an exponentially small measure (for example if we consider the operator norm as the distance metric in this matrix space), although I could be wrong. This is certainly true if we consider the measure of vectors with all positive entries in high dimensions.
2. The time complexity of the algorithm seems to be d^6 and n^3 where d is the input dimension and n is the dataset size. This is prohibitively large for modern ML tasks.
3. The consistency results in Section 6 are asymptotic. It would be better to have non-asymptotic results (maybe by using some distributional assumptions on inputs).
4. The networks considered do not have any biases. Although, this might not be a big problem to fix, maybe the authors can comment on how biases can be included in the networks presented in this paper?

---

> ### Author Response · Authors · 2022-08-15
> **response to review**
>
> Thank you for your comments, they are really helpful, and we appreciate the quick review!
>
> We include a response below, quoting parts of your review. We will be happy to follow with further clarifications.
>
> > The time complexity of the algorithm seems to be d^6 and n^3 where d is the input dimension and n is the dataset size. This is prohibitively large for modern ML tasks.
>
> This is indeed the case that the worst-case complexity with standard solvers is as you state. This is a worst-case analysis, though. In practice, modern solvers can quite effectively solve large problems using heuristics and state-of-the-art algorithms.
>
> > The consistency results in Section 6 are asymptotic. It would be better to have non-asymptotic results (maybe by using some distributional assumptions on inputs).
>
> Indeed the consistency results are asymptotic, but we confirm throughout our experiments that our algorithm is highly efficient for the number of samples required. In addition, we include sample complexity results in Appendix I.
>
> > The networks considered do not have any biases. Although, this might not be a big problem to fix, maybe the authors can comment on how biases can be included in the networks presented in this paper?
>
> In practice, bias terms can be accommodated by using a dimension in x with a constant value.
>
> > Requested change, critical: To verify the claim that first layer being all non-negative is indeed not restrictive, please re-run the experiments on synthetic dataset, but where the ground truth comes from a residual network with negative values allowed in first layer (use Gaussian instead of folded Gaussian). Then, please report the comparison of output error for your algorithm vs SGD. Note that even if the proposed algorithm performs much worse than SGD, the comparison should be included, so that the readers can have a better understanding of the algorithm's performance.
>
> We include a short discussion in the conclusion section regarding the nonnegativity entries of A. Such neural networks have been widely (and successfully) studied.
>
> In addition, we now include experiments with negative entries (in A) in Appendix J. Our findings are that indeed the algorithm is sensitive to having a significant portion of the entries of A as negative. However, it can still cope well with a small number of them. Thank you for this suggestion, it was valuable for us.
>
> > In the experiments measuring output error, was SGD constrained to have non-negative first layer?
>
> SGD was not constrained to have nonnegative layer.
>
> > It would be nice to have a discussion in the main paper why the requirement of A* being non-negative is needed, and if the authors have any ideas about how it can be bypassed by future work.
>
> The reason is that a particular chain of inequalities that the proofs follow (Theorem 4.1, for example) requires A to be nonnegative so that the inequalities follow. It *might* be possible to overcome this by parameterizing the number of nonnegative entries. We may then break these inequalities based on this parameter, but it would make things highly cumbersome.
>
> >  Can the idea of learning the last layer and then the ones before it layer-wise be extended to deeper networks?
>
> We discuss this issue in Appendix C.
>
> > Minor: 5. This seems to be an incomplete sentence on Page 5: "If we parameterize h, and show that the nonlinearity w.r.t. its parameters would make Gˆ2 lose its quadratic form."
>
> We fixed that sentence: "If we parameterize $h$, such nonlinearity w.r.t. its parameters would make $\hat{G}_2$ lose its quadratic form."

---

> > ### Author Response · Authors · 2022-08-20
> > **an additional comment regarding the nonnegativity condition**
> >
> > We wanted to note to the reviewer that we have reconsidered our answer about the nonnegativity again, especially the request to try and pinpoint why it is needed, and added the following note to Appendix J:
> >
> > While the theory of our algorithm does require $A$ to be nonnegative, we use this condition in a specific part of the proof of Theorem E.1 in Appendix F.1. This theorem is only necessary to prove that the objective functional for layer 2 achieves the correct estimate of the network. It is not necessary for proofs for layer 1 or for proofs of the transition from the layer objectives into the QP or LP formulation. In that sense, we can aim to alleviate this nonnegativity condition quite in a modular way when addressing this issue. For example, we could parametrize the number of negative entries in $A$ and make more nuanced arguments (albeit more complex) in the proofs in Appendix F.1.

---

### Review · Reviewer_XZBi · 2022-09-12

**Summary Of Contributions:**

This paper studies the problem of learning a two-hidden layer relu network.  The authors propose an algorithm based on QP/LP and the basic idea is that they first learn the second layer and then using the new information they learn the first layer. They show that the algorithm is consistent and provide a comparison with linear regression. They work in the realizable case and they assume non-negative entries in matrix A and they also require the matrix not to be a scale transformation, ( Definition 2.1), which is required so there is a unique solution.



**Broader Impact Concerns:**

no ethical concerns.

**Requested Changes:**

See above. 1,2,3,4. I suggest putting some effort and improve the clarity of this work and the presentation. The paper will be improved if the authors explain their proofs and provide details on why each lemma is required.


**Strengths And Weaknesses:**

- Strengths: This is an important problem and the algorithm is simple. The authors provide experiments for their results.
- Weaknesses: The main weakness is that this is a very restrictive setting. Are these conditions required? The answer is: It depends. If you want to explicitly learn the parameters then you required to have a unique solution, in the literature usually we required assumptions in the condition number of $A$. For learning, it is well known that a uniqueness of the solution is not required. Now the assumption that the matrix is non-negative is a bit of strange and I do not believe it is an interesting setting.
In my opinion this work has several problems that the authors need to address.
1. The fact that the entries of A are non-negative is hidden and need to be explicitly stated in the Condition 2.1.
2. Many statements are not well-supported. For example: Appendix A. It does not make sense to have a section for a different model without any proofs.
3. Vanilla Linear Regression. I do not see the reason to compare with an algorithm that does not really work. No ones expects to draw samples from the space X>0. Even, for gaussian distribution, this area has tiny probability.
4. Appendix I is a mess. I suggest the authors to put somewhere a theorem with the sample complexity and put more explanations.

Overall, this work may be interested for some audience but the authors should fix some of the issues raised above.

---

> ### Author Response · Authors · 2022-09-13
> **response to review**
>
>
> We thank the reviewer for their comments. Below is our response.
>
> > They show that the algorithm is consistent and provide a comparison with linear regression.
>
> To be more specific, the algorithm is not only consistent, but strongly consistent w.r.t. any 2nd-order moment finite input distribution.
>
> We do *not* use linear regression as a baseline, but rather to exemplify the non-triviality of the problem and our algorithm as vanilla LR has solid exponential sample complexity in this case.
>
> > they also require the matrix not to be a scale transformation, (Definition 2.1), which is required so there is a unique solution.
>
> No, we do *not* require the matrix not to be a scale transformation. We add this condition along with B* being square as Condition 2.1, and use it for simplicity of exposition in the main body of the paper, and for an easier grasp in a first read. But it is not a strict requirement. Condition 2.1 is *completely* eliminated later and we discussed the generalization *without* Condition 2.1 explicitly in Appendix E: *Neither* A* being scale transformation *nor* B* being square is needed.
>
> > Strengths: This is an important problem and the algorithm is simple. The authors provide experiments for their results.
>
> To be more specific, our algorithm leverages nonparametric estimation and QP/LP-fication to the objectives over input distribution and follow-up adjustments to the solutions that both layers can be learned exactly through this pattern. We provide experimental results on *both* synthetic and real-world data and justify our results experimentally.
>
> But we would like to highlight that, more importantly and mainly for this paper, we provide the theory underlying our algorithm that shows specific guarantees on our objectives, algorithm consistency and sample complexity to estimate the networks with relatively mild conditions.
>
> > If you want to explicitly learn the parameters then you required to have a unique solution.
> > For learning, it is well known that a uniqueness of the solution is not required.
>
> We do not require a unique solution. Here is the path of logic in our paper: We have Condition 2.1 to make the solution to our objectives unique, so that the problem is simplified to be discussed in scope of the main content of the paper. In Appendix E we completely eliminated Condition 2.1. This is mentioned above the definition of the condition and in footnote 3.
>
> > in the literature usually we required assumptions in the condition number of A.
>
> 1. Strong consistency does not require a condition number assumption because it studies asymptotic properties based on almost sure convergence.
>
> 2. In Appendix I, we include $\sigma^*$ (a singular value of a matrix representing the samples) which represents the sensitivity information of A*, B* and the input distribution.
>
> 3. We indeed take care of the condition number, as we estimate B* by estimating its inverse. We present experimental results of B* condition number robustness in Section 7.1 and showed that our algorithm is hardly affected by the condition number.
>
> > Now the assumption that the matrix is non-negative is a bit of strange and I do not believe it is an interesting setting.
> > The fact that the entries of A are non-negative is hidden
>
> The nonnegativity constraint is mentioned quite early and explicitly in the abstract and throughout the paper. A significant body of work discusses nonnegative weights. We added a discussion of this at the end of the conclusion with some references.
>
> > and need to be explicitly stated in the Condition 2.1.
>
> As we mentioned, Condition 2.1 is just used in the main body of the text. It is not required for the correctness of the algorithm.
>
> We added another mention of this visibly below Condition 2.1, as suggested (we note that we eliminate Condition 2.1 later, so we cannot include this as part of the condition itself).
>
> > Many statements are not well-supported.
>
> We would be happy to know which statements are not well-supported. Regarding the example that you gave:
>
> > For example: Appendix A. It does not make sense to have a section for a different model without any proofs.
>
> Along with the multi-layer extension Appendix C, Appendix A is an initiation and a prototype of an extension for this work (future work). The scope of a paper is wide but also limited to a single paper. We did intend for others to follow up with these directions.
>
> > Vanilla Linear Regression. I do not see the reason to compare with an algorithm that does not really work. No ones expects to draw samples from the space X>0. Even, for gaussian distribution, this area has tiny probability.
>
> The vanilla linear regression example is there to demonstrate precisely the issue there is with standard learning techniques to estimate a ReLU network. It is not there as a practical algorithm. We can omit it if needed.

---

> > ### Author Response · Authors · 2022-09-13
> > **... in continuation to the commments ...**
> >
> >
> > > Appendix I is a mess. I suggest the authors to put somewhere a theorem with the sample complexity and put more explanations.
> >
> > Our sample complexity results follow several intermediate results. We first demonstrate that the constraints that define our optimization objective give a bounded polytope (Thm I.2). This happens with a high probability, given a set of samples. Once we have shown that, we then show that the polytope constraints define a polytope with a small diameter with a high probability given a sufficient number of samples (Thm I.5). The boundnesses of the constraint polytope is required there.
> >
> > We added a roadmap with this explanation to Appendix I and also clarified some things there.

---

### Decision · Action_Editors · 2022-11-13

**Recommendation:** Accept with minor revision

**Comment:**

The reviewers are all in favour of accepting the paper and to the extent that the paper is sound and adequately presents the full mathematical derivations and description of experimental results, it should be accepted. The authors should follow-up on the comments made by the reviewers and add a fuller description of the positivity constraint on A in particular, and also discuss the new experimental results about what happens when the conditions don't hold.

**Audience:**

This is primarily likely to be of interest to a theoretical ML audience. It is possible that the algorithms may lead to improved practical algorithms, but for now the restrictions e.g. positivity, seem to suggest that the result is primarily of theoretical interest.

**Claims And Evidence:**

The claims made in the paper are adequately proved. The mathematical results have proofs in the paper, which the reviewers have read and no concerns have been raised. The experimental results are also adequately explained.

---

> ### Author Response · Authors · 2022-11-23
> **Camera-ready submitted**
>
> Dear reviewers and action editor,
>
> Thank you very much for your feedback throughout the reviewing process. It has significantly improved our paper, and we now include more experiments on what happens when the constraints on the algorithm are not satisfied and suggestions on how to alleviate such limitations in future work. We also added more reference to work that supports the use of positive weights. Finally, we also addressed other, more minor, comments made by the reviewers.
>
> We have uploaded the camera-ready version and made the GitHub repository public.
>
> The authors